# ADAM-MINI: USE FEWER LEARNING RATES TO GAIN MORE

**Yushun Zhang**[*13], **Congliang Chen**[*13], **Ziniu Li**[13], **Tian Ding**[23], **Chenwei Wu**[4],
**Diederik P. Kingma**[5], **Yinyu Ye**[16], **Zhi-Quan Luo**[13], **Ruoyu Sun**[†123]

[1] The Chinese University of Hong Kong, Shenzhen, China;
[2] Shenzhen International Center for Industrial and Applied Mathematics;
[3] Shenzhen Research Institute of Big Data; [4] Duke University;
[5] Anthropic; [6] Stanford University

## ABSTRACT

We propose Adam-mini, an optimizer that achieves on-par or better performance than AdamW with 50% less memory footprint. Adam-mini reduces memory by cutting down the learning rate resources in Adam (i.e., $1/\sqrt{v}$). By delving into the Hessian structure of neural nets, we find Adam's $v$ might not function at its full potential as effectively as we expected. We find that $\geq 99.9\%$ of these learning rates in $v$ could be *harmlessly* removed if we (1) carefully partition the parameters into blocks following our proposed principle on Hessian structure; (2) assign a single but good learning rate to each parameter block. We then provide one simple way to find good learning rates and propose Adam-mini. Empirically, we verify that Adam-mini performs on par or better than AdamW on various language models sized from 39M to 13B for pre-training, supervised fine-tuning, and RLHF. The reduced memory footprint of Adam-mini also alleviates communication overheads among GPUs, thereby increasing throughput. For instance, Adam-mini achieves $49.6\%$ higher throughput than AdamW when pre-training Llama 2-7B on $2\times$ A800-80GB GPUs, which saves 33% wall-clock time for pre-training [1].

## 1 INTRODUCTION

Adam (Kingma & Ba, 2014) has become the de-facto optimizer for training large language models (LLMs) (e.g., (Vaswani et al., 2017; Achiam et al., 2023; Touvron et al., 2023; Team et al., 2023)). Despite its superior performance, Adam is expensive to use. Specifically, Adam requires the memory for its optimizer states: the first-order momentum $m$, and the second-order momentum $v$. These in total take at least $2\times$ the memory of the model size [2]. This memory consumption has become a major burden in LLM training. For instance, to train a 7B model, Adam alone requires about 56 GB for $m$ and $v$, and with the gradients included, a total of 86 GB is needed. This is expensive even for cutting-edge graphics cards (e.g., A100-80GB). To support training, CPU-offload and optimizer state sharding (Rajbhandari et al., 2020) must be used in practice, which unfortunately increases the latency and slows down the training (Rajbhandari et al., 2021).

It is intriguing to design effective optimizers that require less memory. **First**, it lowers the threshold of training LLMs and encourages participation from more diverse researchers, especially those with limited GPU resources. **Second**, it requires fewer GPUs to train a model with a desired size, leading to substantial savings in both cost and energy. **Third**, it can ease the burden of CPU offloading and model sharding, which in turn, can enhance the throughput and accelerate the training process.

It is challenging to modify Adam without sacrificing its performance. One primary reason is that we still lack understanding of the role of Adam's $m$ and $v$ (Zhang et al., 2020; Kunstner et al., 2023). It remains uncertain which components in Adam are indispensable for superior performance, and which components could be re-designed or improved. One notable attempt is Adafactor (Shazeer & Stern, 2018), which cuts down memory by low-rank factorization on $v$. However, we find that Adafactor is not easy to tune and often performs worse than Adam (see evidence in (Luo et al., 2023) and Section 3.4). One possible reason is that the current $v$ in Adam is crucial and cannot be simplified.

---

∗: Equal contribution. †: Correspondence author.

[1]Our implementation of Adam-mini is available at https://github.com/zyushun/Adam-mini

[2]We restate the update rules of Adam and AdamW (Loshchilov & Hutter, 2017) in Appendix E.1.

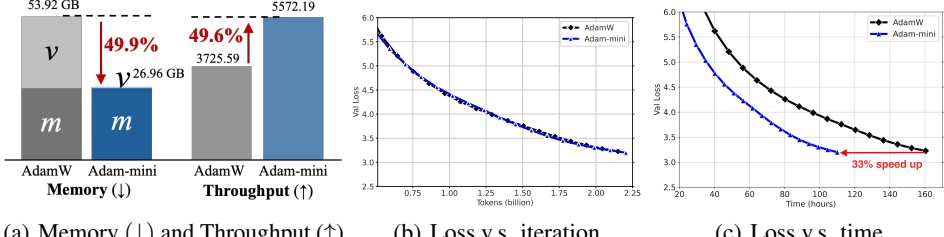

| (a) Memory (↓) and Throughput (↑) | (b) Loss v.s. iteration | (c) Loss v.s. time |

Figure 1: Results for Llama 2-7B pre-training. (a) Adam-mini takes less memory and can reach higher throughput (# tokens per second). The throughput is tested on 2× A800-80GB GPUs. (b, c) Adam-mini performs on-par with AdamW, but takes 33% less time to process the same # tokens.

This is possible as most existing Adam variants that attempt to modify $v$ to varying extents have been reported to perform worse than Adam (Orabona, 2020). Another possible reason is that there is potential to cut down $v$, but Adafactor does not use the most suitable way: matrix factorization is a generic approach that could be applied broadly, but it does not leverage much problem-specific structure, thus it does not work well on specific neural-net tasks.

In this work, we find it is possible to significantly reduce the usage of $v$. Currently, Adam assigns an individual learning rate for each parameter, i.e., $i$-th parameter receives learning rate $\frac{\eta}{\sqrt{v_i}}$, where $v_i$ is the $i$-th component of $v$. For a billion-parameter model, Adam requires billions of learning rates. We argue that it is possible to achieve on-par or better performance with much fewer learning rates. We first recall a classical result that the Hessian of neural nets is near-block-diagonal with several dense princi-

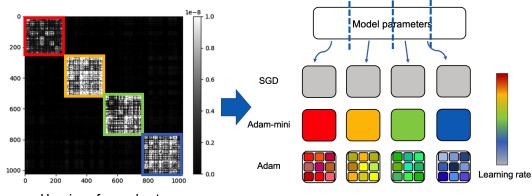

Figure 2: An illustration of Adam-mini. Adam-mini assigns learning rates (lrs) by Hessian structure. It uses more lrs than SGD but fewer than Adam.

ple sub-blocks (Collobert, 2004). We then find that, for each of these dense sub-blocks, there exists a single high-quality learning rate that outperforms Adam, provided that we have enough resources to search it out. Since the number of dense sub-blocks is much fewer than the number of parameters, our findings imply that it is possible to achieve good performance with much fewer learning rates. The remaining question is how to find them efficiently.

We then propose a cheap and simple way to find good learning rates that are sufficient to perform on-par or better than Adam. We introduce the proposed design principle here: we first partition the gradient vector into $B$ sub-vectors according to the dense Hessian sub-blocks, and call it $g_b$ for $b \in \{1, \cdots, B\}$. For each $g_b$, we calculate the quantity below.

$$v_b = (1 - \beta_2) * \texttt{mean}(g_b \odot g_b) + \beta_2 * v_b, \quad b = 1, \cdots B.$$

We then use $\eta/\sqrt{v_b}$ as the learning rate for the parameters in the block associated with $g_b$. Such design changes almost all Adam's $v$ to a negligible amount of scalars and thus reduces the memory. We call the corresponding method Adam-mini. We provide a simple illustration in Figure 2 and relegate the complete form later in **Algorithm 2**. We summarize our main contribution as follows.

- **New optimizer.** We propose a new optimizer called Adam-mini. First, Adam-mini partitions the model parameters based on the principle we established upon the Hessian structure. Then, it chooses a single learning rate for each block using the average of Adam's $v$ in that block. Adam-mini has the following advantages.

  - **Lightweightness:** By design, Adam-mini largely reduces the number of learning rates used in Adam. For mainstream LLMs, Adam-mini could cut down $\geq 99.9\%$ proportion of Adam's $v$, which saves 50% of the memory cost of Adam.

  - **Effectiveness:** Despite the memory cut down, we empirically verify that Adam-mini performs on par or even better than AdamW on various language models sized from 39M to 13B, including pre-training, supervised fine-tuning (SFT), and reinforcement learning from human feedback (RLHF). Adam-mini also performs similarly to Adam on non-LLM tasks such as training diffusion models, vision models, and graph neural networks.

  - **Efficiency:** Adam-mini can reach higher throughput than AdamW. We observe that Adam-mini reaches 49.6% higher throughput of AdamW when pre-training Llama 2-7B on 2× A800-80GB, which saves 33.1% wall-clock time for pre-training. The efficiency comes from two factors. First, Adam-mini does not introduce extra computation in per-step updates.

Second, the memory cut-down allows larger batch sizes per GPU, and at the same time, it eases the burden of communication among GPUs, which is usually a major overhead.

- **Generic partition principle.** A key component in Adam-mini is the strategy for parameter partition. We propose to partition parameters based on the smallest dense sub-block in Hessian. This principle can apply to generic problems with block diagonal Hessian: we find that more learning rates do not necessarily bring extra gain for these problems. In particular, for the problem associated with each dense sub-block, a single (but good) learning rate suffices to bring better performance.

- **Hessian structure and partition principle of Transformers.** We empirically apply the above principle to Transformers. We find that Transformer Hessian's smallest dense blocks are: (1) `query`, `key` by heads; (2) `value`, `attn.proj` and `mlp` by output neurons; (3) `embed` and `output` by tokens. We emphasize that our Hessian-based partition principle is crucial, as naive or default partitions (e.g. partitioning by layers) would cause training instability on LLMs.

## 2 METHOD

### 2.1 MOTIVATIONS AND OBSERVATIONS

Now we discuss our observations that motivate the design of Adam-mini. [3] We start by investigating the role of Adam's $v$ and explore possibilities for improvement. In Adam, $v$ provides an individual learning rate for each parameter, i.e., $i$-th parameter receives the learning rate $\frac{\eta}{\sqrt{v_i}}$, where $v_i$ is the $i$-th component of $v$. Very recently, Zhang et al. (2024) pointed out that such design is crucial for modern architectures such as Transformers. This is because these models often exhibit Hessian-block heterogeneity, i.e., the Hessian of different parameter blocks have dramatically different eigenvalue distributions (We restate their findings in Appendix E.2). This phenomenon suggests that different parameter blocks need different learning rates. This can be provided by Adam's $v$.

The findings in (Zhang et al., 2024) suggest that it is necessary to use a different learning rate for *each block*. Nonetheless, Adam does much more than that: it assigns an individual learning rate not just for each block, but for *each parameter*. Note that the number of parameters is much larger than the number of blocks. This begs the question:

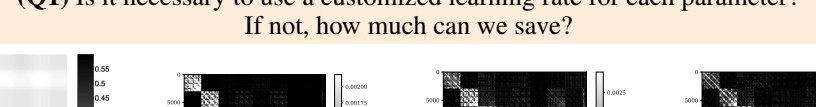

**(Q1)** Is it necessary to use a customized learning rate for each parameter? If not, how much can we save?

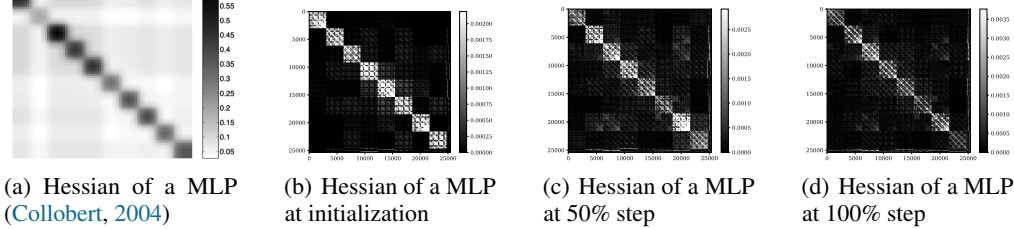

| (a) Hessian of a MLP (Collobert, 2004) | (b) Hessian of a MLP at initialization | (c) Hessian of a MLP at 50% step | (d) Hessian of a MLP at 100% step |

Figure 3: The near-block-diagonal Hessian structure of neural nets. (a) is the Hessian of an MLP after 1 training step reported in (Collobert, 2004). (b,c,d): the Hessians of a 1-hidden-layer MLP on CIFAR-100. The near-block-diagonal structure maintains throughout training, where each block corresponds to one neuron.

To answer **(Q1)**, we delve into the Hessian structures of neural networks. First, we recall an important (but often overlooked) result: the Hessian of neural nets is near-block-diagonal. This is an old result that has been reported for two decades; see (Collobert, 2004, Section 7). The authors also provided theoretical explanations. We restate their analysis in Appendix C. We now provide some case studies.

**Case study I: random quadratic problems.** With the above observation in mind, we now explore **(Q1)** on generic optimization problems with block-diagonal Hessian. We consider the random quadratic minimization problem $\min_w \frac{1}{2} w^\top H w$ where the Hessian $H$ is a random positive definite (PD) matrix and is visualized in Figure 4 (a). We compare the coordinate-wise learning-rate method, i.e., Adam, with the single-learning-rate method, i.e., gradient descent (GD). We choose quadratic minimization because the optimal learning rate has a close form. We have the following findings.

- **(1):** as shown in Figure 4 (a) and (b), Adam outperforms the optimal single-learning-rate method. This is expected since Adam deploys different learning rates to different parameters.

---

[3]All experimental details in Section 2 are shown in Appendix F.2.

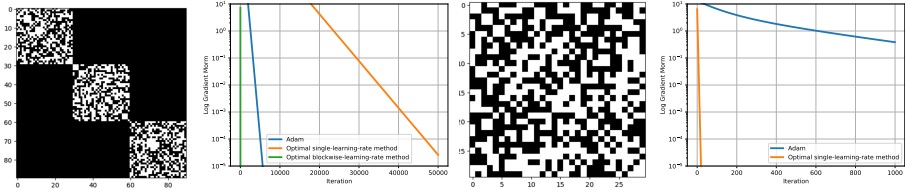

| (a) Hessian matrix | (b) Total loss | (c) Dense sub-block | (d) Sub-problem loss |

Figure 4: **(a):** The Hessian of a three-block random quadratic problem. **(b):** Training curves for the problem associated with the full Hessian in (a). The optimal single (blockwise) learning rate is chosen based on the full (blockwise) Hessian in (a). **(c):** The 1st dense Hessian sub-blocks in (a). **(d):** Training curves for the new problem associated with the Hessian in (c).

- **(2):** as shown in Figure 4 (c) and (d), we consider a new problem whose Hessian is a dense sub-block of (a). We consider the optimal single learning-rate method for this new problem and find it outperforms Adam, even though Adam assigns much more learning rates. Similar phenomena apply to all the three sub-blocks of (a).

- **(3):** If we collect these optimal learning rates in **(2)** and apply them to a "blockwise" version of GD, it would be faster than Adam on the original problem (the green line in Figure 4 (b)).

In summary, for generic problems with block-diagonal Hessian, we find that more learning rates do not necessarily bring extra gain. In particular, **for each dense sub-block, a single (but good) learning rate suffices to bring better performance** than using tens or hundreds more.

**More discussions on case study I.** Why would this happen? We provide one possible explanation from a linear algebra perspective. Adam can be viewed as a diagonal preconditioned method , i.e., at the $t$-th step:

$$w_{t+1} = w_t - \eta_t D_t m_t, \tag{1}$$

where $D_t = \mathrm{Diag}(1/\sqrt{v_t})$ is a diagonal matrix, $m_t$ is the 1st-order momentum, $w_t$ and $\eta_t$ are model parameters and learning rate. However, Adam may not be an optimal preconditioner and thus cannot effectively reduce the condition number of the dense sub-matrix. In the field of optimization, the effectiveness of a diagonal preconditioner $D$ is often measured by "how much is $\kappa(DH)$ reduced over $\kappa(H)$", where $H$ usually refers to the Hessian matrix and $\kappa(\cdot)$ is the condition number (smaller is better). Unfortunately, there is no guarantee of $\kappa(DH) \leq \kappa(H)$ and this inequality often requires strict assumptions on both $D$ and $H$. For instance, $\kappa(DH)$ would be small if $H$ is close to diagonal and $D$ is a cleverly designed compressor of $H$ (Forsythe & Straus, 1955; Young, 1954; Sun & Ye, 2021; Qu et al., 2022).

Here, we numerically explore the effectiveness of Adam's preconditioner within each dense Hessian sub-block. We generate a random dense PD matrix $H_b \in \mathbb{R}^{d \times d}$ and use it as a proxy for the dense Hessian sub-block of neural nets in Figure 3. We define $D_{\mathrm{Adam}} = \mathrm{Diag}(1/\sqrt{v})$, where $v = g \odot g$, $g = H_b x \in \mathbb{R}^d$, and each entry $x_i \sim \mathcal{N}(0, 1/\sqrt{d})$ follows Xavier initialization. We explore the interplay between the following two metrics:

$$\tau = \frac{\sum_i |H_{b,i,i}|}{\sum_{i,j} |H_{b,i,j}|}, \quad r = \frac{\kappa(D_{\mathrm{Adam}} H_b)}{\kappa(H_b)}, \tag{2}$$

where $\tau \in [0, 1]$ is the "diagonal-over-off-diagonal ratio", and we use it to measure how dense $H_b$ is ($H_b$ is pure diagonal when $\tau = 1$). $r \geq 0$ measures the effectiveness of Adam's preconditioner $D_{\mathrm{Adam}}$ when operating on the Hessian-block $H_b$ (the smaller the better). We investigate the change of $r$ when changing the structure of $H_b$, including changing $\tau$, dimension $d$, and also $\kappa(H_b)$. We emphasize that for a fixed $d$ or $\kappa(H_b)$, we change $\tau$ by only rotating the eigenvectors, but not changing the eigenvalues of $H_b$. This ensures $\tau$ is the only changing factor in the experiments.

We summarize the key findings in Figure 5: for $H_b$ with most dimension $d$ and $\kappa(H_b)$, $r$ decreases as $\tau \to 1$. That is, $D_{\mathrm{Adam}}$ is effective when $H_b$ is close to diagonal, and $D_{\mathbf{Adam}}$ **is not so effective when** $H_b$ **is dense.** This aligns with the convergence rates in Figure 4. It is intriguing to provide a lower bound on $\kappa(D_{\mathrm{Adam}} H_b)$ to ground the observation in Figure 5, and we are not aware of any existing lower bound of this kind. Note that it is rather difficult to characterize $\kappa(D_{\mathrm{Adam}} H)$, partially because the extreme eigenvalues are neither sub-additive nor sub-multiplicative (Kittaneh, 2006). We leave it as an important but challenging future direction. **To summarize, for the dense Hessian-blocks, it is possible to outperform Adam with only one good learning rate.**

**Case study II: Transformers.** The above analysis suggests there is room to cut down the number of learning rates. We also observe similar phenomena in Transformers. We consider a 4-layer Transformer and under the PyTorch default partition, and we randomly choose one parameter block as

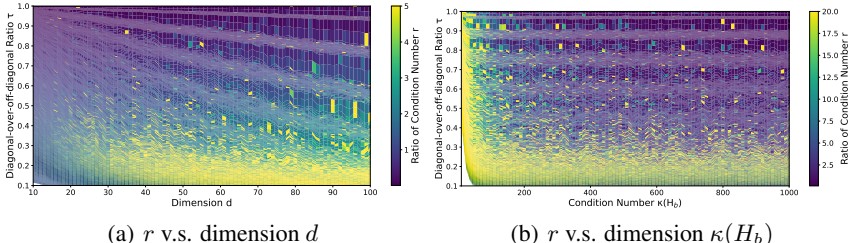

(a) $r$ v.s. dimension $d$        (b) $r$ v.s. dimension $\kappa(H_b)$

Figure 5: The effectiveness of Adam's preconditioner $D_{\text{Adam}}$ on different matrix structures of $H_b$. (a): for most dimension $d$, $r$ is large when $\tau$ is small ( $r$ and $\tau$ are defined in Eq. (2)). This indicates that Adam might not be so effective when $H_b$ is dense. We fix $\kappa(H_b) = 500$ here. (b): We use the same setups as (a), except that we fix the dimension $d = 50$ and change the $x$-axis to $\kappa(H_b)$.

the "left-out" block and change the coordinate-wise learning rate to a single-learning rate counter-part. We use Adam for the rest of the blocks. We grid-search the learning rate for the left-out block and apply the cosine decay schedule like the rest of the blocks. We report the best result and call this method "Adam (leave-one-out)". Figure 6 shows that Adam (leave-one-out) can achieve similar or better performance than Adam. A similar phenomenon is also observed when we randomly leave out up to three blocks and search three learning rates. We do not explore the possibility of leaving more blocks out since the cost of grid search grows exponentially.

To summarize this section, we find that it is possible to reach similar or better performance with much fewer learning rates than Adam. The remaining issue is how to find them without grid-search. In the next part, we propose a simple and effective method called Adam-mini, which could bring comparable or even better performance than Adam, but with 99.9% fewer learning rates.

## 2.2 Proposed Method: Adam-mini

We now introduce Adam-mini. We will first state the **"general principled form"** of Adam-mini and then introduce the **"the realization"** of Adam-mini on specific architectures. In this section, we present the general form of Adam-mini in **Algorithm 1**. Following this general principled form, Adam-mini will have different realizations on different architectures, and the concrete example on Transformers is shown in Appendix B. As shown in **Algorithm 1**, Adam-mini contains two steps.

---

**Algorithm 1** Adam-mini (General form)

1: Input weight-decay coefficient $\lambda$ and current step $t$
2: Partition params into `param_blocks` by **Principle 1** in Section 2.3
3: **for** `param` in `param_blocks` **do**
4:    `g = param.grad`
5:    `param = param - `$\eta_t$` * `$\lambda$` * param`
6:    `m = `$(1 - \beta_1)$` * g + `$\beta_1$` * m`
7:    $\hat{\text{m}} = \frac{\text{m}}{1 - \beta_1^t}$
8:    `v = `$(1 - \beta_2)$` * mean(g`$\odot$`g) + `$\beta_2$` * v`
9:    $\hat{\text{v}} = \frac{\text{v}}{1 - \beta_2^t}$
10:   `param = param - `$\eta_t$` * `$\frac{\hat{\text{m}}}{\sqrt{\hat{\text{v}}} + \epsilon}$
11: **end for**

---

**Step 1** Partition the model parameters into blocks by Hessian structure. We discuss **Principle 1** later in Section 2.3. For different architectures, the principle will be realized in different forms; see **Algorithm 3**: "Partition for non-Transformers". and **Algorithm 3**: "Partition for Transformers".

**Step 2.** For each parameter block, we use a single learning rate. To efficiently choose a suitable learning rate in each block, Adam-mini simply replaces g$\odot$g in vanilla Adam by its mean value. We adopt the moving average on these mean values as in Adam.

**A simple example of Adam-mini.** We use a simple example to illustrate the key design of Adam-mini. For a problem with 5 parameters $w \in \mathbb{R}^5$, Adam and Adam-mini both perform $w = w - u \odot m$, where $m$ is the 1st-order momentum and $u$ has different forms as follows:

- For Adam: $u_{\text{Adam}} = \left( \frac{\eta}{\sqrt{v_1}}, \frac{\eta}{\sqrt{v_2}}, \frac{\eta}{\sqrt{v_3}}, \frac{\eta}{\sqrt{v_4}}, \frac{\eta}{\sqrt{v_5}} \right)$.

- For Adam-mini: suppose the partition is $(1, 2, 3)$ and $(4, 5)$ then

$$u_{\text{mini}} = \left( \frac{\eta}{\sqrt{(v_1 + v_2 + v_3)/3}}, \frac{\eta}{\sqrt{(v_1 + v_2 + v_3)/3}}, \frac{\eta}{\sqrt{(v_1 + v_2 + v_3)/3}}, \frac{\eta}{\sqrt{(v_4 + v_5)/2}}, \frac{\eta}{\sqrt{(v_4 + v_5)/2}} \right).$$

Note that the number of effective elements $u_{\text{mini}}$ equals the number of blocks, which could be significantly smaller than that of $u_{\text{Adam}}$, which equals the number of parameters. For LLMs, this will free $\geq 99.9\%$ elements in $v$.

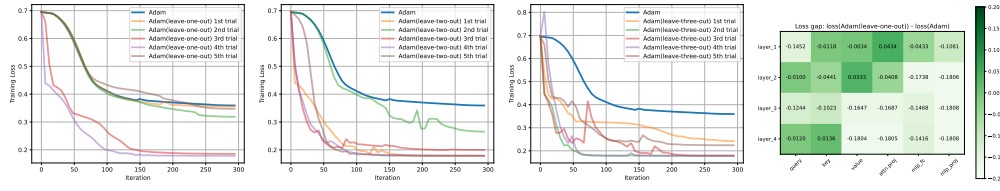

(a) Randomly leave one block out (b) Randomly leave two blocks out (c) Randomly leave three blocks out (d) Performance gap

Figure 6: (a) (b) (c) Adam (leave-$x$-out) can reach a similar or better performance than Adam for all randomly picked left-out blocks. $x = 1, 2, 3$. (d) The performance gap between Adam and Adam (leave-one-out) for all possible blocks. We find Adam (leave-one-out) always performs on par with Adam, and for most blocks, Adam (leave-one-out) performs better. It seems possible to perform well with much fewer learning rates than Adam.

## 2.3 PRINCIPLE FOR THE PARTITION STRATEGY

We now discuss how to choose the parameter partition for Adam-mini. A straightforward way is to use PyTorch default partition. Unfortunately, we find that the PyTorch default partition does not work well on larger-scaled tasks. In particular, we find that Adam-mini encounters training instability on 1B models (see Figure 7 (i)). We suspect this is because the default PyTorch partition did not fully capture the Hessian structure. We propose a general principle in **Principle 1** below.

> **Principle 1:** We should partition parameters into blocks, such that each parameter block is associated with the smallest dense sub-block in Hessian.

**Principle 1** comes from the analysis in Section 2.1: it is possible to harmlessly reduce the number of Adam's learning rates within each dense Hessian block. However, if the partition is too coarse and violates **Principle 1**, we might accidentally remove some crucial learning rates and oversimplify the problem, causing training failure. It is important to follow **Principle 1** since it is necessary to use (at least) one distinct learning rate for each Hessian block (as evident in Appendix E.2).

Does the PyTorch default partition follow **Principle 1**? To find out, we explore the Hessian of a small Transformer as in Figure 7. Under the default PyTorch partition, we compute the Hessian for each parameter block after 1 training step. We find four classes of Hessian sub-blocks.

- **Class 1: `query` and `key`.** The Hessian of `query` and `key` have near-block-diagonal structures. . The number of blocks equals the number of heads.

- **Class 2: `attn.proj` and MLPs.** The Hessian of `attn.proj` and MLPs have block-diagonal structures. The number of blocks equals the number of output neurons.

- **Class 3: `value`.** For `value`, the structure of Hessian seems less clear. It seems to have the hint of 16 diagonal blocks (16 is the number of output neurons), but the pattern is less obvious. This Hessian structure is significantly different from that of `query` and `key`, although they all consist of four heads. The Hessian entries of `value` are also about $10^6$ larger than those of `query` and `key` [4]. One possible reason is that `value` is positioned outside the softmax operator in the self-attention design, while `query` and `key` are not.

- **Class 4: `embed` and `output`.** For these two layers, the Hessian sub-block has a near-block-diagonal structure and the number of blocks equals the number of tokens.

Based on the above findings, we find that the PyTorch default partition is indeed not the best fit for Transformers. By **Principle 1**, `query` and `key` should be further partitioned by heads; `value`, `attn.proj`, and MLPs should be partitioned by output neurons; `embed` and `output` should be partitioned by tokens. As for `value`, the Hessian shows the hint of 16 diagonal blocks (where 16 is the number of output neurons), but the pattern is less clear. Our experiments show that "partition `value` by output neurons" works well in general, yet there are also some special cases where it is better to "treat `value` as a whole" (see discussions in Appendix D.6). By default, we will partition `value` by output neurons.

We then introduce the resulting **Algorithm 3**: "Partition for Transformers" in Appendix B. As shown in Figure 7 (i). This strategy indeed stabilizes the training and boosts the performance.

## 2.4 SOME CHARACTERISTICS OF ADAM-MINI AND DISCUSSIONS

**Memory cut down.** Adam-mini reduces the number of learning rates from the number of model parameters to the number of total number of blocks by our partition strategies. As a result, Adam-mini cuts down more than 99.9% of Adam's $v$, which saves 50% of Adam's memory.

---

[4]This might be one source of the heterogeneity of Hessian eigenvalues as reported by (Zhang et al., 2024).

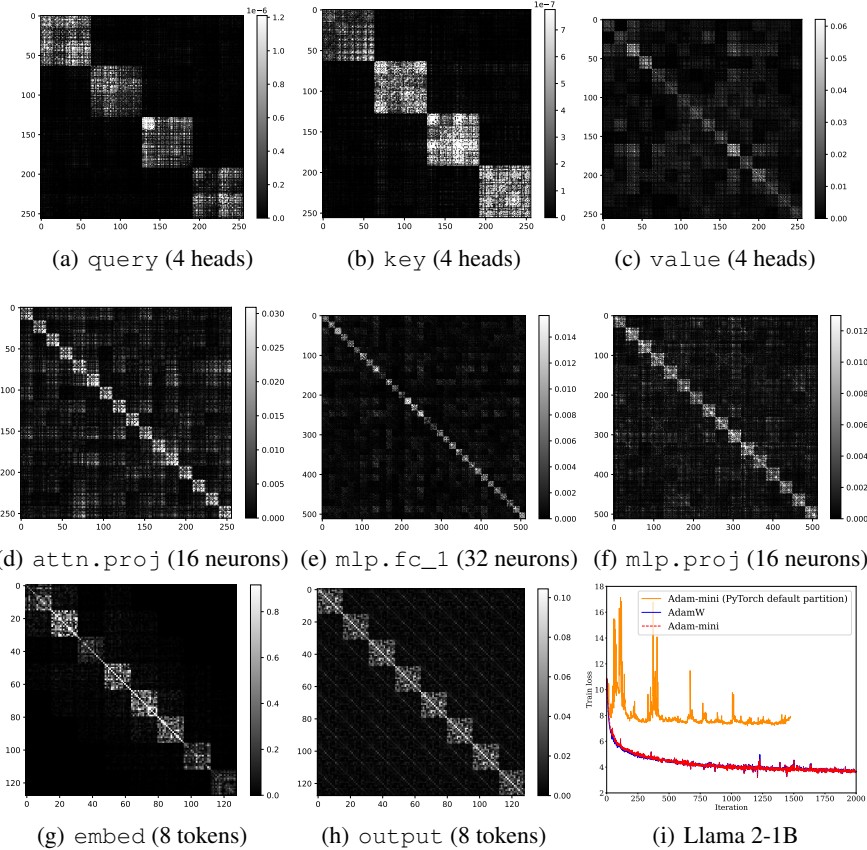

Figure 7: (a-h): The Hessian of different parameter blocks in a small Transformer at 1% training step. Here, neuron refers to the "output neuron". We find that these Hessian sub-blocks have near-block-diagonal structure and consists of smaller dense matrices. Different parameter blocks have different numbers of small dense matrices. (i) Training curves on Llama 2-1B. When using the PyTorch default partition, Adam-mini could suffer loss spikes. The spike disappears when we use the partition strategy in **Algorithm 3**.

Table 1: Memory cost of AdamW v.s. Adam-mini. Calculation is based on `float32`, which is a standard choice for optimizer states.

| Model | Optimizer | Memory (GB) |
|---|---|---|
| GPT-2-1.5B | AdamW | 12.48 |
| GPT-2-1.5B | Adam-mini | 6.24 (**50%** ↓) |
| Llama 2-1B | AdamW | 8.80 |
| Llama 2-1B | Adam-mini | 4.40 (**50%** ↓) |
| Llama 2-7B | AdamW | 53.92 |
| Llama 2-7B | Adam-mini | 26.96 (**50%** ↓) |
| Llama 3-8B | AdamW | 64.24 |
| Llama 3-8B | Adam-mini | 32.12 (**50%** ↓) |
| Llama 2-13B | AdamW | 104.16 |
| Llama 2-13B | Adam-mini | 52.08 (**50%** ↓) |

Table 2: Throughput (↑) test on $2\times$ A800-80GB GPUs for Llama 2-7B pre-training. ✗means out of memory. GPU hours (↓) to pre-train Llama 2-7B with the optimal token amount by Chinchila's law.

| Optimizer | bs_per_GPU | total_bs | Throughput (↑) |
|---|---|---|---|
| Adam-mini | 4 | 256 | 5572.19 (↑ **49.6%**) |
| AdamW | 2 | 256 | ✗ |
| AdamW | 1 | 256 | 3725.59 |

| Optimizer | # Tokens (B) | GPU hours (h) (↓) |
|---|---|---|
| AdamW | 1 | 74.56 |
| Adam-mini | 1 | 49.85 (↓ **33.1%**) |
| AdamW | 70 | 5219.16 |
| Adam-mini | 70 | 3489.55 (↓ **33.1%**) |
| AdamW | 140 | 10438.32 |
| Adam-mini | 140 | 6979.10 (↓ **33.1%**) |

**Higher throughput.** Adam-mini can reach a higher throughput than AdamW, especially under limited GPU resources. There are two reasons. First, Adam-mini does not introduce extra computation in its update rules. The averaging operation in **Algorithm 1** incurs negligible cost and it significantly reduces the number of vector-square-root and vector-division operations in AdamW. Second, thanks to the memory cut-down, Adam-mini can support larger batch sizes per GPU. It also reduces the communication among GPUs, which is known to be a major overhead (Rajbhandari et al., 2021). We report evidence in Table 2. When pre-training Llama 2-7B on $2\times$ A800-80GB GPUs, we find Adam-mini could reach 49.6% higher throughput than AdamW. This translates to **33.1% reduction of wall-clock time** on processing the same amount of tokens for pre-training.

**Why using mean($v$) as learning rates.** Due to limited space, we move the discussions to Appendix C.

**Has room to improve.** Adam-mini designs the learning rate for each dense Hessian sub-block using the average of Adam's $v$ in that block. Such a design achieves cheap computation, but it might not be

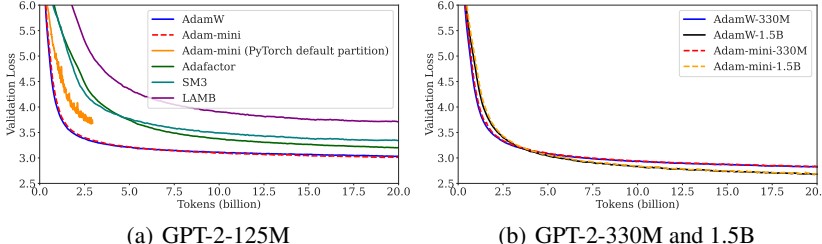

(a) GPT-2-125M | (b) GPT-2-330M and 1.5B

Figure 8: For GPT-2 series pre-training, Adam-mini performs similarly to AdamW with 50% less memory, while other methods perform worse.

optimal. We believe there is great room to improve the learning rate design. As shown in Figure 4, we can reach much faster convergence if we utilize more information in the dense block to design the learning rate (e.g., using eigenvalues of each block), However, such a design requires expensive computation. We leave it as an important future direction.

## 3 EXPERIMENTS

We now verify the efficacy of Adam-mini on two types of neural-net tasks: (1) LLM tasks including pre-training, supervised fine-tuning (SFT), and reinforcement learning from human feedback (RLHF). (2) Non-LLM tasks including vision, graph, and diffusion model training. Due to the limited space, we primarily focus on LLM tasks in this section, **and we relegate the non-LLM tasks to Appendix D.5.** All LLM experiments are conducted on four NVIDIA A800-80GB GPUs and the rest are conducted on four V100 GPUs. All the experimental details are explained in Appendix F.1.

### 3.1 PRE-TRAINING

**Setups.** We pre-train LLMs including GPT-2 series and Llama series. We train these models on mainstream English Corpus from scratch. In particular, We train GPT-2 (Radford et al., 2019) series (125M to 1.5B) on Openwebtext (Gokaslan et al., 2019). We train Llama series (20M to 13B) (Touvron et al., 2023) on C4 (Raffel et al., 2020). We compare Adam-mini with AdamW (Loshchilov & Hutter, 2017) as well as popular memory-efficient methods including Adafactor (Shazeer & Stern, 2018), CAME (Luo et al., 2023), and SM3 (Anil et al., 2019). For Adafactor and SM3, we incorporate momentum with $\beta_1 = 0.9$ to ensure a fair comparison with other methods. We tune the learning rate for all methods, using the same tuning budget for each, and report the best performance.

**GPT-2 series.** Figure 8 shows the results for GPT-2 series pre-training. We find that Adam-mini performs similarly to AdamW with 50% less memory, while other methods perform worse. In Figure 8 (a), we run *Adam-mini (PyTorch default partition)*, which partition parameters by PyTorch default partition. We find that *Adam-mini (PyTorch default partition)* performs poorly. We stop the trial since it shows clear unstable behavior. In Figure 9, we further present the training loss curves. We find that **the loss curves of Adam-mini closely resemble those of AdamW.**

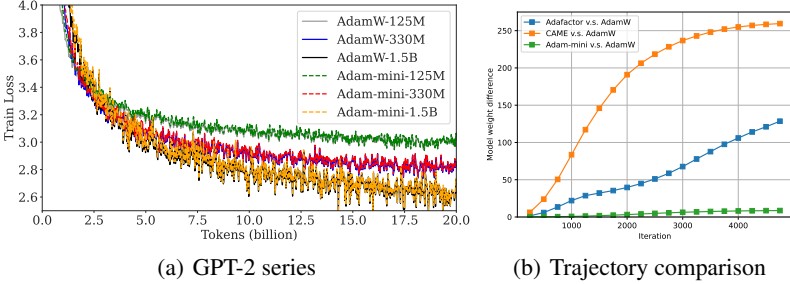

(a) GPT-2 series | (b) Trajectory comparison

Figure 9: (a): The training loss curves of Adam-mini closely resemble those of AdamW. (b): The trajectory of Adam-mini stays close to the trajectory of AdamW (in terms of the $\ell$-2 distance of model checkpoints).

**Llama series.** Figure 10 shows the results for pre-training Llama series. We also train Llama 2-7B as shown in Figure 1 (c) in Section 1. We find that Adam-mini performs on par with AdamW, while other methods do not. **Further, Adam-mini's loss curves closely resemble the curves by AdamW.**

**Trajectory comparison.** On a small Transformer, Adam-mini generates similar trajectories to that of AdamW, while other methods cannot. This can be seen in Figure 9 (b) and the detailed description is in Appendix F. This might be because Adam-mini makes fewer modifications over AdamW.

**Sensitivity analysis.** On GPT-2-125M pre-training task, we test the sensitivity of Adam-mini to hyperparameters. We report the validation loss after training with 2.5B tokens (by Chinchilla's law). As shown in Figure 12 (c), Adam-mini seems not overly sensitive to hyperparameters.

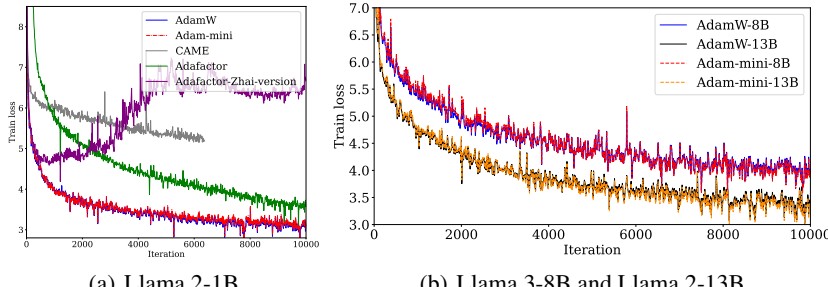

(a) Llama 2-1B      (b) Llama 3-8B and Llama 2-13B

Figure 10: (a, b): Loss curves of pre-training Llama series from 1B to 13B. Adam-mini performs on par or better than AdamW with 50% less memory, while other methods perform worse. Further, loss curves of Adam-mini closely resemble the curves by AdamW.

## 3.2 Scaling Laws of Adam-mini

We now show the efficacy of Adam-mini through scaling law experiments. We use C4 dataset to pre-train the Llama 2 architecture from 39M to 1B. For the model with size $n_{param}$, we train the model with about $20 * n_{param}$ tokens, which is suggested to be the optimal amount by Chinchilla's law (Hoffmann et al., 2022). The largest-scaled experiment we conducted is Llama 2-1B pre-training with 26.2B tokens, which takes about 170 GPU hours on $4\times$ A800-80GB GPUs. The total running time for the scaling law experiments is about 300 GPU hours.

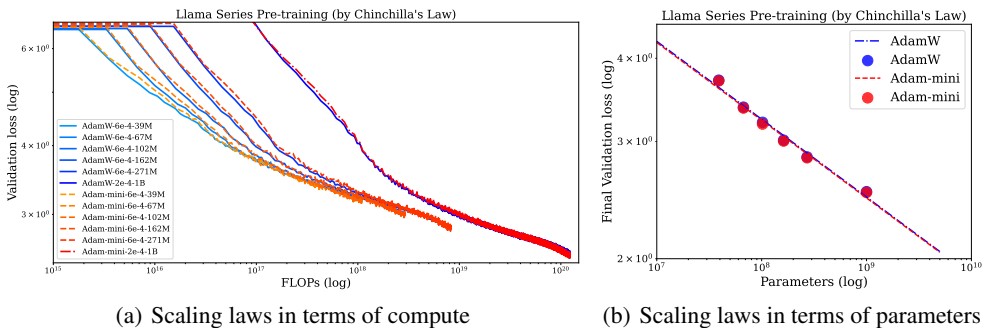

(a) Scaling laws in terms of compute     (b) Scaling laws in terms of parameters

Figure 11: (a, b): Scaling laws of Adam-mini. We pre-train Llama 2 architectures by Chinchilla's law. For all models sized from 39M to 1B, Adam-mini's loss curves are consistently similar to AdamW, but Adam-mini uses 50% less memory. Further, as shown in (b), Adam-mini reaches a lower final loss than AdamW for all models. The fitted lines in (b) suggest that Adam-mini can be scaled up to larger models (if the scaling law holds).

As shown in Figure 11, **Adam-mini's loss curves are consistently similar to AdamW.** We also present the final validation perplexity and find that Adam-mini reaches **a slightly lower perplexity** than AdamW for all models (see Figure 11 (b), also see Table 4 in Appendix D). The fitted lines in Figure 11 (b) suggest that Adam-mini can be scaled up to larger models (if the scaling law holds).

Another advantage of Adam-mini is its ability to **reduce computational costs for scaling law experiments**. Scaling law experiments are typically used to predict the optimal configurations for large-scale models by fitting the performance of smaller-scale proxy models. To accelerate the development of large-scale models, it is crucial to minimize costs during the fitting process (Hägele et al., 2024). Adam-mini achieves this by delivering the same scaling results while using significantly less memory and time cost (e.g., 33% less GPU hours, as shown in Figure 1).

## 3.3 Supervised Fine-tuning and RLHF

We now test Adam-mini on SFT and RLHF. We use the Llama 2-7B pretrained model (Touvron et al., 2023) for our study. We use the `ultrafeedback` dataset and implement the RLHF workflow from (Ouyang et al., 2022). We use ReMax (Li et al., 2023), a memory-efficient alternative to PPO

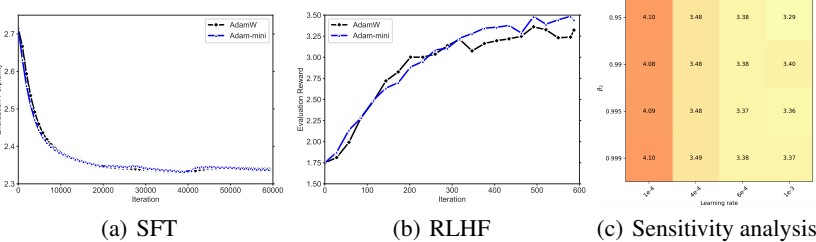

(a) SFT      (b) RLHF      (c) Sensitivity analysis

Figure 12: (a, b): SFT, and RLHF when aligning Llama 2-7B. Adam-mini reaches better performance (smaller perplexity, higher reward) than AdamW with 50% less memory. (c) Sensitivity analysis of Adam-mini on GPT-2-125M pre-training (by Chinchilla's law). Adam-mini seems not overly sensitive to hyperparameters.

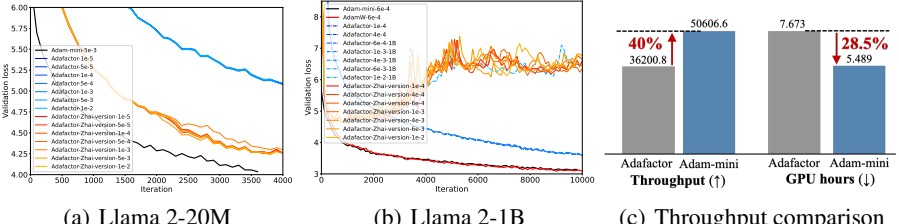

|(a) Llama 2-20M|(b) Llama 2-1B|(c) Throughput comparison|

Figure 13: (a, b): Training curves of Adafactor & Adafactor-Zhai-version on Llama 2-20M & Llama 2-1B pre-training. We find these two methods consistently underperform Adam-mini. (c): On Llama 2-1B, Adam-mini achieves 40% higher throughput than Adafactor (tested on $2\times$ A800-80GB GPUs).

(Schulman et al., 2017), to optimize the preference reward. As shown in Figure 12, Adam-mini performs on par or better than AdamW. Adam-mini also achieves better alignment performance on MT-Bench using GPT-4 as a judge. The results are shown later in Table 5 in Appendix D.4.

### 3.4 DETAILED COMPARISON WITH ADAFACTOR

We now carefully compare Adam-mini and the popular memory-efficient optimizer Adafactor. Besides the original Adafactor, we also consider a modified version in (Zhai et al., 2022), which we call "Adafactor-Zhai-version". For both versions, we use momentum with $\beta_1 = 0.9$.

We first conduct learning rate grid-search on Llama 2-20M and train it following Chinchilla's law. As shown in Figure 13 (a), we find that Adafactor-Zhai-version improves over the original version, but both versions of Adafactor are still consistently worse than Adam-mini. We further sweep over other hyperparameters including (1) $\beta_2 = 0.95$; (2) $\epsilon = \{10^{-30}, 10^{-16}, 10^{-8}, 10^{-6}\}$; (3) warm-up steps = $\{1\%, 2\%, 3\%, 4\%, 5\%, 10\%\}$ total steps. The results are shown in Appendix D.7. We find that the change of hyperparameters does not significantly boost the performance of Adafactor, and both versions still underperform Adam-mini.

We further sweep hyperparameters on Llama 2-1B. In contrast to the case of Llama 2-20M, we find that the Adafactor-Zhai-version now suffers from training instability and the original version performs better. Nevertheless, they still underperform Adam-mini. In Appendix D.8, we conduct a similar hyperparameter search for Lion (Chen et al., 2024b) and we find it also underperforms Adam-mini.

**About hyperparameter tuning.** We acknowledge that it might be possible to improve these methods if we spend more resources on grid search (as claimed by a recent work (Zhao et al., 2024b)). However, based on our experience so far, it is not easy to tune these methods, and to our knowledge, there is no much open-source guidance. Recall that **there are 9 tunable hyperparameters in Adafactor**, so it is rather non-trivial to find the correct combination. In contrast, Adam-mini is much easier to use. In all our experiments, Adam-mini performs well using **the same hyperparameters as AdamW** (including learning rate, $\beta_1, \beta_2, \epsilon$, etc.).

**Throughput comparison.** Besides the performance comparison, we further find that Adafactor has a higher latency than Adam-mini (Figure 13 (c)). This is primarily due to two reasons. First, Adam-mini only requires computing the mean by rows of the weight matrix, whereas Adafactor needs to sum across both the rows and the columns. Second, the dimension of $v$ in Adam-mini equals the output dimension or the number of heads, which is significantly smaller than the dimension of $v$ in Adafactor, which equals the product of the input and output dimension. Note that similar latency issues also apply to other variants of Adafactors such as CAME. In contrast, Adam-mini saves computation when taking the square root of $v$. As such, Adam-mini reaches a higher throughput.

**Summary of Section 3.** Finally, we summarize three key observations from all the experiments above. For all the models we tried, we observed that:

1. Adam-mini performs on par with AdamW with **50% less memory**.

2. Adam-mini performs well using **the same hyperparameters** as AdamW.

3. Adam-mini's loss curves **closely resemble** those of AdamW.

### 4 CONCLUDING REMARKS

We proposed Adam-mini, an optimizer that saves 50% memory of Adam. We remark that there is great room to improve the design of Adam-mini: currently Adam-mini uses a simple and cost-effective way to design a learning rate for each dense Hessian sub-block, but it might not be an optimal way. We leave the development of stronger designs as a future direction.

## ACKNOWLEDGEMENT

Yushun Zhang would like to sincerely thank Less Wright, Andrew Gu, and other open-source contributors for their invaluable support in integrating Adam-mini into the mainstream codebases. The work of Ruoyu Sun was supported by NSFC (No. 12326608); Hetao Shenzhen-Hong Kong Science and Technology Innovation Cooperation Zone Project (No. HZQSWS-KCCYB-2024016), together with Tian Ding; University Development Fund UDF01001491, the Chinese University of Hong Kong, Shenzhen; Guangdong Provincial Key Laboratory of Mathematical Foundations for Artificial Intelligence (2023B1212010001). The work of Z.-Q. Luo was supported by the Guangdong Major Project of Basic and Applied Basic Research (No.2023B0303000001), the Guangdong Provincial Key Laboratory of Big Data Computing, and the National Key Research and Development Project under grant 2022YFA1003900. The work of Tian Ding is also supported by Internal Project of Shenzhen Research Institute of Big Data under Grant J00220240005.

## BROADER IMPACT

This work designs a new algorithm to reduce the training cost for LLMs. Our algorithm can help the community better train large AI models. However, it could be a potential threat if the AI models are used for illegal usage.

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

## A  RELATED WORKS

**Understanding of Adam.** There is an active line of works trying to understand why Adam works well (Zhang et al., 2019b; Wu et al., 2020; Shi et al., 2020; Zhang et al., 2022; Wang et al., 2022; Pan & Li, 2023; Jiang et al., 2023; Kunstner et al., 2023; Zhang et al., 2024; Ahn et al.; Kunstner et al., 2024). In contrast to these works, we point out that Adam's $v$ might not function at its full potential as effectively as we expected: sometimes fewer learning rates can reach the same or better results (due to the dense Hessian sub-blocks). Our findings might motivate stronger optimizers that better fit the neural-net Hessian structure.

Similarly to in our Section 2.1, a recent work (Das et al., 2024) also explores the effectiveness of Adam's preconditioner $D_{\text{Adam}}$ from a linear algebra perspective. They focus on (a variant of) Adam and prove the following result: First, for diagonal dominant (DD) matrix $H_b$ when the dimension $d$ is less than $\kappa^{1/3}$, their modified version of Adam exhibits a faster convergence rate compared to gradient descent (GD); Second, for non-DD matrix, the constant terms in Adam's upper bound can be much larger than that of GD. Their results take a valuable and important step towards understanding $D_{\text{Adam}}$. However, these results cannot fully support our numerical findings in Figure 5. This is because they only provide an upper bound for the non-DD case, while we need a lower bound. We note that it is rather difficult to derive the desired lower bound, and we leave it as a future direction.

**On the Hessian of Neural Nets.** Hessian matrix is crucial for the behaviors of gradient methods. There are several important attempts to study the Hessian of MLPs and CNNs (Collobert, 2004; Roux et al., 2007; Martens & Grosse, 2015; Sagun et al., 2016; 2017; Chaudhari et al., 2019; Papyan, 2020; Wu et al., 2020; Liao & Mahoney, 2021; Papyan, 2018; 2019; Sankar et al., 2021; Gur-Ari et al., 2018; Yao et al., 2018; Zhang et al., 2019a; Ghorbani et al., 2019; Yao et al., 2020; Dauphin et al., 2024). Inspired by these works, we explore the Hessian structure of Transformers and connect it to the behaviors of Adam. We then find room to improve and propose to slim down Adam into Adam-mini.

**Lightweight optimizers for general tasks.** There are several attempts to reduce the memory cost of Adam. Adafactor (Shazeer & Stern, 2018) and its variant CAME (Luo et al., 2023) conduct nonnegative low-rank factorization over Adam's $v$. SM3 (Anil et al., 2019) is a lightweight version of AdaGrad (Duchi et al., 2011). SM3 chooses the learning rate of the $i$-th parameter by taking the minimal value in a certain candidate set, and each element in the candidate set is related to the maximal squared gradient under a predetermined cover. All these aforementioned methods could release almost all memory for $v$ and save about 48% of Adam's memory. However, we find that their performance degenerate in various experiments, while Adam-mini maintains as effective as AdamW (as shown in Section 3).

After completing this work, we noticed two methods that share some of the ideas of Adam-mini: BAGM (Zheng & Kwok, 2019) and NovoGrad (Ginsburg et al., 2019). Both of them use block-wise or layer-wise adaptive learning rates to achieve robust performance and better generalization. We summarize their key differences with Adam-mini. BAGM partitions parameters to reach minimal-norm solutions and achieve provable robustness. In particular, their theory in Proposition 1 states that layer-by-layer parameter partition can lead to minimum $\ell$-2 norm solutions. Aligning with the theory, they find that the PyTorch default partition (BAGM-B.1) indeed brings overall the best performance on both CIFAR-10 and ImageNet. Although the PyTorch default partition may have benefits on robustness, we find that it overlooks the Hessian structure and oversimplifies the training problem for Transformers (as we discussed in Section 2.3). As a result, the PyTorch default partition will lead to training instability in large-scale LLMs, and this is evident in our failed preliminary versions of Adam-mini in Figure 7 and 8. We then propose a new partition strategy **Algorithm 3** which partition parameters by the smallest dense Hessian sub-blocks. For Transformers, **Algorithm 3** uses different strategies for different building blocks (e.g., partition the embedding layer by tokens, and partition Query by heads) and we find that **Algorithm 3** is necessary to stabilize the training.

As for NovoGrad, it also uses a layer-wise learning rate design (by PyTorch default partition) and thus face the same training instability issues as BAGM. Further, NovoGrad introduces a different design to 1st-order momentum: instead of performing weighted-sum on the past gradients, it performs weighted-sum on "the current gradient divided by the 2nd-order momentum". Such design is largely different from AdamW and Adam-mini. It remain unclear whether this design can work on large-scale tasks like LLMs.

Besides algorithmic design, our work also provides new understandings of Adam, and particularly, how Adam behaves on generic optimization problems with near-block-diagonal Hessian. We also provide new findings on the Hessian structure of Transformers and provide new principles for designing better algorithms.

**Other orthogonal methods.** The idea of Adam-mini can be orthogonally combined with various existing methods. We list two most relevant examples here.

1. GaLore (Zhao et al., 2024a) is a new memory-efficient optimizer for LLMs. Given a gradient matrix $g$, GaLore calculates a low-rank gradient estimator $\hat{g}$ and then calculates $m$ and $v$ based on this $\hat{g}$. Adam-mini can potentially be combined with GaLore to reach further memory reduction on $v$. The combined method, e.g., "GaLore-mini", can further reduce about $40\%$ memory on GaLore and about $81\%$ on AdamW in total.[5] Additionally, GaLore-mini can ease the offload burden and enhance the throughput of GaLore, especially when training on customer-level GPUs with limited memory.

2. Sophia (Liu et al., 2023) is another recent diagonal preconditioned optimizer. Just as Adam, Sophia requires memory for $m$ and $v$. It is possible to combine Adam-mini and Sophia to get "Sophia-mini", which saves up to 50% of memory in Sophia. Sophia-mini can also enhance throughput and further speed up Sophia on wall-clock time as in Table 2.

We list more potential combinations here. LoRA (Hu et al., 2021) is a memory-efficient method for SFT tasks. This method fine-tunes the model via additive low-rank adaptors and uses Adam to update these adaptors. Note that the Adam steps in LoRA can be replaced by Adam-mini. As a result, Adam-mini brings better performance (Figure 22). In parallel to our work, BAdam (Luo et al., 2024) conducts SFT in a block-coordinate-descent (BCD) fashion. This method requires repeated Adam steps to solve the sub-problem in BCD. Similarly as in LoRA, the Adam steps in BAdam can be replaced by Adam-mini to further reduce memory. Nero optimizer (Liu et al., 2021a) also cuts down the memory of Adam. It removes the 1st-order momentum and uses a neuron-specific projected gradient-style update. According to (Liu et al., 2021a), their design imposes constraints on weight matrices and has the advantage of "balanced excitation and inhibition". Such design can potentially be combined with Adam-mini to further boost performance. To save the memory cost for fine-tuning LLMs, MeZO (Malladi et al., 2023) uses zeroth-order methods to approximate the gradient information. It is possible to combine this idea with Adam-mini to further save memory for SFT. Adam-mini can also potentially be combined with other diagonal preconditioned methods (such as AdaGrad (Duchi et al., 2011) and Adan (Xie et al., 2022)) as well as recent schedule-free optimizers such as SchedulefreeAdamW (Defazio et al., 2024).

There are several other tricks that ease GPU memory burden but are orthogonal to optimizer design. These tricks include gradient checkpointing (Chen et al., 2016), model offloading and sharding (Rajbhandari et al., 2020; 2021), quantization (Dettmers et al., 2021; Li et al., 2024), and fused update (Lv et al., 2023a;b). Adam-mini can be implemented upon these tricks.

Finally, we discuss another popular adaptive optimizer called LAMB (You et al., 2019) (see **Algorithm 7** in Appendix E.1). LAMB might be misunderstood as a similar optimizer to Adam-mini, but actually, it is not. We emphasize that Adam-mini is *significantly different* from LAMB. First, LAMB still keeps the same coordinate-wise learning-rate design $1/\sqrt{v}$ as in Adam. Second, in addition to this $1/\sqrt{v}$, LAMB re-scales the parameters in a layer-by-layer fashion. This re-scaling design is often known as the "layer-wise learning rates", but to be precise, it is actually an additional "layer-wise scaling" besides the "coordinate-wise learning rates $1/\sqrt{v}$". As a result, LAMB does not save memory over Adam and its overall design is quite different from Adam-mini. This is understandable because LAMB is designed for large-batch training, not for memory saving. Numerically, we find that LAMB performs worse than Adam-mini on GPT2 pre-training (Figure 10).

---

[5]These results are calculated based on (Zhao et al., 2024a, Table 1). We consider Llama 2-7B and $r = 1024$ in GaLore.

## B    THE COMPLETE FORM OF ADAM-MINI

We now present the specific realization of Adam-mini on Transformers and other architectures. To be precise, **Algorithm 3** should be renamed as "Partition for CNNs, Diffusion models, and Graph Neural Networks", since we have only tested **Algorithm 3** on these models. In the future, it is possible that we will have more complicated non-Transformer architectures on which **Algorithm 3** fails. In those cases, we need to investigate the Hessian structure of these new architectures (like what we did for Transformers) and then develop the concrete partition algorithms following our **Principle 1** in Section 2.2.

---

**Algorithm 2** Adam-mini in Pytorch style

1: Input weight-decay coefficient $\lambda$ and current step $t$
2: Choose `param_blocks` from **Algorithm 3 or 3**
3: **for** `param` in `param_blocks` **do**
4:    `g = param.grad`
5:    `param = param` - $\eta_t * \lambda * $ `param`
6:    `m` = $(1 - \beta_1) * $ `g` $+ \beta_1 * $ `m`
7:    $\hat{\mathrm{m}} = \frac{\mathrm{m}}{1 - \beta_1^t}$
8:    `v` = $(1 - \beta_2) * $ `mean(g⊙g)` $+ \beta_2 * $ `v`
9:    $\hat{\mathrm{v}} = \frac{\mathrm{v}}{1 - \beta_2^t}$
10:    `param = param` - $\eta_t * \frac{\hat{\mathrm{m}}}{\sqrt{\hat{\mathrm{v}}} + \epsilon}$
11: **end for**

---

**Algorithm 3** Partition for non-Transformers

1: `param_blocks = {}`
2: **for** `name, param` in parameters **do**
3:    `param_blocks[name]=param`
4: **end for**
5: **return** `param_blocks`

---

**Algorithm 3** Partition for Transformers

1: `param_blocks = {}`
2: **for** `name, param` in parameters **do**
3:    **if** `'embed'` or `'output'` in name **then**
4:       Partition `param` by tokens
5:       **for** `i = 0...tokens-1` **do**
6:          `param_blocks[name+i]=param[i]`
7:       **end for**
8:    **else if** `'query'` or `'key'` in name **then**
9:       Partition `param` by heads
10:      **for** `i = 0...heads-1` **do**
11:         `param_blocks[name+i]=param[i]`
12:      **end for**
13:   **else if** `'value'`, `'attn.proj'`, or `'mlp'` in name **then**
14:      Partition `param` by output neurons
15:      **for** `i = 0...output_neurons-1` **do**
16:         `param_blocks[name+i]=param[i]`
17:      **end for**
18:   **else**
19:      `param_blocks[name]=param`
20:   **end if**
21: **end for**
22: **return** `param_blocks`

---

## C    MORE DISCUSSIONS

**Analysis in (Collobert, 2004).** We now briefly restate the analysis in (Collobert, 2004) on "why does the Hessian of neural networks exihibts near-block-diagonal structure?". Consider a standard supervised learning problem: minimizing $\ell(f(\theta, x), y)$ where $\ell(\cdot, \cdot)$ is the Cross-Entropy (CE) loss, $f(\theta, x) = \sum_{i=1}^{n} v_i \phi(w_i^\top x)$ is an 1-hidden-layer neural net with input $x \in \mathbb{R}^d$, weight $w_i \in \mathbb{R}^d$, $v_i \in \mathbb{R}$, and label $y \in \{0, 1\}$, then the off-diagonal-block Hessian elements would contain

$$\frac{\partial^2 \ell(f(\theta, x), y)}{\partial w_i \partial w_j} = p(x)(1 - p(x))v_i v_j \phi'\left(w_i^\top x\right) \phi'\left(w_j^\top x\right) x x^\top \quad \text{for } i \neq j, \tag{3}$$

where $p(x) = 1/(1 + \exp(-y f(\theta, x)))$ and $\phi'(\cdot)$ is the derivative of $\phi(\cdot)$. Since the training objective is to maximize $p(x)$, the term $p(x)(1 - p(x))$ will quickly shrink to zero. This term will push the Hessian to near-block-diagonal structure where each block corresponds to one output neuron. The authors report that this can happen just after 1 training step, as restated in Figure 3 (a). We also numerically reproduce this result on a small MLP on CIFAR-100 in Figure 3 (b,c,d), and show that the near-block-diagonal Hessian structure maintains along training.

Finally, we note that this analysis is rather informal. We leave a more rigorous theoretical study is left for future work.

**Why using average $v$ as learning rates.** In Line 9 of **Algorithm 1**, we use the average of $v$ in a block as the learning rate for that block. We choose such a design due to the following reasons.

- **First: grid-search is too expensive.** Optimal blockwise learning rates can be powerful (as evident in Figure 6), but they are too expensive to search. Such a searching procedure is not scalable.

- **Second: average of $v$ can be borrowed from Adam.** Compared to searching all the learning rates from scratch, it is much easier to "borrow" them from the current design of Adam. The average of $v$ is the most natural quantity to "borrow" and it performs the best among other candidates such as the maximum of $v$ (see the ablation studies in Appendix D.2). We find that the average of $v$ helps Adam-mini to be as effective as Adam (though not significantly surpassing it).

- **Third: average of $v$ keeps us close to Adam.** For neural nets, we find that the average of $v$ is a good representative for the whole $v$ in the block, and can help Adam-mini keep close to Adam. The reason comes from backpropagation (BP) rule: for one data sample, the gradient of the weight matrix $W \in \mathbb{R}^{d \times d}$ can be expressed as $G := \frac{\partial \ell}{\partial W} = ez^\top \in \mathbb{R}^{d \times d}$, where $e$ is certain BP error vector and $z$ is the input feature to the current weight. For all entries in the $i$-th row of $G$, they all share the same BP error term $e_i$, which is usually non-negligible when $G \neq 0$. Therefore, $G$ usually has similar entries within a row (which associates with the same output neuron), and its mean value can be a good representative of the whole row. As a result, we find that Adam-mini's trajectory closely resembles that of Adam (see the curves in Figure 8 and Figure 10). **One resulting advantage is that Adam-mini can maintain the scaling laws of LLMs trained by Adam, while substantially saving the training cost (see evidence in Figure 11).**

## D MORE EXPERIMENTAL RESULTS

### D.1 MORE RESULTS FOR MOTIVATION

In Section 2, we showed that "Adam's $v$ is redundant on dense Hessian subblock". We provide experiments on random quadratic functions to support the claim. Here, we conduct the following new experiments on a 1-layer Transformer. We will show that "using single learning rate per block" is also sufficient for Transformers.

The following **Exp 1 and 2** extend the random quadratic experiments in **Figure 4 and 5** to Transformers.

**Exp 1: Adam's learning rate is redundant on the dense Hessian subblock.** We take some small dense blocks in the Hessian of 1-layer Transformer and denoted as $H$. We compare $\kappa(H)$ and $\kappa(D_{Adam}H)$ as in the paper. We find Adam is not effective in reducing the kappa of these blocks, and many lrs in Adam can be redundant.

Table 3: Comparison of $\kappa(H)$ and $\kappa(D_{Adam}H)$ for the dense blocks in the Hessian of 1-layer Transformer.

| Hessian Block | $\kappa(H)$ | $\kappa(D_{Adam}H)$ |
|---|---|---|
| 1st head in Query | 103.80 | 176.88 |
| 1st head in Key | 103.46 | 213.82 |
| 1st head in Value | 165.66 | 332.76 |
| 1st neuron in attn.proj | 39.92 | 94.56 |
| 1st neuron in MLP_fc1 | 22.04 | 70.92 |
| 1st neuron in MLP_c_proj | 63.85 | 236.71 |

**Exp 2: Single learning rate per block is sufficient.** We conduct the "block-wise GD" and we grid-search the learning rate for each block. The result is shown in the following figure. We find that block-wise GD outperforms AdamW. This extends the setting from the random quadratic problem in Figure 4.

Combining **Exp 1 and 2**, we can see that Adam is redundant on the dense Hessian subblocks (**Exp 1**), and a single lr for each block can work well (**Exp 2**). These experiments show that our conclusions on random quadratic problems can be extended to Transformers.

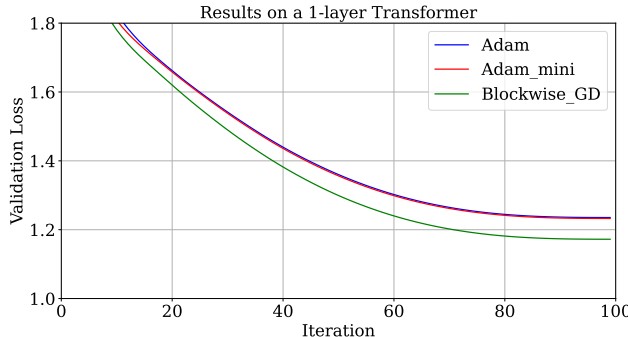

Figure 14: On a 1-layer Transformer, we conduct the "blockwise GD" and we grid-search the learning rate for each block. We find that blockwise GD outperforms AdamW.

### D.2 Ablation Studies on the Design of Adam-mini

We here provide more reasons why we choose $\text{mean}(v)$ as the blockwise learning rates. We conduct ablation studies on different choices of quantities that we can borrow from Adam, including $\text{2-norm}(v)$, $\text{1-norm}(v)$, $\text{max}(v)$, and $\text{min}(v)$. we found that all these candidates perform worse than $\text{mean}(v)$ in Adam-mini.

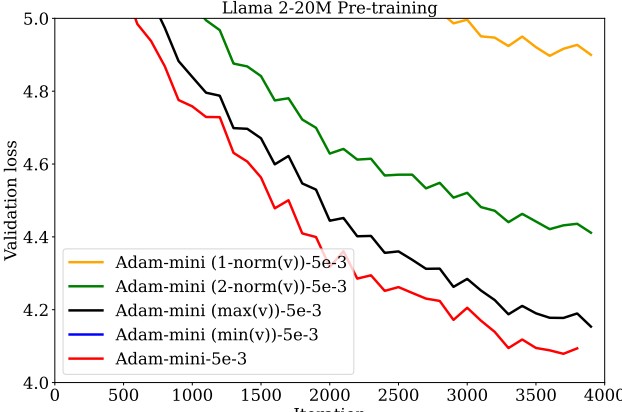

Figure 15: Ablation studies on the design of Adam-mini. We find that $\text{mean}(v)$ performs better than other candidates. The blue curve does not show because the algorithm diverges and the curve is out of range.

### D.3 More Results on the Scaling Law Experiments

**The complete loss curves of Llama 2-1B.** We here present the complete validation loss curve of Llama 2-1B, training on 20B tokens, which corresponds to the rightmost curve in the scaling law experiments in Figure 11 (a). We note that this is a complete pre-training run under the definition of Chinchila's law (Hoffmann et al., 2022). We find that Adam-mini's loss curves closely resemble those of AdamW.

**The final validation perplexity.** In Table 4, we present the final validation perplexity for all models after training on the token amount suggested by Chinchilla's law (Hoffmann et al., 2022). For all models from 39M to 1B, we find that Adam-mini reaches a lower validation perplexity than AdamW.

### D.4 GPT-4 Evaluation Score of SFT and RLHF

On the pre-trained Llama 2-7B (released by Meta (Touvron et al., 2023)), we run Adam-mini on downstream tasks including SFT and RLHF. We evaluate the alignment performance in terms of chat ability using the MT-Bench (Zheng et al., 2024), where GPT-4 assesses multi-turn chatting capabilities and assigns a score from 0 to 10 (higher is better). Our results, presented in Table 5, demonstrate that Adam-mini can outperform AdamW.

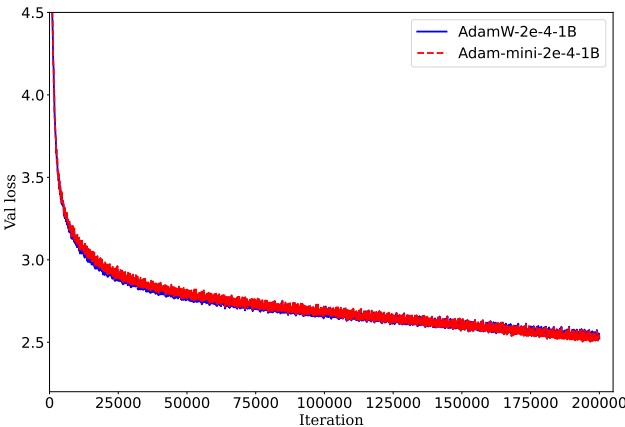

Figure 16: Loss curves of pre-training Llama 2-1B on 26B tokens. This is a complete pre-training run under the definition Chinchila's law. We find that Adam-mini performs similarly to AdamW throughout the training, but with 50% less memory.

Table 4: Final validation perplexity for all models pre-trained by Adam and Adam-mini. The token amount follows Chinchila's law. After pre-training, Adam-mini reaches a lower validation perplexity than AdamW.

| Model size | 39M | 67M | 102M | 162M | 271M | 1B |
|---|---|---|---|---|---|---|
| Total tokens | 1.02B | 1.76B | 2.67B | 4.25B | 7.10B | 26.21B |
| AdamW | 40.795 | 29.319 | 24.670 | 20.360 | 17.178 | 12.452 |
| Adam-mini | **40.407** | **29.014** | **24.192** | **20.172** | **17.035** | **12.372** |

Table 5: Averaged GPT-4 evaluation score (↑) of SFT and RLHF on the MT-Bench.

| | SFT (LoRA) | | SFT | | RLHF | |
|---|---|---|---|---|---|---|
| | AdamW | Adam-mini | AdamW | Adam-mini | AdamW | Adam-mini |
| MT-Bench | 4.23 | **4.41** | 5.37 | **5.40** | 5.54 | **5.68** |

## D.5 Non-LLM Tasks

We now evaluate Adam-mini on non-LLM tasks. Table 6 shows the results for the training of ResNet18(He et al., 2016), Swin-Transformer (Liu et al., 2021b), DiT-XL-2 (Peebles & Xie, 2023), DC-AE-Diffusion (Chen et al., 2024a) on ImageNet, DDPM diffusion model (Ho et al., 2020) on CelebA, a Graph Convolution Net (GCN) (Kipf & Welling, 2016), and a Graph Attention Net (GAT) (Veličković et al., 2017) on OGB-arxiv. The training curves are shown in Figure 17 and 18. We find the performance of Adam-mini to be comparable or better than AdamW, but with less memory.

Table 6: On popular non-LLM tasks, Adam-mini performs on par or better than AdamW.

| Domain | Model | Optimizer | Metric | 25% steps | 50% steps | 75% steps | 100% steps |
|---|---|---|---|---|---|---|---|
| Vision | DDPM | AdamW | Train loss (↓) | 0.0529 | 0.0497 | 0.0420 | 0.0394 |
| Vision | DDPM | Adam-mini | Train loss (↓) | **0.0525** | **0.0495** | **0.0416** | **0.0388** |
| Vision | ResNet18 | AdamW | Val acc (↑) | 0.6149 | 0.6478 | 0.6613 | 0.6669 |
| Vision | ResNet18 | Adam-mini | Val acc (↑) | 0.6140 | **0.6501** | **0.6629** | 0.6667 |
| Vision | Swin-Transformer | AdamW | Val acc (↑) | **0.6290** | 0.6940 | **0.7180** | **0.7310** |
| Vision | Swin-Transformer | Adam-mini | Val acc (↑) | 0.6230 | **0.6960** | 0.7160 | 0.7300 |
| Vision | DiT-XL-2 | AdamW | Train loss (↓) | 0.1605 | 0.1696 | 0.1607 | 0.1431 |
| Vision | DiT-XL-2 | Adam-mini | Train loss (↓) | **0.1601** | **0.1693** | 0.1605 | **0.1430** |
| Vision | DC-AE-Diffusion | AdamW | Train loss (↓) | 0.2860 | **0.2820** | 0.2800 | 0.2780 |
| Vision | DC-AE-Diffusion | Adam-mini | Train loss (↓) | 0.2860 | 0.2830 | **0.2800** | **0.2780** |
| Graph | GAT | AdamW | Val acc(↑) | 0.7277 | 0.7367 | 0.7399 | 0.7421 |
| Graph | GAT | Adam-mini | Val acc (↑) | **0.7378** | **0.7394** | **0.7403** | **0.7429** |
| Graph | GCN | AdamW | Val acc (↑) | 0.7347 | 0.7428 | 0.7379 | 0.7374 |
| Graph | GCN | Adam-mini | Val acc (↑) | **0.7406** | 0.7427 | **0.7380** | **0.7423** |

In Table 7, we further evaluate the image quality from the model trained by Adam-mini. We find that the Adam-mini performs on par with AdamW.

Table 7: Evaluation scores: Adam-mini performs on par with AdamW.

| Domain | Model | Optimizer | FID ($\downarrow$) | Inception Score ($\uparrow$) |
|--------|-------|-----------|---------|-----------------|
| Vision | DiT-XL-2 | AdamW | 91.83 | 12.38 |
| Vision | DiT-XL-2 | Adam-mini | **88.20** | **13.90** |
| Vision | DC-AE-Diffusion | AdamW | 34.72 | 41.79 |
| Vision | DC-AE-Diffusion | Adam-mini | **33.15** | **44.38** |

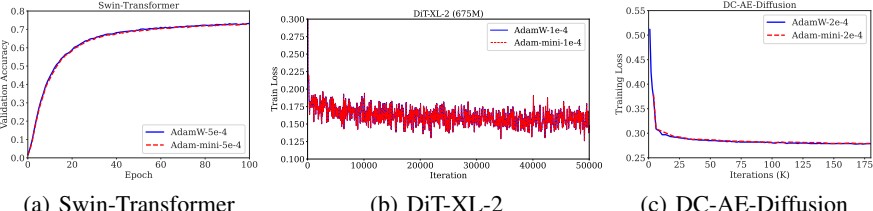

(a) Swin-Transformer  (b) DiT-XL-2  (c) DC-AE-Diffusion

Figure 17: The training curves of Swin-Transformer, DiT-XL-2, and DC-AE-Diffusion. We find that Adam-mini performs on par with AdamW

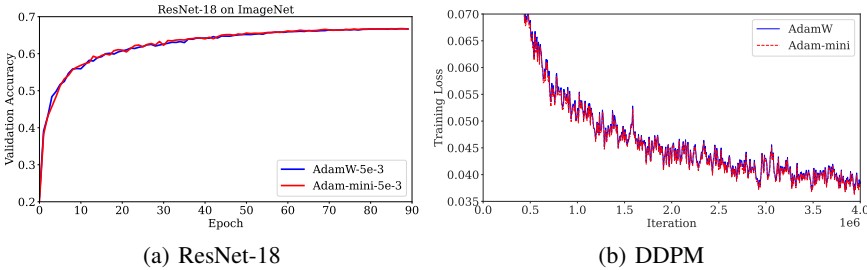

(a) ResNet-18  (b) DDPM

Figure 18: The training curves of ResNet-18 and DDPM diffusion model. We find that Adam-mini performs on par with AdamW.

### D.6  MORE DISCUSSIONS ON THE PARTITION STRATEGIES OF VALUE

As shown in Figure 7, the Hessian structure of value is less clear compared to other blocks: it shows the hint of 16 diagonal blocks (where 16 is the number of output neurons), but the pattern is not that clear. This gives rise to two potential partition strategies: (I) partition by output neuron; (II) treat as a whole. Numerically, we find that strategy (I) works well when the number of total training steps is large. This includes most of our experiments such as GPT-2 in Figure 8 (with more than 50k total steps) and the scaling law experiments of Llama models in Figure 11 (e.g., Llama 2-1B is trained with more than 200k total steps). On the other hand, we find that strategy (II) works better when the number of total training steps is small. This includes our Llama experiments with 10k total steps in Figure 10.

Based on these findings, we recommend using strategy (I) when the total number of training steps is large, and using strategy (II) if otherwise. Note that strategy (II) can be used simply by adding one line of code after creating the optimizer: `optimizer.wv_names = {}`.

### D.7  DETAILED COMPARISON WITH ADAFACTOR

In this section, we conduct a more hyperparameter search for Adafactor on Llama 2-20M pre-training. We will focus on tuning Adafactor-Zhai-version since it performs better than the original Adafactor (see Figure 13). We consider the following three setups.

- **Setup 1:** We change the default $\beta_2 = 0.999$ to $\beta_2 = 0.95$ and sweep over learning rates.
- **Setup 2:** We use learning rate = 5e-3, $\beta_2 = 0.95$ and sweep over warm-up step = $\{1\%, 2\%, 3\%, 4\%, 5\%, 10\%\}$ total steps.
- **Setup 3:** We use learning rate = 5e-3, $\beta_2 = 0.95$ and warm-up step = $1\%$ total steps and sweep over $\epsilon = \{10^{-30}, 10^{-16}, 10^{-8}, 10^{-6}\}$.

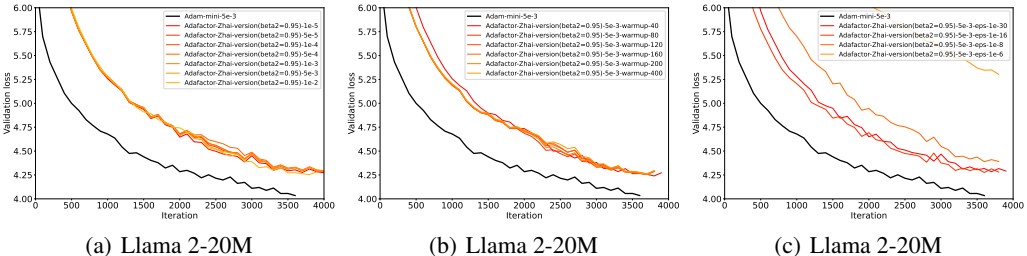

| (a) Llama 2-20M | (b) Llama 2-20M | (c) Llama 2-20M |

Figure 19: The training curves of Adafactor-Zhai-version on Llama 2-20M pre-training. (a,b,c) corresponds to the aforementioned **Setup 1, 2, 3**, respectively. We find that Adafactor consistently underperforms Adam-mini.

The results are shown in Figure 19. In all these cases, Adafactor-Zhai-version consistently underperforms Adam-mini and the change of hyperparameters does not help much.

## D.8 Detailed Comparison with Lion

We now conduct the hyperparameter grid search over Lion. We find that Lion is not easy to tune and we have *not* managed to make Lion work. We consider the following settings.

**Tuning strategies in (authors, 2024).** authors (2024) carefully tune Lion on Llama models (150M, 300M, 600M, 1.2B). We will adopt their optimal tuning strategies in (authors, 2024, Table 1) [6]. We here summarize their key messages.

- **Message 1:** The optimal learning rate (lr) of Lion is usually 10 times smaller than AdamW.
- **Message 2:** The magical number lr = 3.16e-4 works the best for most models (Llama 150M, 300M, 600M).
- **Message 3:** $\beta_1 = \{0.95, 0.9\}$ perform similarly and perform significantly better than other $\beta_1$ candidates including $\beta_1 = \{0.99, 0.98, 0.8, 0.5, 0\}$.
- **Message 4:** $\beta_2 = \{0.99, 0.98, 0.95\}$ perform similarly and perform significantly better than other $\beta_2$ candidates including $\beta_2 = \{0.9999, 0.999, 0.995, 0.9, 0.8\}$.

In the following, we will use the above messages to tune the hyperparameters of Lion.

**Architecture.** We consider Llama 2-20M, which is the same architecture as the ones investigated in (authors, 2024), but with different model size. We also consider GPT-2-125M, which is a task that Lion is not tested before (neither in (authors, 2024) nor in other literatures to our knowledge).

**Our tuning strategies.** Following the above **Message 1 and 2** from (authors, 2024), we will use the following tuning strategies for Lion on Llama 2-20M and GPT-2-125M.

- **Learning rate for Llama 2-20M:** The standard lr is 5e-3, so we try lr = [5e-4, 6e-4, 7e-4, 8e-4, 9e-4, 1e-3, 2e-3, 3e-3, 4e-3, 5e-3]. For completeness, we also investigate lr = [4e-4, 3.16e-4, 2e-4, 1e-4].
- **Learning rate for GPT-2-125M:** The standard lr is 6e-4, so we try lr = [6e-5, 7e-5, 8e-5, 9e-5, 1e-4, 2e-4, 0.000316, 4e-4, 5e-4, 6e-4]

As for $(\beta_1, \beta_2)$, we will use $(\beta_1, \beta_2) = (0.95, 0.98)$. We use these hyperparameters for two reasons. First, they are the optimal choice among other candidates by **Message 3 and 4**. Second, $(\beta_1, \beta_2) =$

---

[6]We would like to mention that (authors, 2024) is a concurrent work to us, and their tuning strategies in is not public available by the time we submitted this script.

$(0.95, 0.98)$ is recommended by the authors of Lion to be "helpful in mitigating instability during training" [7].

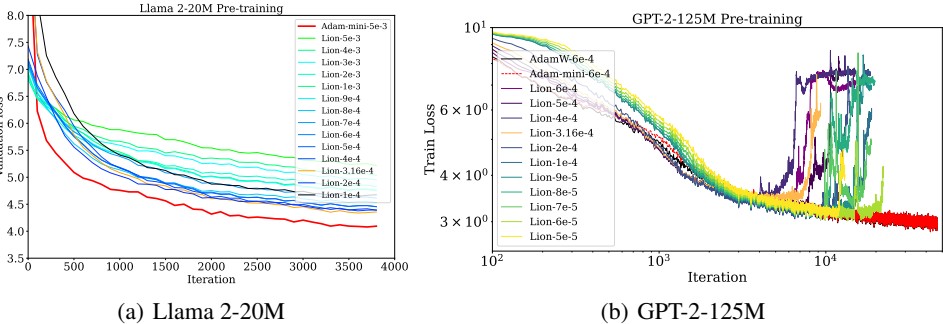

(a) Llama 2-20M            (b) GPT-2-125M

Figure 20: The training curves of Lion on Llama 2-20M and GPT-2-125M pre-training. The hyperparameters are chosen under the optimal strategies by (authors, 2024). We find that Lion consistently underperforms Adam-mini on Llama 2-20M, and it encounters loss spikes on GPT-2-125M.

The results are shown in Figure 20. After using the above tuning strategies, we find that **Lion still underperforms Adam-mini and AdamW** on Llama 2-20M and GPT-2-125M. In particular, Lion **encounters loss spikes** on GPT-2-125M for all the learning rate candidates above.

We summarize our findings on Lion below.

- **First: worse performance.** With all the effort above, we haven't managed to make Lion work, and we haven't been able to reproduce (authors, 2024) on Llama 2-20M and GPT-2-125M (different model size and architectures from (authors, 2024)). One possible reason is that Lion might work under their specific setup (dataset, architecture, batch size, etc.), but the effectiveness is not easily transferable.

- **Second: no general tuning guidance.** We find that there are no general tuning strategies for Lion. We emphasize that authors (2024) only focuses on Llama architectures, and their resulting tuning strategy seems not robust and transferable to other architectures. In particular, their optimal strategy on Llama causes loss spikes on GPT-2. To our knowledge, Table 1 in (authors, 2024) is the only public tuning strategy for Lion, so it seems unclear how to tune Lion in general.
  **In contrast, Adam-mini is much easier to use.** In all our experiments (a wide range of tasks and models), Adam-mini performs on par with Adam **using the same hyperparameters as AdamW** (including learning rate, $\beta_1$, $\beta_2$, $\epsilon$, etc.). We believe that "easy adaptation of the hyperparameters" can serve as one advantage of Adam-mini over Lion, apart from the performance superiority.

- **Third: Adam-mini is more principled and explainable.** We emphasize that Lion is designed by symbolic search, and its design principle is largely unclear. In contrast, the Design principle of Adam-mini is much more understandable: we remove the redundant lrs in Adam according to the Hessian structure. We believe Adam-mini is more "white-box" than Lion and more explainable to users.

### D.9 ADDITIONAL FINDINGS ON GPT-2-330M

In Figure 21, we report some unexpected results on GPT-2-330M. When using the recommended learning rate 3e-4 by (Liu et al., 2023), we find that AdamW encounters loss spikes, while Adam-mini does not. The loss spike of AdamW can be mitigated by changing $\epsilon$ from 1e-8 to 1e-6. We have not fully understood the cause of the loss spike and how Adam-mini prevents it in this experiment. We are not sure whether such benefit of Adam-mini maintains on larger models. We leave more investigation as an interesting future direction.

### D.10 COMBINING WITH LORA

We note that Adam-mini can be combined with LoRA (Hu et al., 2021): we can change the Adam steps in LoRA by Adam-mini. The results are shown in Figure 22. We find that such changes with

---

[7]https://github.com/lucidrains/lion-pytorch

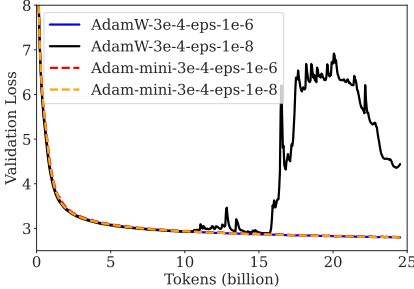

Figure 21: On GPT-2-330M, AdamW encounters loss spikes, while Adam-mini does not. The loss spike of AdamW can be mitigated by changing $\epsilon$ from 1e-8 to 1e-6.

boost performance. We grid-search learning rates for both methods and report the best performance. The detailed learning rate configuration is shown in Section F.

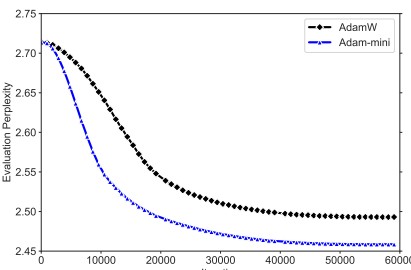

Figure 22: Training curves of SFT (LoRA). We find that LoRA performs better when replacing the Adam steps by Adam-mini.

### D.11 SAMPLE RESPONSES FROM LLMS TRAINED BY ADAM-MINI

The sample response in shown in Figure 23.

---

**Prompt: Compose an engaging travel blog post about a recent trip to Hawaii, highlighting cultural experiences and must-see attractions.**

---

`AdamW:` Sure! Here's a travel blog post about a recent trip to Hawaii: ...... Whether you're interested in history, nature, or simply soaking up the sun, Hawaii has something for everyone. Until next time, Aloha!

`Adam-mini:` Sure! Here's a travel blog post about a recent trip to Hawaii: ...... I highly recommend visiting this beautiful archipelago for a truly unforgettable adventure.

---

**Prompt: Rewrite your previous response. Start every sentence with the letter A.**

---

`AdamW:` A trip to Hawaii is a must-visit destination for any traveler, offering a variety of cultural experiences and must-see attractions. From the Polynesian Cultural Center to the USS Arizona Memorial, and from Lanikai Beach to the lively night markets, every corner of the islands reveals a new surprise and a new way to connect with the spirit of Hawaii.

`Adam-mini:` A trip to Hawaii is a must-do for any travel enthusiast. A trip to the beautiful Hawaiian Islands is a paradise on earth. A visit to Oahu is a great way to start your adventure. A helicopter tour of the island is both educational and breathtaking. A visit to Maui is a must-see. A visit to Haleakala National Park is a highlight of your trip. A visit to the Polynesian Cultural Center is a great way to learn about the history and traditions of the islands. A traditional luau is a must-attend event. Overall, a trip to Hawaii is an unforgettable experience.

---

Figure 23: Response Sample of Llama 2-7B finedtuned by AdamW and Adam-mini to the #81 test question from the MT-Bench (Zheng et al., 2024) dataset. Since there is little information in the first round response, we omit the answers from the models.

# E SOME PRELIMINARY RESULTS

## E.1 PRELIMINARIES ON ADAM, ADAMW AND LAMB

We introduce Adam, AdamW, and LAMB in **Algorithm 5**, **6**, and **7**. These methods need to track $m$ and $v$ along the training. Both $m$ and $v$ are vectors of the same size as # model parameter.

---
**Algorithm 5** Adam in Pytorch style

---
1: Let $\lambda$ be the weight decay coefficient
2: **for** `param` in `parameter_blocks` **do**
3:    `g` = `param.grad`
4:    **if** $\lambda > 0$ **then**
5:      `g` = `g` $+ \lambda *$ `param`
6:    **end if**
7:    `param` = `param` $- \eta_t * \lambda *$ `g`
8:    `m` = $(1 - \beta_1) *$ `g` $+ \beta_1 *$ `m`
9:    $\hat{m} = \frac{m}{1 - \beta_1^t}$
10:    `v` = $(1 - \beta_2) *$ `g` $\odot$ `g` $+ \beta_2 *$ `v`
11:    $\hat{v} = \frac{v}{1 - \beta_2^t}$
12:    `param` = `param` $- \eta_t * \frac{\hat{m}}{\sqrt{\hat{v}} + \epsilon}$
13: **end for**

---

---
**Algorithm 6** AdamW in Pytorch style

---
1: Let $\lambda$ be the weight decay coefficient
2: **for** `param` in `parameter_blocks` **do**
3:    `g` = `param.grad`
4:    `param` = `param` $- \eta_t * \lambda *$ `g`
5:    `m` = $(1 - \beta_1) *$ `g` $+ \beta_1 *$ `m`
6:    $\hat{m} = \frac{m}{1 - \beta_1^t}$
7:    `v` = $(1 - \beta_2) *$ `g` $\odot$ `g` $+ \beta_2 *$ `v`
8:    $\hat{v} = \frac{v}{1 - \beta_2^t}$
9:    `param` = `param` $- \eta_t * \frac{\hat{m}}{\sqrt{\hat{v}} + \epsilon}$
10: **end for**

---

---
**Algorithm 7** LAMB in Pytorch style

---
1: Let $\lambda$ be the weight decay coefficient, let $\phi$ be a scaling function.
2: **for** `param` in `all_layers` **do**
3:    `g` = `param.grad`
4:    `param` = `param` $- \eta_t * \lambda *$ `g`
5:    `m` = $(1 - \beta_1) *$ `g` $+ \beta_1 *$ `m`
6:    $\hat{m} = \frac{m}{1 - \beta_1^t}$
7:    `v` = $(1 - \beta_2) *$ `g` $\odot$ `g` $+ \beta_2 *$ `v`
8:    $\hat{v} = \frac{v}{1 - \beta_2^t}$
9:    `r` = $\frac{\hat{m}}{\sqrt{\hat{v}} + \epsilon}$
10:    `param` = `param` $- \eta_t * \frac{\phi(\|\text{param}\|)}{\|r + \lambda * \text{param}\|} *$ `r`
11: **end for**

---

## E.2 PRELIMINARY RESULTS IN (ZHANG ET AL., 2024)

We here restate (Zhang et al., 2024, Figure 3). This figure shows that: for Transformers, different parameter blocks have different Hessian eigenvalue distributions, while for CNNs, the eigenvalue

distributions are similar among blocks. This suggests that Transformers need different learning rates for different blocks to handle the heterogeneity in eigenvalue distributions.

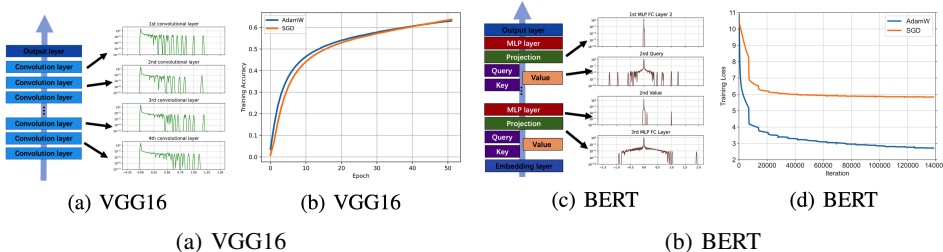

|(a) VGG16|(b) VGG16|(c) BERT|(d) BERT|

(a) VGG16                                    (b) BERT

Figure 24: Figure 3 in (Zhang et al., 2024). The eigenvalues distribution are similar among blocks for CNNs, while they differ significantly across blocks for Transformers. This indicates Transformers need different learning rates for different blocks to handle the heterogeneity in eigenvalues.

## F  EXPERIMENTAL DETAILS

### F.1  TRAINING CONFIGURATIONS FOR SECTION 3

Unless mentioned otherwise, we choose the model configurations by their standard protocols. We choose the learning rates by the recommendation from open-source platforms if applicable. For instance, for GPT2 series, we use the recommended learning rates by (Liu et al., 2023), which are reported to be optimal by grid search. Unless mentioned otherwise, Adam-mini, Adafactor, CAME, SM3, and LAMB use the same learning rate as the recommended ones of AdamW. If there is no public recommended learning rate for AdamW, we tune the learning rate for all optimizers within the same computational budget and report the best performance. For other hyperparameters, we follow the recommendation from open-source platforms or by their default setting. For SM3 and Adafactor, we incorporate momentum with $\beta_1 = 0.9$ to offer a fair comparison with other optimizers and the rest of the hyperparameters are set as default. The detailed configurations are explained as follows.

**GPT2 pre-training.**   We use the nanoGPT codebase[8] to train GPT2 sized 125M (small), 330M (medium), and 1.5B (XL) on Openwebtext. For all models, we use `seq_len` = 1024, batch size = 480, weight decay coefficient $\lambda = 0.1$, $\epsilon$ = 1e-8, $\beta_1 = 0.9, \beta_2 = 0.95$. We use cosine-decay learning rate schedule with 2000 iterations of warm-up. For GPT2-small and medium, we use the recommended peak learning rate by (Liu et al., 2023), which are reported to be the optimal ones found by grid search. For GPT2-XL, we use the recommended peak learning rate by the Levanter[9]. The chosen peak learning rates are 6e-4, 3e-4, 1e-4 for GPT2-small, medium, XL, respectively. The minimal learning rate is chosen as 3e-5, 6e-5, 1e-5 for these models.

**Llama pre-training.**   For all experiments on the Llama series (from 20M to 13B), we use the Torchtitan codebase[10] and C4 dataset (Raffel et al., 2020). For all experiments, we use weight decay coefficient $\lambda = 0.1$, $\epsilon$ = 1e-8, $\beta_1 = 0.9, \beta_2 = 0.95$. For Llama 2-1B, Llama 3-8B, we use learning rate = 3e-4. For Llama 2-13B, we use learning rate = 1e-4. As for the learning rate schedule, we use warm-up step = 1% total step and use linear decay schedule after the warm-up (this is the default setting in the Torchtitan codebase). For Figure 10 (a) and all the experiments of Adafactor and Lion, we use `seq_len` = 512 and batch size = 128. For Figure 10 (b), we use `seq_len` = 2048 and batch size = 8. For Figure 10 (b), we shrink the batch size due to the limited hardware. For all the scaling law experiments, we use `seq_len` = 512 and batch size = 256. We summarize the detailed setups for the scaling law experiments in later paragraphs.

---

[8]https://github.com/karpathy/nanoGPT/tree/master
[9]https://github.com/stanford-crfm/levanter/blob/e183ec80ec5971b12d4a3fb08a160268de342670/config/gpt2_xl.yaml
[10]https://github.com/pytorch/torchtitan

**SFT and RLHF.** We use the Llama 2-7B pretrained model (Touvron et al., 2023) for our study. We use the `ultrafeedback` dataset [11]. The implementation of SFT and RLHF code is based on the ReMax codebase[12]. Specifically, we train a SFT model with 40% of the chosen data and train a reward model using the remaining 60%. Then, we apply the reinforcement learning algorithm ReMax (Li et al., 2023), a memory-efficient alternative to PPO (Schulman et al., 2017), to optimize the preference reward.

We use DeepSpeed ZeRO-2 in our training. GPT-4 evaluation template in Table 5 is from the codebase[13]. In the reward optimization stage, We use ReMax, a memory-efficient alternative to PPO. We use UltraFeedback dataset Cui et al. (2023) and use 40% data for SFT and 60% data for ReMax.

**SFT.** We use 80 samples in a batch and train the model for 3 epochs. For the full parameter tuning, we search the learning rate from {1e-6, 2e-6, 3e-6, 4e-6, 5e-6, 1e-5, 2e-5} based on validation loss, and we use 2e-6 with cosine annealing for both AdamW and Adam-mini. For LoRA, We apply LoRA for all layers except the embedding layer. The rank of LoRA is set to 128. After selecting the learning rate from the same set as the full parameter tuning, we use 2e-5 for both AdamW and Adam-mini when LoRA is applied. The weight decay coefficient is set to 0 as recommended by LlamaFactory[14]. The rest of the hyperparameters of AdamW and Adam-mini are $\epsilon = 1e\text{-}8$, $\beta_1 = 0.9$, $\beta_2 = 0.95$.

**ReMax.** We use 48 samples in a batch and train the model for 1 epoch. By searching the peak learning rate from {5e-7, 1e-6, 2e-6} based on validation reward, AdamW uses 1e-6 while Adam-mini selects 5e-7 as the peak learning rate. The weight decay coefficient is set to 0. The rest of the hyperparameters of AdamW and Adam-mini are $\epsilon = 1e\text{-}8$, $\beta_1 = 0.9$, $\beta_2 = 0.95$.

**ResNet.** We use the PyTorch official implementation codebase[15] to train ResNet18 (He et al., 2016) on ImageNet (Deng et al., 2009). We use cosine-decay learning rate, epoch =90, $\beta_1 = 0.9, \beta_2 = 0.999, \epsilon = $1e-8. For ResNet18, we use batch size = 256, peak learning rate = 0.005. For ViT-base, we use batch size = 128, peak learning rate = 0.0001. These configurations are used for both Adam-mini and AdamW.

**DDPM.** We use the codebase[16] to train DDPM diffusion models (Ho et al., 2020). The image size is 64 and the training objective is to predict the noise as in (Ho et al., 2020). We use the default U-Net archiecture hyper-parameters and the dimension multiply in U-Net is (1, 2, 4, 8). We use the CelebA dataset[17] and train the diffusion model with a learning rate $5 \times 10^{-5}$ with cosine decay. The batch size is 128 and the training epoch is 50.

**Swin-Transformers.** We use the official implementation[18] of Swin-Transformers (Liu et al., 2021b) on ImageNet. All configurations as default. For both Adam-mini and AdamW, we use the default learning rate = 5e-4.

**DiT-XL-2.** We use the official implementation[19] of DiT-XL-2 (Peebles & Xie, 2023) on ImageNet. All configurations as default. For both Adam-mini and AdamW, we use the default learning rate = 1e-4.

**DC-AE-Diffusion.** We use the official implementation[20] of DC-AE-Diffusion (Chen et al., 2024a) on ImageNet. All configurations as default. For both Adam-mini and AdamW, we use the default learning rate = 2e-4.

---

[11] https://huggingface.co/datasets/argilla/ultrafeedback-binarized-preferences-cleaned
[12] https://github.com/liziniu/ReMax
[13] https://github.com/lm-sys/FastChat/tree/main/fastchat/llm_judge
[14] https://github.com/hiyouga/LLaMA-Factory
[15] https://github.com/pytorch/examples/blob/main/imagenet/main.py
[16] https://github.com/lucidrains/denoising-diffusion-pytorch
[17] https://cseweb.ucsd.edu/~weijian/static/datasets/celeba/
[18] https://github.com/microsoft/Swin-Transformer
[19] hhttps://github.com/facebookresearch/DiT
[20] https://github.com/mit-han-lab/efficientvit/blob/master/applications/dc_ae/README.md

**Graph Neural Networks.** We use the DGL implementation[21] of Graph Convolution Networks (GCN) (Kipf & Welling, 2016) and Graph Attention Networks (GAT) (Velickovic et al., 2017) for OGBN-arxiv[22] dataset. All configurations as default. For both Adam-mini and AdamW, we use the default learning rate = 0.005 for GCN and the default learning rate = 0.002 for GAT.

**Scaling law experiments.** We use the codebase Torchtitan[23] to train Llama models of different sizes. All the model configurations are shown in Table 8 and all the training configurations are shown in Table 9. The experimental setups are inspired by (Hägele et al., 2024). In all experiments, we fix the warm-up steps to be 1% of the total steps, as suggested by (Ibrahim et al., 2024).

| Model Size | $d_{model}$ | $n_{layers}$ | $n_{heads}$ | seq_len |
|---|---|---|---|---|
| 39M | 384 | 8 | 6 | 512 |
| 67M | 512 | 10 | 8 | 512 |
| 102M | 640 | 12 | 10 | 512 |
| 162M | 768 | 16 | 12 | 512 |
| 271M | 1024 | 16 | 16 | 512 |
| 1B | 2048 | 18 | 16 | 512 |

Table 8: The model configurations in the scaling law experiments.

| Model | LR | Batch size (# tokens) | Steps | Tokens | Token/Params Ratio |
|---|---|---|---|---|---|
| 39M | 6e-4 | 0.13M | 7.8K | 1.02B | 26.15 |
| 67M | 6e-4 | 0.13M | 13.4K | 1.76B | 26.27 |
| 102M | 6e-4 | 0.13M | 20.4K | 2.67B | 26.17 |
| 162M | 6e-4 | 0.13M | 32.4K | 4.25B | 26.23 |
| 271M | 6e-4 | 0.13M | 54.2K | 7.10B | 26.21 |
| 1B | 2e-4 | 0.13M | 200K | 26.21B | 26.21 |

Table 9: Training configurations for the scaling law experiments.

**Trajectory comparison in Figure 10 (c).** We train a 8-layer Transformer sized 11M on Openwebtext and launch AdamW, Adam-mini, and other memory-efficient optimizers under the same random seed and same learning rate 1e-5. We save the model weights for every 250 iterations and compare their Euclidean distance to the weights along AdamW's trajectory.

### F.2 DETAILED SETUP FOR OTHER EXPERIMENTS

**Configurations for Figure 3.** We train a 1-hidden-layer MLP with 8 neurons on CIFAR-100. We use Adam with 1e-4 learning rate with Cosine decay schedule. We train the network with 128 batch size and 20 epochs (i.e., about 7900 total training steps). With the help of auto-differentiation framework, we calculate the Hessian with two passes of backpropagation (Pearlmutter, 1994) and the calculation is exact.

**Configurations for Figure 4.** For each dense sub-block $H_l, l = 1, 2, 3$, we use random positive definite matrices. We fix the choose the eigenvalues of each $H_l$ as follows: for $l = 1$, we independently sample from $\{1,2,3\}$ for 30 times; for $l = 2$, we repeat this procedure for $\{99,100,101\}$; for $l = 3$, we repeat this procedure for $\{4998, 4999, 5000\}$. For the single (blockwise) learning rate method, we use GD with optimal constant learning rate $2/(L + \mu)$, where $L, \mu$ are the largest and smallest eigenvalue of the (blockwise) Hessian. We use Adam with $\beta_1 = 0$. This helps us focus on the effect of coordinatewise learning rate in Adam. We also set $\beta_2 = 1$ to the time-varying learning rate. This is necessary because, for any $\beta_2 < 1$, Adam with constant learning rate will oscillate on quadratic functions. This is theoretically proved in (Da Silva & Gazeau, 2020, Proposition 12, Figure 1) and empirically observed in (Zhang et al., 2024, Section 3.3).

---

[21] https://github.com/dmlc/dgl/tree/master/examples/pytorch/ogb/ogbn-arxiv
[22] https://ogb.stanford.edu/docs/nodeprop/
[23] https://github.com/pytorch/torchtitan

**Configurations for Figure 5.** To generate a positive definite matrix $H_b$, we first uniformly sample $\frac{d(d-1)}{2}$ independent angles $\theta_{i,j}$ from the interval $[-\frac{\pi}{2}, \frac{\pi}{2}]$, where $i < j$. Starting with the identity matrix, we perform a rotation of the $i$-th and $j$-th rows by the angle $\theta_{i,j}$ for each sampled pair. Through $\frac{d(d-1)}{2}$ rotation operations, we obtain the orthogonal matrix $Q$. We define $\Lambda = \mathrm{diag}(\kappa, 1, \ldots, 1)$, and the matrix $H_b$ is generated using the expression $H_b = Q\Lambda Q^T$. The python code for $H_b$ generation is listed as follows:

```python
def generate_Hb(theta, kappa, d):
    Q = np.eye(d)
    for i in range(d):
        for j in range(i+1,d):
            P = np.eye(d)
            P[i,i] = math.cos(theta[i,j])
            P[i,j] = math.sin(theta[i,j])
            P[j,i] = -math.sin(theta[i,j])
            P[j,j] = math.cos(theta[i,j])
            Q = P @ Q
    Lambda = np.eye(d)
    Lambda[0,0] = kappa
    return Q @ Lambda @ Q.transpose()
```

We note that as $\theta$ approaches 0, the diagonal-over-off-diagonal ratio of the matrix $Q$ decreases. For the sampled values of $\theta$, we utilize $R\theta$ to produce $H_b$ with varying ratios, where $R \in \{\frac{k}{50}|k = 0, 1, \ldots, 50\}$. For each matrix, we sample 100 initial points from the Xavier initialization distribution to compute the resulting $\kappa$ of Adam algorithm. For each pair of $d$ and $\kappa$, we sample 40 different $\theta$ values. By averaging the results obtained, we plot the Figure 5.

**Configurations for Figure 7.** We use the codebase[24]. We enlarge the vocabulary size from 2 to 8, and we change the objective function to the standard pre-training objective function (i.e., predict the next token). We consider a 1-layer Transfomer with n_emb = 16, n_head = 4, and the width (i.e., the number of output neurons) of `mlp.fc_1` equals 32. The Hessian calculation follows the same procedure as in Figure 3.

It is worth mentioning that the weight matrices might have different shapes depending on the different codebases. For instance, the weight matrix $W$ in one codebase might be implemented as $W^\top$ in another codebase (this can happen for the embedding layer), and the same $W$ might also be stretched into an one-dimensional vector (this can happen for all weight matrices). In these cases, some extra re-ordering is needed. For all the Hessian calculations in Figure 7, we re-order the Hessian rows & columns such that the block-diagonal structure is visible.

**Throughput Comparison in Table 2.** The results are tested on $2\times$ A800-80GB GPUs. We did not turn on CPU offload. We report the throughput from the summary file of the Wandb log.

---

[24]https://colab.research.google.com/drive/1SiF0KZJp75rUeetKOWqpsA8clmHP6jMg?usp=sharing

