# OpenReview forum: "Adam-mini: Use Fewer Learning Rates To Gain More"
_ICLR.cc/2025/Conference — ICLR 2025 Poster_

### Official Review · Reviewer_rJgs · 2024-11-03

**Soundness:** 2
**Presentation:** 3
**Contribution:** 2
**Rating:** 6
**Confidence:** 4

**Summary:**

This paper proposes a novel memory-efficient optimizer Adam-mini to reduce the memory cost by cutting down the learning rate resources in Adam. I tries to partition the parameters into different blocks and assign a single learning rate to each parameter block to reduce the memory cost from `v` in adam. The experimental results illustrate that the proposed method can achieve 49.6% higher throughput than AdamW for LLaMA2-7b pre-training.

**Strengths:**

1. This paper focus on an interesting and important problem. memory-efficient optimization is very important for llm training.

2. The paper provides the results from diverse domains to verify the efficiency and performance of Adam-mini.

3. This paper provides the results on LLM pre-training, SFT and RLHF to verify the performance of proposed method Adam-mini.

**Weaknesses:**

1. The paper is not very easy to follow for me. For example, the paper first aims to partition the model parameters into blocks by Hessian structure. However, in the algorithm part, the paper first tried the PyTorch default partition to construct the blocks for middle-scaled non-transformer tasks and then said the default partition can not work well for transformers. Therefore, they analyzed the exact Hessian at
initialization of transformer training and define the method of partitioning for different layers (such as attention layers, MLP layers). That make me feel a little confused about your proposed method.

2. The motivation is also not very clear to me. I can understand that different blocks need different learning rates. But why a single learning rate can work well for the parameters of the whole block is not very clear to me. I noticed that the authors provided a toy example (random quadratic minimization problem) to show that, but I am not clear whether the conclusion can be general (for neural network training). Because the conclusion is very important and the proposed algorithm is also based on this conclusion, I think the authors could provide more analysis.

3. The most visualization in this paper (Figure 2, 3, 4, 7) can just say different blocks show different patterns, but we cannot obtain the conclusion that the whole parameters of each block should use the same learning rate.

4. The introduction about using the mean of (v * v) to calculate the learning rate is also not very clear. That is also the main contribution of your proposed method. Since you use the mean of (v * v) to change second-order momentum.

5. I can understand that the paper proposes many valuable experimental findings, but the proposed method make me feel a little tricky. In the proposed method, the authors design too many components and make me consider the generality and scalability of the method.

6. The paper should also provide some comparisons with strong baselines, such as Lion. These memory-efficient optimizers is very easy to use and general for different models (resnet and transformers). Therefore, you do not need to consider different algorithm design for different model architecture. They can also save the memory cost and obtain higher flops. Based on that, actually, I don't think adam-mini has too many advantages compared with them.

Although the paper provides the results of Lion, I am not sure whether the authors fully tune the hyper-parameters, because Lion usually need a very different LR. In addition, [1] provides thorough comparisons between different optimizers (memory-efficient optimizers) and the results illustrate that the memory-efficient optimizers can obtain comparable results with Adam.  Therefore, I guess these memory-efficient optimizers can finally achieve a very similar performance.


[1] Deconstructing What Makes a Good Optimizer for Language Models

**Questions:**

1. Figure 6, training loss is not a great metric to compare the performance of different methods, a lower training loss may mean overfitting. It could be better id the authors can provide results on downstream tasks.

---

> ### Author Response · Authors · 2024-11-21
> **Response (Part I)**
>
> We would like to thank the reviewer for the careful evaluation of our paper. We respectfully provide our response as follows.
>
> **Q1: "The paper is not very easy to follow for me. For example, the paper first aims to partition the model parameters into blocks by Hessian structure. However, in the algorithm part, the paper first tried the PyTorch default partition to construct the blocks for middle-scaled non-transformer tasks and then said the default partition can not work well for transformers. Therefore, they analyzed the exact Hessian at initialization of transformer training and define the method of partitioning for different layers (such as attention layers, MLP layers). That make me feel a little confused about your proposed method."**
>
> Thanks for the valuable comment. We will rewrite the paper accordingly to avoid confusion. In the revised version, we will emphasize that there is **only one principled design of Adam-mini**, which has a very simple form as follows:
>
> **Adam-mini (general form):**
>
>      - Step 1. Partition the parameters by Principle 1 in Section 2.3.
>
>      - Step 2. Use mean (v) to calculate the learning rate for each block.
>
>
> Following this general form, we will then get specific realization of partition algorithms on speficic architectures, like Algorithm 2 for Transformers.
>
> **Potential source of confusion:** In the previous version, we directly introduced the specific realization without first presenting the general form.  This might cause the confusion that our proposed method is not principled enough.
>
> **Our proposal to avoid confusion:** In the revised version, we will firstly state the "general principled form of Adam-mini" and then introduce the "the realization of Adam-mini on specific architectures". We will emphasize the there is only one principled design of Adam-mini (shown in the above two steps), and there is no ad-hoc design.
>
> We hope the revised version is clear and does not cause confusion.
>
> **Q2 & 3: " why a single learning rate can work well for the parameters of the whole block is not very clear to me...  I noticed that the authors provided a toy example (random quadratic minimization problem) to show that, but I am not clear whether the conclusion can be general (for neural network training)."  "we cannot obtain the conclusion that the whole parameters of each block should use the same learning rate."**
>
> Thanks for the questions and suggestions. Following your comments, we conduct the following new experiments on a 1-layer Transformer. We will show that "using single lr per block" is also sufficient for Transformers.
>
> The following Exp 1 and 2 extend the random quadratic experiments in Figure 4 and 5 to  Transformers.
>
> **Exp 1: Adam's lr is redundant on the dense Hessian subblock.** We take some small dense blocks in the Hessian of 1-layer Transformer and denoted as $H$. We compare  $\kappa(H)$ and $\kappa(D_{Adam} H)$ as in the paper. We find Adam is not effective in reducing the kappa of these blocks, and many lrs in Adam can be redundant.
>
> **[Table 1 (new)]: Comparison of $\kappa(H)$ and $\kappa(D_{Adam} H)$ for the dense blocks in the Hessian of 1-layer Transformer.**
> | Hessian Block |$\kappa(H)$ |  $\kappa(D_{Adam} H)$|
> |--------|--------|-----------|
> | 1st head in Query| 103.80 | 176.88 |
> | 1st head in Key | 103.46 | 213.82 |
> | 1st head in Value | 165.66 | 332.76 |
> | 1st neuron in attn.proj | 39.92 | 94.56 |
> | 1st neruon in MLP_fc1 | 22.04 | 70.92 |
> | 1st neruon in MLP_c_proj | 63.85| 236.71 |
>
> **Exp 2: single lr per block is sufficient.** We conduct the "block-wise GD" and we grid-search the lr for each block. The result is shown in the following figure. We find that block-wise GD outperforms AdamW.
> This extends the setting from the random quadratic problem in Figure 4.
>
> [[Results on 1-layer Transformer: we can outperform Adam using a single lr per block.]](https://anonymous.4open.science/r/Adam-mini-ICLR-2025-rebuttal-new-014B/1118_rebuttal_blockwisegd_0.001.pdf)
>
> Combining Exp 1 and 2, we can see that Adam is redundant on the dense Hessian subblocks (Exp 1), and a single lr for each block can work well (Exp 2).  These experiments show that our conclusions on random quadratic problems can be extended to Transformers.
>
> We will include these experiments in the revised version.

---

> ### Author Response · Authors · 2024-11-21
> **Response (Part II)**
>
> **Q4: "using the mean of (v) to calculate the learning rate is also not very clear."**
>
> Thanks for the valuable question. In our following response, we assume that you have convinced that "a single lr is sufficient for one block" in our response to **Q2** and **Q3** (please tell us if you still have concern on these questions). We will now further explain "why we choose  mean(v) as the single lr."
>
> - **First: optimal blockwise lrs are too expensive to search.** As shown in the response to **Q3** above, blockwise lr can be powerful given suffcient resource to search them out. However, such procedure too expensive and is not scalable.
>
> - **Second: mean(v) can be borrowed from Adam.** Comparing to  search all the lrs from scratch, it is much easier to "borrow" lrs from the current design of Adam, and mean (v)  is a most natrual quantity to "borrow".  Although mean(v) might not be so powerful as grid-searching, it is much more cost-effective and simple to use. As a result, we find that mean(v) helps Adam-mini to be as  effectiveness as Adam (though not significantly surpassing it).
>
> - **Third: mean(v) keeps us close to Adam.**  Additionally, we find  that mean (v) is a good repsentative for the whole block-v and can help Adam-mini keep close to Adam. This is because: by the backpropagation rules in neural nets, v usually has similar values within a row (which associates with the same output neuron). Therefore, if we want to use a single lr to replace the v for the whole block, the mean of (v) is a good choice.  In the submission script line 372, We provided an intuitive explanation. We paste it here for your convenience.
>
>     **Line 372 in the submission:**  "We now provide an intuitive explanation of why mean (v)  can be a reasonable choice. For one data sample, the gradient of the weight matrix $W \in \mathbb{R}^{d\times d}$ can be expressed as $G:=\frac{\partial \ell}{\partial W} =e z^\top \in \mathbb{R}^{d\times d}$, where $e$ is certain backpropagation (BP) error vector and  $z$ is the input feature to the current weight. For all entries in the $i$-th row of $G$, they all share the same BP error term $e_i$, which is usually non-negligible when $G \neq 0$. Therefore, $G$ usually has similar entries within a row (which associates with the same output neuron), and its mean value can be a good representative of the whole row."
>
>      As a result, we find that mean(v) can make Adam-mini's trajectory close to Adam, this is shown in the figure below.
>
>      [[Figure 10 (c) in the submission: The trajectory of Adam-mini is largley close to that of Adam]](https://anonymous.4open.science/r/Adam-mini-ICLR-2025-rebuttal-new-014B/figure10_in_submission.png)
>
> **An additional side remark:** We provide one additional comment to make this response more complete, but  it may not be directly related to your question and you can skip this if you are not interested.  As for your comment "mean(v * v) is also the main contribution of your proposed method. Since you use the mean of (v * v) to change second-order momentum.", we would like to emphasize that mean(v) is important, but it is just a part of our contribution. Another equally important (if not more important) part is to propose partitioning the parameters by block-diag Hessian structure and reduce the redundant part of Adam's v accordingly. We believe this is a new perspective to slim down optimizers, and we are not aware of any similar idea in the literature.
>
>
> -----**Updated response**-----
>
> **Ablation studies on mean(v)**. We here provide another reason why we chose mean(v) as the block-wise learning rates. We conduct ablation studies on different choices of quantities that we can borrow from Adam, including 1-norm(v), 2-norm(v), max(v), and min(v). we found that all these candidates perform worse than mean(v) in Adam-mini.
>
> [[Click here to view the ablations studies: mean(v) performs better than other candidates]](https://anonymous.4open.science/r/Adam-mini-ICLR-2025-rebuttal-new-014B/1126_ablation_mini.pdf)
>
> These experiments has been updated in the revised version (Appendix C.2)

---

> ### Author Response · Authors · 2024-11-21
> **Response (Part III)**
>
> **Q5: "The authors design too many components and make me consider the generality and scalability of the method."**
>
> Thanks for raising the concern. We believe that Adam-mini is general and scalable. We first respond to the concern on "generality" and then respond to  "scalability".
>
> **As for generality.** We believe Adam-mini is general because it follows a very simple and principled design. We re-state the general form of Adam-mini as follows:
>
> **Adam-mini (general form):**
>
>     - Step 1. Partition the parameters by Principle 1 in Section 2.3.
>
>     - Step 2. Use mean (v) to calculate the learning rate for each block.
>
>
> Following this general form, we will then get the specific realization of partition algorithms on specific architectures, like Algorithm 2 for Transformers.
>
>
> - **Potential source of concern on "generality":** In the submission version, we directly introduce the specific realization without first presenting the general form.  This might cause the concern that our proposed method is complicated and seem not general enough.
>
> - **Our proposal to avoid confusion:** In the revised version, we will firstly state the "general principled form of Adam-mini" and then introduce the "the realization of Adam-mini on specific architectures". We will emphasize the there is only one principled design of Adam-mini (shown in the above two steps), and there is no ad-hoc design.
>
> We hope the revised version can address your concern about "generality". Since the design of Adam-mini is principled, we  believe it is also scalable among diverse tasks and model sizes. We provide more evidence below.
>
>
> --------------------------**Updated Response**-----------------------
>
> **As for scalability.** We believe Adam-mini is scalable and can be broadly applied to the modern models with different architectures and different sizes. We evaborate as follows.
>
> **First, Adam-mini is scalable to diverse tasks and models.** Under our proposed partition Algorithms 2 (for non-Transformers) and 3 (for Transformers), Adam-mini can be directly applied to diverse architectures and tasks and **there is no need to re-design**. We provide the following experiments:
>
>  [[Results in the submission:  ResNet-18]](https://anonymous.4open.science/r/Adam-mini-ICLR-2025-rebuttal-new-014B/figure17_resnet_ddpm.png)
>
> [[Results in the submission:  DDPM diffusion model]](https://anonymous.4open.science/r/Adam-mini-ICLR-2025-rebuttal-new-014B/figure17_resnet_ddpm.png)
>
> [[Results in the submission:  Graph neural networks including GAT and GCN]](https://anonymous.4open.science/r/Adam-mini-ICLR-2025-rebuttal-new-014B/revised_paper_non_llm.png)
>
>
> [[Results in the submission:  GPT-2 series pre-train]](https://anonymous.4open.science/r/Adam-mini-ICLR-2025-rebuttal-new-014B/figure9_in_submission.png)
>
> [[Results in the submission: Llama pre-train in the submission]](https://anonymous.4open.science/r/Adam-mini-ICLR-2025-rebuttal-new-014B/figure12_in_submission.png)
>
> [[New results: RNN pre-train]](https://anonymous.4open.science/r/Adam-mini-ICLR-2025-rebuttal-new-014B/rnn_loss.pdf)
>
> [[New results: BERT pre-train]](https://anonymous.4open.science/r/Adam-mini-ICLR-2025-rebuttal-new-014B/bert_valloss.pdf)
>
> [[New results: ViT (Swin-Transformer) pre-train]](https://anonymous.4open.science/r/Adam-mini-ICLR-2025-rebuttal-new-014B/swin-test-acc.pdf)
>
> [[New results: DiT-XL-2 pre-train]](https://anonymous.4open.science/r/Adam-mini-ICLR-2025-rebuttal-new-014B/1117_dit.pdf)
>
> [[New results: DC-AE-Diffusion pre-train]](https://anonymous.4open.science/r/Adam-mini-ICLR-2025-rebuttal-new-014B/DC-AE-Diffusion.pdf)
>
>
> **Second, Adam-mini is scalable to different model sizes.** In addition to the scalability among different architectures, we also believe Adam-mini is scalable in terms of the model size. This is evident in our experiments in Section 3.3 in the submission: we demonstrated that **Adam-mini is effective on a wide range of LLMs including 39M, 67M, 102M, 167M, 271M, 1B.** This serves as an evidence that Adam-mini Adam-mini can be scaled up to larger models (if the scaling law holds).
>
>
> [[Figure 12 in the submission: the scaling law of Adam-mini]](https://anonymous.4open.science/r/Adam-mini-ICLR-2025-rebuttal-new-014B/figure12_in_submission.png)
>
>
> Hope the above discussion can address your concern on "scalability".

---

> ### Author Response · Authors · 2024-11-23
> **Response (Part IV)**
>
> **Q6. "The paper should also provide some comparisons with strong baselines, such as Lion." "Although the paper provides the results of Lion, I am not sure whether the authors fully tune the hyper-parameters, because Lion usually need a very different LR."**
>
> Thanks for the question.  We conduct more careful lr-grid-search for  Lion in the following experiments.  We acknowledge that Lion has a simple form, but we find that Lion is not easy to tune and we  **have NOT managed to make Lion work** . Here is the setting of our new experiments.
>
> **Architectures:** We will conduct more careful tuning on Lion on Llama 20M. We also tried Lion on GPT-2-125M, which is NOT tested before (neither in Zhao et al. 24 [1] nor in our submission).
>
> **Hyperparameter candidates:** We will learn the optimal tuning strategies in the latest version of Zhao et al. 24 [1, Table 1].  We here summarize their key strategies from their Table 1.
>
> - **Strategy 1 from [1]:** The optimal learning rate of Lion is usually 10 times smaller than AdamW.
>
> - **Strategy 2 from [1]:** The magical number lr = 3.16e-4 works the best for most models (Llama 150M, 300M, 600M).
>
>  We use these strategies on Llama 2-20M and GPT-2-125M. In particular, we try the following lr for Lion.
>
>  1. **For Llama 2-20M:** the standard lr is 5e-3, so we try lr = [5e-4, 6e-4, 7e-4, 8e-4, 9e-4, 1e-3, 2e-3, 3e-3, 4e-3, 5e-3, 0.000316]
>  2. **For GPT-2-125M:**  the standard lr is 6e-4, so we try lr = [6e-5, 7e-5, 8e-5, 9e-5, 1e-4, 2e-4, 0.000316, 4e-4, 5e-4, 6e-4]
>
> **Results:** After using the above tuning strategies, we find that Lion **still underperforms** Adam-mini and AdamW on Llama 2-20M and GPT-2-125M. In particular, **Lion encounters loss spikes** on GPT-2-125M for all the lr candidates above.
>
> [[Click here to view the results of Llama 2-20M: Lion underperforms Adam-mini]](https://anonymous.4open.science/r/Adam-mini-ICLR-2025-rebuttal-new-014B/1114_lion_20M.pdf)
>
> [[Click here to view the results of GPT-2-125M: Lion encounters loss spikes]](https://anonymous.4open.science/r/Adam-mini-ICLR-2025-rebuttal-new-014B/lion_gpt2.pdf)
>
> We summarize our findings below.
>
> **First: worse performance.** With all the effort above,  we haven't managed to make Lion work, and we haven't been able to reproduce the results in  [1]. One possible reason is that Lion might work under their specific setup (dataset, architecture, batch size, etc.), but the effectiveness is not easily transferable.
>
> **Second: no general tuning guidance.** We find that there are no general tuning strategies for Lion. We emphasize that [1] only focuses on Llama architectures,  and their resulting tuning strategy seems not robust and transferable to other architectures. In particular, **their optimal strategy on Llama causes loss spikes on  GPT-2**. To our knowledge, Table 1 in Zhao et al. (2024) [1] is the only public tuning strategy for Lion, so it seems unclear how to tune Lion in general.
>
> **In contrast, Adam-mini is much easier to use.** In all our experiments (wide range of tasks and models), Adam-mini performs on par with Adam **using the same hyperparameters as AdamW** (including learning rate, $\beta_1,\beta_2, \epsilon $, etc.). We believe that "easy adaptation of the hyperparameters" can serve as one advantage of Adam-mini over Lion, apart from the performance superiority.
>
> **Third: Adam-mini is more principled and explainable.** We emphasize that Lion is designed by symbolic search, and its design principle is largely unclear. In contrast, the Design principle of Adam-mini is much more understandable: we remove the redundant lrs in Adam according to the Hessian structure. We believe Adam-mini is more "white-box" than Lion and more explainable to users.
>
>
>  We will include these results in the revised script.
>
>  [1] Deconstructing What Makes a Good Optimizer for Language Models  https://openreview.net/forum?id=zfeso8ceqr

---

> > ### Author Response · Authors · 2024-11-23
> > **Response （Part V）**
> >
> > **Q7. "Figure 6, training loss is not a great metric to compare the performance of different methods, a lower training loss may mean overfitting. It could be better if the authors can provide results on downstream tasks."**
> >
> > Thanks for the suggestion. Figure 6 focuses on small-scaled Transformers and we did not find suitable downstream tasks. Nevertheless, we evaluate the model performance on an independently sampled validation set and report the validation perplexity.  We find that Adam(leave-one-out) still outperforms AdamW. This shows that it is possible to reach good performance using much fewer lrs.
> >
> > [[Click here to see the validation perplexity of the Transformers in Figure 6 (a)]](https://anonymous.4open.science/r/Adam-mini-ICLR-2025-rebuttal-new-014B/1118loss_random_leave_one_out_0.001.pdf)
> >
> > [[Click here to see the validation perplexity of the Transformers in Figure 6 (b)]](https://anonymous.4open.science/r/Adam-mini-ICLR-2025-rebuttal-new-014B/1118loss_random_leave_two_out_0.001.pdf)
> >
> > [[Click here to see the validation perplexity of the Transformers in Figure 6 (c)]](https://anonymous.4open.science/r/Adam-mini-ICLR-2025-rebuttal-new-014B/1118loss_random_leave_three_out_0.001.pdf)
> >
> > Thanks again for the great questions. We are happy to further discuss any other potential follow-up questions.

---

> ### Author Response · Authors · 2024-11-25
> **We have revised the paper following your suggestions (related to Q1 and Q5)**
>
> Dear reviewer:
>
> We have revised the paper following your comments related to the presentation (**Q1** and **Q5** in particular).
>
> **Q1 restated: "The paper is not very easy to follow for me. For example, the paper first aims to partition the model parameters into blocks by Hessian structure. However, in the algorithm part, the paper first tried the PyTorch default partition to construct the blocks for middle-scaled non-transformer tasks and then said the default partition can not work well for transformers. Therefore, they analyzed the exact Hessian at initialization of transformer training and define the method of partitioning for different layers (such as attention layers, MLP layers). That make me feel a little confused about your proposed method."**
>
> **Q5: "The authors design too many components and make me consider the generality and scalability of the method."**
>
> **Our revision to address Q1 and Q5:** To address your concern, we have rewritten  Section 2.2 (Introducing Adam-mini). We now introduce Adam-mini. We will first state the **"general principled form of Adam-mini"** and
> then introduce the **"the realization of Adam-mini on specific architectures"**. This shows that Adam-mini is a general and principled method.
>
> [[The 1st part relevant to Q1 and Q5]](https://anonymous.4open.science/r/Adam-mini-ICLR-2025-rebuttal-new-014B/revised_paper.png)
>
>
> [[The 2nd part relevant to Q1 and Q5]](https://anonymous.4open.science/r/Adam-mini-ICLR-2025-rebuttal-new-014B/revised_paper_part2.png)
>
> We also provide new experiments on diverse Transformer tasks. These results show that Adam-mini is scalable among different Transformers.
>
> [[The part relevant to Q5]](https://anonymous.4open.science/r/Adam-mini-ICLR-2025-rebuttal-new-014B/revised_paper_part2.png)
>
>
> Hope these revisions will make the method easier to understand and show the generality and scalability of Adam-mini.

---

> ### Author Response · Authors · 2024-11-26
> **We have revised the paper following your suggestions (related to Q4)**
>
> **Q4 restated: "The introduction about using the mean of (v * v) to calculate the learning rate is also not very clear. "**
>
> **Our reivision to address Q4:** To eliminate the confusion you mentioned in **Q4**, we have added the following discussions in Section 2.4. Note that the first and second bullets are new, and the third bullet is re-arranged based on the original paragraph in the submission.
>
> **Why using average $v$ as learning rates.** In Line 9 of **Algorithm 1**, we use the average of $v$ in a block as the learning rate for that block. We choose such a design due to the following reasons:
>
> - **First: grid-search is too expensive.**
> - **Second: average of $v$ can be borrowed from Adam.**
> - **Third: average of $v$ keeps us close to Adam.**
>
> [[The part relevant to Q4]](https://anonymous.4open.science/r/Adam-mini-ICLR-2025-rebuttal-new-014B/revised_paper_part3.png)
>
> -----**Updated response**-----
>
> **Ablation studies on mean(v)**. We here provide another reason why we chose mean(v) as the block-wise learning rates. We conduct ablation studies on different choices of quantities that we can borrow from Adam, including 1-norm(v), 2-norm(v), max(v), and min(v). we found that all these candidates perform worse than mean(v) in Adam-mini.
>
> [[Ablations studies in the revised version (Appendix C.2): mean(v) performs better than other candidates.]](https://anonymous.4open.science/r/Adam-mini-ICLR-2025-rebuttal-new-014B/revised_paper_appendixc2.png)

---

> ### Author Response · Authors · 2024-11-26
> **We have revised the paper following your suggestions (related to Q2, Q3, and Q6)**
>
> Dear reviewer:
>
> We also revised the paper following your comments on **Q2, Q3, Q6**. Please click the following links to see the corresponding changes in the paper.
>
> **As for Q2 and Q3:** We add more experiments on a small Transformer to show that “one lr per block” is sufficient for Transformer training. These experiments extend our previous random quadratic experiments in Figure 4 and 5.
>
> [[The part relevant to Q2 and Q3]](https://anonymous.4open.science/r/Adam-mini-ICLR-2025-rebuttal-new-014B/revised_paper_Q23.png)
>
> **As for Q6.** We compare Adam-mini and Lion on GPT-2 and Llama models. We find that Lion is not easy to tune and the optimal tuning strategies in [1] is not robust and transfer-able among different models. It seems unclear how to tune Lion in general and we still haven't managed to make Lion perform well.
>
> [[The part relevant to Q6 (Part I)]](https://anonymous.4open.science/r/Adam-mini-ICLR-2025-rebuttal-new-014B/revised_paper_Q6_1.png)
>
> [[The part relevant to Q6 (Part II)]](https://anonymous.4open.science/r/Adam-mini-ICLR-2025-rebuttal-new-014B/revised_paper_Q6_2.png)
>
> [1] Anonymous authors. Deconstructing what makes a good optimizer for language models.
> https://openreview.net/pdf?id=zfeso8ceqr, 2024.
>
> We hope the above explanation fully addresses your concern. Your feedback is highly valuable,  Your comments are valuable and we have put much effort into addressing your concern.  We would love to receive feedback from you. We are more than happy to engage in further discussions.

---

> ### Author Response · Authors · 2024-11-27
> **An one-page slides to explain the logic of the paper**
>
> Dear Reviewer:
>
> Thank you once again for the time and effort you have dedicated to reviewing our paper. We greatly appreciate your insightful comments and suggestions. As we have not yet received a response to our previous reply, we are concerned that may still be aspects of the paper that are unclear to you. To better address your concerns and to help you better understand the paper, we respectfully prepared a one-page slide that outlines the underlying logic behind the design of Adam-mini. We hope this will be helpful in enhancing your understanding of our work.
>
>
> [[Click here to view the one-page slide: The logic behind the design of Adam-mini]](https://anonymous.4open.science/r/Adam-mini-ICLR-2025-rebuttal-new-014B/logic_adam_mini.png)
>
> -------------------**Summary of the slide**-------------------
>
> We also summarize the logic here at your convenience. Here is the logic behind the design of Adam-mini.
>
> **Observation 1:**  Adam is good because it assigns different lr to different Hessian blocks (Evident in Figure 4)
>
> **Observation 2:** Adam is bad on the dense Hessian sub-block (Evident in Figure 4, 5, 6, Table 4, and Figure 14)
>
>
> Together they reveal:
>
> **Implication:** Adam mainly handles the “cross-block” challenge, NOT the “within-block” challenge.
>
> **Further Implication:** We may replace Adam’s coordinate-wise lr by block-wise lr
>
> **But how to design block-wise lr?**  We can borrow from Adam’s v!
>
> **Then we arrive at Adam-mini.**
>
> -----------------**End**---------------------
>
> We hope the above explanation is helpful. We would love to receive feedback from you. Your comments and suggestions would be valuable for us in improving the paper. We are more than happy to further discuss with you.

---

> ### Comment · Reviewer_rJgs · 2024-11-28
> **Thank you very much for your rebuttal!**
>
> Thank you for addressing my concerns. I'm pleased to increase the score.
>
> Currently, I still have several concern:
>
> - Q2: Regarding single learning rate per block: While experiments show this works well, the paper lacks theoretical analysis or intuition to support this design choice.
>
> - Q5: On generality: the approach requires partitioning parameters based on Hessian sub-block structure and layer types. Unlike Adam, this makes adaptation to new architectures or layer types challenging, as the partitioning strategy isn't immediately clear.
>
> - Lion comparison: The experimental results cannot convince me. The authors mentioned they cannot find a great lr to make Lion work well. Your claim about Lion's performance would be more convincing with experiments on the public models and datasets from the Lion paper.
>
> - Baselines: Given the active research in memory-efficient optimizers, comparing only against Adam is insufficient.

---

> ### Author Response · Authors · 2024-11-30
> **Response to your follow-up comments (Part I)**
>
> We sincerely thank the reviewer for the careful re-evaluating our paper and raising the score! We reply to your follow-up questions as follows.
>
>
>
> **Comment 1: "Q2: Regarding single learning rate per block: While experiments show this works well, the paper lacks theoretical analysis or intuition to support this design choice."**
>
> Thanks for mentioning this. We agree it is important to investigate the theory to support the design of "single lr per block". However, believe that our current numerical findings on Transformer Hessian structures are new, valuable, and they are sufficient to support the design choice of Adam-mini.  A rigorous theory on Transformer models would be challenging and we believe it is worth an independent theoretical paper.
>
> **Comment 2: "Q5: On generality: the approach requires partitioning parameters based on Hessian sub-block structure and layer types. Unlike Adam, this makes adaptation to new architectures or layer types challenging, as the partitioning strategy isn't immediately clear."**
>
>
>
> Thanks for the comment. We acknowledge that it can be expensive to re-design partition on a new architecture. Meanwhile, we would like to emphasize that Adam-mini can be directly applied to the currently mainstream Transformer models and **there is no need to re-design**. We believe Adam-mini still have substantial generality in the current AI community. We elaborate as follows.
>
>
> - **First: Our Algorithm 2 (partition for Trasformers) works well on a wide range of Transformers.** For Transformers, we propose to partition QK by head and MLP by output neurons. We find this strategy is effective on a wide range of Transformers, and there is no need to re-design the partition. In the submission, we provide evidence on various GPT architectures, including:
>
>     [[Results in the submission:  GPT-2 series pre-train]](https://anonymous.4open.science/r/Adam-mini-ICLR-2025-rebuttal-new-014B/figure9_in_submission.png)
>
>     [[Results in the submission: Llama pre-train in the submission]](https://anonymous.4open.science/r/Adam-mini-ICLR-2025-rebuttal-new-014B/figure12_in_submission.png)
>
>     For completeness, we also provide more new results on other Transformers. We find Adam-mini is also effective without changing the partition algorithm.
>
>     [[New results: BERT pre-train]](https://anonymous.4open.science/r/Adam-mini-ICLR-2025-rebuttal-new-014B/bert_valloss.pdf)
>
>     [[New results: ViT (Swin-Transformer) pre-train]](https://anonymous.4open.science/r/Adam-mini-ICLR-2025-rebuttal-new-014B/swin-test-acc.pdf)
>
>     [[New results: DiT pre-train in the submission]](https://anonymous.4open.science/r/Adam-mini-ICLR-2025-rebuttal-new-014B/1117_dit.pdf)
>
>
>     These results indicate that our Algorithm 2 (partition for Transformers) can be directly applied to a wide range of Transformers, and there is no need to re-design. This indicates that the current form of  Adam-mini (under Algorithm 2)  is sufficiently scalable among Transformers.
>
> - **Second: Transformers are ubiquitous.** We acknowledge that we put the majority of efforts on designing Adam-mini for Transformers. However, given the tremendous impact of Transformers and LLMs, we believe that the resulting form of  Adam-mini (under the partition Algorithm 2) will be repeatedly used in a wide range of applications.
>
>
> - **Third: Some designs in Adam-mini can be repeatedly used.** We would also like to point out that some partition strategies are actually quite general. For instance, we always partition the MLP layer by output neuron. This partition rule is applicable not only for Transformers, but also for generic MLPs as in Figure 3. Such a strategy can be repeatedly used in various of architectures.

---

> ### Author Response · Authors · 2024-11-30
> **Response to your follow-up comments (Part II)**
>
> **Comment 3: "Lion comparison: The experimental results cannot convince me. The authors mentioned they cannot find a great lr to make Lion work well. Your claim about Lion's performance would be more convincing with experiments on the public models and datasets from the Lion paper."**
>
> We acknowledge that Lion can be effective on some tasks such as vision, and BERT-like model (as shown in Lion paper [1]). We would like to emphasize that we trust these results from Lion paper and we did NOT argue that "Lion is universally bad". Instead, we emphasize that  Lion is **not easy to tune on modern LLM tasks**, and the currently-known optimal tuning strategy (from both Lion paper [1] and [2]) is **NOT transfer-able among different  LLM architectures.**  This is evident in the screenshot below: Lion **encounters loss spike** on GPT-2-125M under the current recommended tuning strategies.
>
> **We restate our experiments here.** We find that Lion encounters loss spikes on GPT-2-125M when using the currently known optimal tuning strategies from Lion paper [1] and [2]. We summarize the tuning strategy below:
>
> 1. **The recommended lr of Lion is "is typically 3-10x smaller than that for AdamW"**. This is stated in the Lion paper [1] (Section 5) and agreed by Table 1 in [2].
> 2. **The recommended hyperparam are "beta1 = 0.95 and beta2 = 0.98"**. This is claimed to "enhanced training stability" in the Lion paper (Section 5) and agreed by Figure 4 and 5 in [2].
>
>
> On GPT2-125M, we test Lion using the above tuning strategies, and we find that Lion encounters loss spikes. It remains unclear how to tune Lion in general.
>
> [[Screenshot for Appendix C.6: Part I]](https://anonymous.4open.science/r/Adam-mini-ICLR-2025-rebuttal-new-014B/revised_paper_Q6_1.png)
>
> [[Screenshot for Appendix C.6: Part II]](https://anonymous.4open.science/r/Adam-mini-ICLR-2025-rebuttal-new-014B/revised_paper_Q6_2.png)
>
>
> We acknowledge that it is still possible to make Lion work on a new LLM when given sufficient resources for tuning.  However, we are not aware of any generally effective tuning guidance. This can cause trouble for engineers: without a reliable tuning strategy, it is rather risky to deploy Lion in super-large models.
>
>
> [1] Chen et al. Symbolic discovery of optimization algorithms.
>
> [2]  Anonymous authors. Deconstructing what makes a good optimizer for language models. https://openreview.net/pdf?id=zfeso8ceqr, 2024.
>
>
> **Comment 4: "Baselines: Given the active research in memory-efficient optimizers, comparing only against Adam is insufficient."**
>
> We kindly disagree that we "compare only against Adam". In Section 3 in the submission, we have carefully compared  Adam-mini with other baseline methods including Adafactor, CAME, and SM3. We restate the results as follows. We find Adam-mini performs better.
>
> [[Figure 9 in the submission: comparison with Adafactor, SM3 on GPT-2-125M ]](https://anonymous.4open.science/r/Adam-mini-ICLR-2025-rebuttal-new-014B/figure9_in_paper.png)
>
> [[Figure 10 in the submission: comparison with Adafactor, CAME on Llama 2-1B ]](https://anonymous.4open.science/r/Adam-mini-ICLR-2025-rebuttal-new-014B/figure10_in_paper.png)
>
> [[Section 3.4 in the submission: comparison with Adafactor on Llama 2-20M]](https://anonymous.4open.science/r/Adam-mini-ICLR-2025-rebuttal-new-014B/section3_4_in_submission.png)
>
> We believe we have compared all the existing baseline methods. We humbly request the reviewer to explicitly point out what other baseline methods that is missing in the script. Please point out the specific paper in your mind, and we are more than happy to compare.
>
>
> We would like to thank the reviewer again for the careful review of our paper! We hope the above explanation is helpful for addressing your follow-up concerns.  We would love to receive feedback from you.

---

> > ### Comment · Reviewer_rJgs · 2024-12-03
> > **Thank you very much for your response**
> >
> > Thank you for your thorough response addressing my concerns. However, I still have concerns regarding the baseline comparisons.
> >
> > My primary concern relates to the comparatively weak performance of the baseline optimizers (Adafactor, CAME, and SM3). This raises questions about whether their hyperparameters were fully optimized for fair comparison. This concern is particularly relevant given the recent findings in [1], where experiments on OLMo pre-training with carefully tuned hyperparameters demonstrated comparable performance among Adam, Adafactor, and Lion optimizers.
> >
> > While I appreciate that Adam-mini achieves performance comparable to Adam, which effectively demonstrates its value and contribution, the reliability of baseline comparisons remains crucial for a comprehensive evaluation. I would encourage the authors to include additional experimental results with fully tuned baseline optimizers in the final version to strengthen the comparative analysis. Finally, I am pleased to raise my score.
> >
> > [1] Deconstructing What Makes a Good Optimizer for Language Models.

---

> ### Author Response · Authors · 2024-12-04
> **Thanks for raising the score and some further clarifications**
>
> We sincerely thank the reviewer for the time and effort in reading our response and raising the score. We totally agree that baseline comparison is important and this is exactly why we have devoted two sections (Section 3.4 and Appendix C.6) to conducting hyperparameter grid-search on Adafactor.  We also provided the throughput comparison as well.  Nevertheless, we will follow the reviewer's suggestions and include more of such experiments. Thanks again for your support and valuable suggestions!
>
>
> ---
>
> **A Restatement of our experiments about Adafactor in the submission.** For the convenience of reviewers, ACs, and future readers, we still would like to clarify that **we have tried our best to tune Adafactor** for a fair comparison, yet we haven't managed to make it work.  We restate our experiments from the submission "Section 3.4 DETAILED COMPARISON WITH ADAFACTOR" and "Appendix C.6 DETAILED COMPARISON WITH ADAFACTOR"
>
> **Experimental setting.** We considered two versions of Adafactor: The original version with momentum, and the modified version in Zhai et al.22 and we call it  Adafactor-Zhai-version. We consider Llama 2-20M and 1B pre-train and try the following hyperparemeters:
>
> 1. Default hyperparams with lr = [1e-5, 5e-5, 1e-4, 5e-4, 1e-3, 5e-3, 1e-2]
>
> 2. We change the default β2 = 0.999 to β2 = 0.95 and sweep over lr =  [1e-5, 5e-5, 1e-4, 5e-4, 1e-3, 5e-3, 1e-2]
>
> 3. We use lr = 5e-3, β2 = 0.95 and sweep over warm-up step = [1%, 2%, 3%, 4%, 5%, 10%] total steps.
>
> 4. We use lr = 5e-3, β2 = 0.95 and warm-up step = 1% total steps and sweep over ϵ = [10−30 , 10−16 , 10−8 , 10−6].
>
>
>
> **Finding 1: Adafactor is not easy to tune.**  With all the effor above, Adafactor is still worse than AdamW and Adam-mini. We would like to quote our comments in line 512 in the submission here:
>
>  "We acknowledge that it might be possible to improve these methods if we spend more resources on grid search (as claimed by a recent work (Zhao et al., 2024b)). However, based on our experience so far, it is not easy to tune these methods, and to our knowledge, there is no much open-source guidance. **Recall that there are 9 tunable hyperparameters in Adafactor, so it is rather non-trivial to find the correct combination.** In contrast, Adam-mini is much easier to use. In all our experiments, Adam-mini performs well using the same hyperparameters as AdamW."
>
>
>
> **Finding 2: Adafactor has higher latency.** Besides the performance comparison, we further find that **Adafactor has 40% higher latency.** This means Adam-mini would take **28.5% less wall-clock time** than Adafactor to handle the same amount of token. The similar latency issues also apply to other variant of Adafactors such as CAME. We quote our comments in line 529 in the submission here:
>
> "Besides the performance comparison, we further find that Adafactor has higher latency than Adam-mini (Figure 13 (c)). This is primarily due to two reasons. First, Adam-mini only requires computing the mean by rows of the weight matrix, whereas Adafactor needs to sum across both the rows and the columns. Second, the dimension of v in Adam-mini equals the output dimension or the number of heads, which is significantly smaller than the dimension of v in Adafactor, which equals the product of the input and output dimension. Consequently, Adam-mini saves computation when taking the square root of v. As such, Adam-mini reaches higher throughput."
>
>
> [[Screenshot for "Section 3.4: DETAILED COMPARISON WITH ADAFACTOR" in the submission]](https://anonymous.4open.science/r/Adam-mini-ICLR-2025-rebuttal-new-014B/appendix_c_6_in_submission.png)
>
> [[Screenshot for "Appendix C.6: DETAILED COMPARISON WITH ADAFACTOR" in the submission]](https://anonymous.4open.science/r/Adam-mini-ICLR-2025-rebuttal-new-014B/section3_4_in_submission.png)
>
> To sum up, we appreciate the results in [1], but it seems unclear how to tune Adafactor in general.   We would like to emphasize again that  there are 9 tunable hyperparameters in Adafactor, so it is rather non-trivial for us (and also for potential users) to find the correct combination. Further, we find that Adafactor has 40% higher latency than Adam-mini. This means Adam-mini would take 28.5% less wall-clock time than Adafactor to handle the same amount of tokens.  We believe that "less wall-clock time" is also an important advantage of Adam-mini, apart from "no-extra-tuning" and "good performance".
>
>
>
> [1] Deconstructing What Makes a Good Optimizer for Language Models.

---

### Official Review · Reviewer_8UzS · 2024-11-03

**Soundness:** 4
**Presentation:** 3
**Contribution:** 3
**Rating:** 8
**Confidence:** 3

**Summary:**

This paper presents Adam-mini, a variant of Adam with significantly reduced memory requirements. The approach leverages the near-block-diagonal structure observed in the Hessian of neural networks, and assigns a single learning rate to groups of parameters within a block, thereby reducing memory usage substantially.

The Authors observe limitations in PyTorch’s default partition, particularly for Transformers. To address this the Authors examine the Hessian structure of a small Transformer and identify three distinct classes of Hessian sub-blocks within Transformers and propose a tailored partitioning strategy based on this. Additionally, the authors introduce a simple method to determine a suitable learning rate for each block by assigning the average of Adam’s $\upsilon$ to that block.

The claims are supported by various experiments including pre-training experiments on GPT-2 and Llama series where Adam-mini is compared against AdamW and other SOTA memory-efficient Adam variants, which demonstrate the effectiveness of Adam-mini in reducing memory overhead while maintaining performance.

**Strengths:**

The paper is well-written, and the proposed idea is novel. The approach, though simple, achieves a substantial reduction in memory usage, higher throughput, and shorter wall-clock time without compromising performance. Given the increasing adoption of LLMs and their high demands on memory resources, this method has the potential to be impactful in memory-constrained LLM applications.

**Weaknesses:**

The method proposed in the paper does not seem to address large-scale non-Transformer models. Although Section C.2 includes experiments on various non-LLM tasks, it would be helpful to discuss whether Adam-mini offers advantages for large-scale non-LLM applications or if AdamW is preferred in these cases.

Moreover, given the success of the method on GPT-2 and Llama series and its potential impact, demonstrating the effectiveness of Adam-mini on more diverse models, such as BERT-like and vision transformers would be highly beneficial.

**Questions:**

Minor referencing errors and typos:

- Line 294: instability on 1B models (see Figure 7 (d)) → instability on 1B models (see Figure 8)
- Line 349-350: As shown in Figure 7 (d). This strategy → As shown in Figure 8, this strategy
- Lines 1053-1054: The results are shown in Figure 15 → The results are shown in Figure 16
- Lines 418-419: Figure 9 (b) shows the loss curves → Figure 9 shows the loss curves
- Line 483: As shown in Figure 12 → As shown in Figure 12 (a)
- Table1 does not seem to be referenced except in the caption of Figure 14 in the appendix
- There appears to be no reference to Figure 19 in Section D.2
- What is “Adam-mini-2e-4-1B-v” in Figure 12 (a)?
- Line 374: BP in “where e is certain BP error” is defined 2 lines later.
- Lines 449-450: This might because → This might be because

---

> ### Author Response · Authors · 2024-11-21
> **Response**
>
> We would like to thank the reviewer for the careful evaluation  and supportive comments!  We respectfully provide our response as follows.
>
>  **Q1. "it would be helpful to discuss whether Adam-mini offers advantages for large-scale non-LLM applications. Moreover, ... the effectiveness of Adam-mini on more diverse models, such as BERT-like and vision transformers would be highly beneficial"**
>
> Following the reviewer's suggestions, we conduct the following extra large-scale non-LLM experiments. We train all models on ImageNet.
> 1. Pre-training Vision Transformers (Swin-Transformer-88M).
> 2. pre-training Diffusion Transformers (DiT-XL-2-675M),
> 3. Pre-training SoTA Diffusion models (DC-AE-Diffusion-500M),
>
> To ensure a fair comparison, we use the recommended hyperparam setting in the codebase (learning rates, batchsize, etc.) and we do not specifically tune hyper-parameters for our methods.
>
> **[Table 1 (new)]: Adam-mini performs on par with AdamW on Swin-Transformer, DiT, Diffusion models.**
> | Domain | Model | Optimizer | Metric | 25% steps | 50% steps | 75% steps | 100% steps |
> |--------|--------|-----------|---------|------------|------------|------------|-------------|
> | Vision | Swin-Transformer | AdamW | Val acc (↑) | **0.6290** |0.6940  | **0.7180**| **0.7310** |
> | Vision | Swin-Transformer | Adam-mini | Val acc (↑) | 0.6230 | **0.6960** | 0.7160 | 0.7300 |
> | Vision | DiT-XL-2 | AdamW | Train loss (↓) | 0.1605 | 0.1696 | 0.1607 | 0.1431 |
> | Vision | DiT-XL-2 | Adam-mini | Train loss (↓) | **0.1601** | **0.1693** | **0.1605** | **0.1430** |
> | Vision | DC-AE-Diffusion | AdamW | Train loss (↓) | 0.2860| **0.2820** | 0.2800 | 0.2780 |
> | Vision | DC-AE-Diffusion | Adam-mini | Train loss (↓) | **0.2860** | 0.2830 | **0.2800** | **0.2780**|
>
>
> [[Click here to view the loss curve of Swin-Transformer]](https://anonymous.4open.science/r/Adam-mini-ICLR-2025-rebuttal-new-014B/swin-test-acc.pdf)
>
> [[Click here to view the loss curve of DiT-XL-2]](https://anonymous.4open.science/r/Adam-mini-ICLR-2025-rebuttal-new-014B/1117_dit.pdf)
>
> [[Click here to view the loss curve of DC-AE-Diffusion]](https://anonymous.4open.science/r/Adam-mini-ICLR-2025-rebuttal-new-014B/DC-AE-Diffusion.pdf)
>
> We also evaluate the FID scores for the images generated by DiT and Diffusion models. The FID score for Adam-mini and AdamW  are very similar.
>
> **[Table 2 (new)]: Evaluation scores: Adam-mini performs on par with AdamW.**
> | Domain | Model | Optimizer | FID (↓) | Inception Score (↑)  |
> |--------|--------|-----------|---------|------------|
> | Vision | DiT-XL-2 | AdamW | 91.83 | 12.38 |
> | Vision | DiT-XL-2 | Adam-mini | **88.20** | **13.90** |
> | Vision | DC-AE-Diffusion | AdamW | 34.72  | 41.79 |
> | Vision |  DC-AE-Diffusion | Adam-mini | **33.15** | **44.38** |
>
> As for BERT, we conduct new experiments of BERT on Cornell Movie-Dialogs Corpus following the standard setup in [1]. We find that Adam-mini performs on par with AdamW.
>
> **[Table 1 (new)]: Adam-mini performs on par with AdamW on BERT.**
> | Domain | Model | Optimizer | Metric | 25% steps | 50% steps | 75% steps | 100% steps |
> |--------|--------|-----------|---------|------------|------------|------------|-------------|
> | NLP | BERT | AdamW | Val loss (↓) | 6.9493 | **4.1619**  |**4.0623** | 3.6647 |
> | NLP | BERT | Adam-mini | Val loss (↓) | **6.9437**| 4.1921|4.1018 | **3.6248** |
>
> [[Click here to view the loss curve of BERT]](https://anonymous.4open.science/r/Adam-mini-ICLR-2025-rebuttal-new-014B/bert_valloss.pdf)
>
> [1] https://medium.com/data-and-beyond/complete-guide-to-building-bert-model-from-sratch-3e6562228891
>
> We will include the above experiments in the revised script, and hope they can address your concerns.
>
> **Q2. Errors and typos.**
>
> Thanks so much for pointing them out! We will correct these errors and typos in the revised version.

---

> > ### Comment · Reviewer_8UzS · 2024-11-25
> >
> > I thank the Authors for their response and the additional experiments. I would like to ask if the Authors have evaluated how Adam-mini compares to AdamW in large models outside the Transformer architecture, such as larger variants of the ResNet family (e.g. ResNet50, ResNet101 / ImageNet)?

---

> ### Author Response · Authors · 2024-11-26
> **Results of ResNet-50 and ResNet-101 on ImageNet**
>
> Dear Reviewer:
>
> Following your suggestion, we ran the experiments overnight with all the machines available. Here is the result so far.  Please note that these runs cannot be finished in just one or two days due to the large scale. Nevertheless, according to the curves so far, **Adam-mini is slightly better than AdamW.**
>
>
> [[ResNet-50 on ImageNet: Adam-mini is slightly better than AdamW]](https://anonymous.4open.science/r/Adam-mini-ICLR-2025-rebuttal-new-014B/resnet50_valacc_adam_mini.pdf)
>
> [[ResNet-101 on ImageNet: Adam-mini is slightly better than AdamW]](https://anonymous.4open.science/r/Adam-mini-ICLR-2025-rebuttal-new-014B/resnet101_valacc_adam_mini.pdf)

---

> > ### Comment · Reviewer_8UzS · 2024-11-26
> >
> > I appreciate the efforts made by the Authors and support the publication of the paper.

---

### Official Review · Reviewer_waYV · 2024-11-03

**Soundness:** 3
**Presentation:** 3
**Contribution:** 3
**Rating:** 8
**Confidence:** 4

**Summary:**

This paper introduces Adam-Mini, a new Adam variant for reducing the training memory cost. It changes Adam's parameter-wise learning rate scaling to block-wise learning rate scaling to reduce the cost of the second-order optimizer state. Extensive experiments show that Adam-mini can achieve on-par or better performance than AdamW with less memory footprint.

**Strengths:**

1. This paper studies an important problem and provides an interesting and novel solution.
2. The paper is well-written with clear motivation and good presentation flow.
3. Extensive experiments are conducted to justify Adam-mini's effectiveness.

**Weaknesses:**

1. Integrating Adam-mini into existing training frameworks seems non-trivial. For example, according to Algo 2, we need to specify the partitions manually for large models. Is there any solution to automatically get a good partition given general Pytorch models?
2. It will be good to add discussions/insights about how to adapt hyper-parameters when switching from AdamW to Adam-mini.
3. In addition to training curves, it will be better to have comparisons after model convergence. Additionally, it will be interesting to see results on diffusion transformers for diffusion model experiments. Some public diffusion transformer training codebases:
- https://github.com/facebookresearch/DiT
- https://github.com/mit-han-lab/efficientvit/blob/master/applications/dc_ae/README.md

**Questions:**

See weaknesses.

---

> ### Author Response · Authors · 2024-11-21
> **Response (Part I)**
>
> We would like to thank the reviewer for the careful evaluation of our paper! We greatly appreciate your insightful comments and suggestions! We provide our response as follows.
>
> **Q1. "Integrating Adam-mini into existing training frameworks seems non-trivial... Is there any solution to automatically get a good partition given general Pytorch models?"**
>
> Thanks for the thoughtful question.  We believe our current implementation (our code in the attachment) is **simple enough**, and the partition can be implemented automatically.  This is done in the following two steps.
>
> **Step 1:** Given an LLM model, we identify each Query, Key, MLP, etc. using the names in model.named_parameters().
>
> **Step 2:**, For each block,   we do NOT actually implement "first partition and then take mean". Instead, we take an equivalent operation: **we directly take mean over certain dimension of the matrix $g \circ g$**.
>
> We will use a simple example to illustrate **Step 2**. For instance, given a linear model $$f_W(x) = Wx \in \mathbb{R}^2$$ with input $x \in \mathbb{R}^3$ and  weight matrix $W = \begin{bmatrix} w_1 & w_2& w_3 \\\\ w_4 &w_5 & w_6 \end{bmatrix} \in \mathbb{R}^{2\times 3}$.
>
> By the Hessian structure analysis in Eq. (1), we need to partition this W by output neuron and then take mean for each output neuron. Finally, we will get 2 learning rates (where 2  = # output neuron). Actually,   this operation is equivalent to directly taking mean by rows (calculate the mean of the first row and 2nd row, respectively, and get 2 learning rates). This is because the weights associated with the 1st output neuron are just the first row (w1, w2, w3) . The code is just three lines:
> ```
>  lr = torch.mean(grad * grad, dim=1, keepdim=True)
>  state["vmean"].mul_(beta2).add_(lr, alpha=1 - beta2)
>  update = state["m"]  / state["vmean"]
> ```
> For Query and key, we need to take mean over heads. In this case, we just need to do one additional step:  reshape the Query matrix to have num_head rows, and then take the mean using the same procedure as above.
>
> ```
>   grad = grad.view(num_head, -1)
>   lr = torch.mean(grad * grad, dim=1, keepdim=True)
>   state["vmean"].mul_(beta2).add_(lr, alpha=1 - beta2)
>   update = state["m"]  / state["vmean"]
> ```
> Therefore, our current implementation only involves 2 or 3 lines and we believe it is simple enough and can be trivially integrated into most codebases.
>
>
> ------------------------------------------------- **Additional response** --------------------------------------------
>
> **Integrating into the existing framework.** Following the idea in “Response (Part I)”, **the partition can be implemented automatically**. As a result, Adam-mini can be easily into existing training frameworks  and codebases including:
>
> 1. DDP (we used it for GPT-2 series pretrain)
> 2. FSDP (we used it for Llama series pretrain)
> 4. Torchtitan (we used it for Llama series pretrain)
> 5. Huggingface trainer (we used it for Llama 2-7B SFT and RLHF)
> 6. Llama factory (we used it for Llama 2-7B SFT and RLHF)
> 7. …
>
> For your models in these frameworks (including the suggested modern DiT model and SoTA diffusion models like DC-AE-Diffusion), you just need to call the optimizers with the following one-line code change (change "optimizer = AdamW()" to "optimizer = Adam-mini()"), and you will see memory reduction immediately.
>
> We will open-source our code once the paper is accepted.

---

> ### Author Response · Authors · 2024-11-21
> **Response (Part II)**
>
> **Q2. "It will be good to add discussions/insights about how to adapt hyper-parameters when switching from AdamW to Adam-mini."**
>
> Thanks for the suggestions. We kindly remind the reviewer that we have already discussed the choice of hyperparamters in the submission. This can be seen in the highlighted sentence in the submission (line 517, page 10 in the submission): **"In all our experiments, Adam-mini performs well using the same hyperparameters as AdamW."** (including learning rate, $\beta_1,\beta_2, \epsilon $, etc.) So we recommend no hyperparameter changes when switching AdamW to Adam-mini.
>
> In the submission script, we have also provided sensitivity analysis in Figure 12 (b)  to show that Adam-mini is not overly sensitive to hyperparams. So we believe hyperparam-adaptation will be easy for practitioners. Especially for the engineers who heavily relies on Adam, the switch to Adam-mini is almost no brainer.
>
> [[Click here to view Figure 12 (b)  in the submission: Adam-mini is not overly sensitive to hyperparams]](https://anonymous.4open.science/r/Adam-mini-ICLR-2025-rebuttal-new-014B/figure12_in_submission.png)
>
> **Q3. "In addition to training curves, it will be better to have comparisons after model convergence. "**
>
> Thanks for the great suggestions. In Appendix C.1  in the submission, we have already reported the final validation perplexity of Adam-mini and AdamW after the complete pre-training (by Chinchilla's law). For all the Llama models from 39M to 1B,  we found that Adam-mini reaches a slightly lower validation perplexity than AdamW. For the convenience of reviewer, we paste the Table 4 here.
>
> **[Table 4 in the submission: The final validation perplexity after the complete pre-training (by Chinchila's law)]**
>
> | Model Size | 39M      | 67M      | 102M     | 162M     | 271M     | 1B       |
> |------------|----------|----------|----------|----------|----------|----------|
> | Total Tokens | 1.02B    | 1.76B    | 2.67B    | 4.25B    | 7.10B    | 26.21B   |
> | AdamW      | 41.741   | 29.413   | 23.873   | 20.149   | 17.178   | 12.452   |
> | Adam-mini  | **38.696** | **27.093** | **23.038** | **19.645** | **17.035** | **12.372** |
>
>
> [[Click here to see Table 4 in the submission]](https://anonymous.4open.science/r/Adam-mini-ICLR-2025-rebuttal-new-014B/table4_in_submission.png)
>
>
>
> **Q4. "Additionally, it will be interesting to see results on diffusion transformers for diffusion model experiments."**
>
> Thanks for the great suggestions. Here is the performance of Adam-mini on the models that you suggested. We find Adam-mini performs on par with AdamW, with 50% memory less.
>
> To ensure a fair comparison, we use the recommended hyperparam setting in the codebase (learning rates, batchsize, etc.) and we do not specifically tune hyper-parameters for our methods.
>
> **[Table 1 (new)]: Adam-mini performs on par with AdamW on  DiT-XL-2-675M and DC-AE-Diffusion-500M.**
> | Domain | Model | Optimizer | Metric | 25% steps | 50% steps | 75% steps | 100% steps |
> |--------|--------|-----------|---------|------------|------------|------------|-------------|
> | Vision | DiT-XL-2-675M | AdamW | Train loss (↓) | 0.1605 | 0.1696 | 0.1607 | 0.1431 |
> | Vision | DiT-XL-2-675M | Adam-mini | Train loss (↓) | **0.1601** | **0.1693** | **0.1605** | **0.1430** |
> | Vision | DC-AE-Diffusion-500M | AdamW | Train loss (↓) | 0.2860| **0.2820** | 0.2800 | 0.2780 |
> | Vision | DC-AE-Diffusion-500M | Adam-mini | Train loss (↓) | **0.2860** | 0.2830 | **0.2800** | **0.2780**|
>
> The loss curves are shown in the links below.
>
> [[Click here to see the loss curve of DiT-XL-2-675M]](https://anonymous.4open.science/r/Adam-mini-ICLR-2025-rebuttal-new-014B/1117_dit.pdf)
>
> [[Click here to see the loss curve of DC-AE-Diffusion-500M]](https://anonymous.4open.science/r/Adam-mini-ICLR-2025-rebuttal-new-014B/DC-AE-Diffusion.pdf)
>
> **[Table 2 (new)]: Evaluation scores: Adam-mini performs on par with AdamW.**
> | Domain | Model | Optimizer | FID (↓) | Inception Score (↑)  |
> |--------|--------|-----------|---------|------------|
> | Vision | DiT-XL-2-675M | AdamW | 91.83 | 12.38 |
> | Vision | DiT-XL-2-675M | Adam-mini | **88.20** | **13.90** |
> | Vision | DC-AE-Diffusion-500M | AdamW | 34.72  | 41.79 |
> | Vision |  DC-AE-Diffusion-500M | Adam-mini | **33.15** | **44.38** |
>
>
>
> We would like to thank the reviewer again for the great suggestions. We will add these experiments to the paper.

---

> > ### Author Response · Authors · 2024-11-29
> > **We have updated the paper**
> >
> > Dear reviewer:
> >
> > Thanks again for the time and effort you have dedicated to reviewing our paper! We greatly appreciate your insightful comments and suggestions.
> >
> > In the revised paper, we have cited and included the experimental results for DiT and DC-AE-Diffusion. Please see the curves and table below.
> >
> > [[Screenshot for the revised paper (Table 6 in Appendix C.4): Adam-mini performs on par with AdamW on DiT and DC-AE-Diffusion]](https://anonymous.4open.science/r/Adam-mini-ICLR-2025-rebuttal-new-014B/revised_paper_non_llm.png)
> >
> >
> > [[Screenshot for the revised paper (Table 7 and Figure 17 in Appendix C.4): Adam-mini performs on par with AdamW on DiT and DC-AE-Diffusion]](https://anonymous.4open.science/r/Adam-mini-ICLR-2025-rebuttal-new-014B/revised_paper_figure17.png)
> >
> >
> > We have also responded your concern on "intergregation" in our previous responses. Hope our responses can help address your concern. Your comments and suggestions would be valuable for us in improving the paper! We would love to receive feedback from you.

---

> ### Author Response · Authors · 2024-11-26
> **Additional response to Q1 and looking foward to your feedback**
>
> Dear reviewer:
>
> We would like to provide an extra comment regarding your **Q1: “	1.	Integrating Adam-mini into existing training frameworks seems non-trivial. For example, according to Algo 2, we need to specify the partitions manually for large models. Is there any solution to automatically get a good partition given general Pytorch models?”.**
>
>  Thank you again for the thoughtful question. We believe that it is simple to integrate Adam-mini into existing frameworks and it can be done automatically. In our **“Response (Part I)”**, we have introduced how to implement Adam-mini on a simple linear model. We now provide more evidence that the integration is simple.
>
> **Recap on Response (Part I).** In our “Response (Part I)”, we have introduced how to implement Adam-mini on a simple linear model. The idea is that: we do NOT actually implement "first partition and then take mean". Instead, we take an equivalent operation: **we directly take mean over a certain dimension of the matrix $g \circ g$**. The later operation only involves 1 extra line and is reconciled with existing training frameworks.
>
>
> **Integrating into the existing framework.** Following the idea in “Response (Part I)”, the partition can be done automatically. As a result, Adam-mini can be easily into existing training frameworks  and codebases including:
>
> 1. DDP (we used it for GPT-2 series pretrain)
> 2. FSDP (we used it for Llama series pretrain)
> 4. Torchtitan (we used it for Llama series pretrain)
> 5. Huggingface trainer (we used it for Llama 2-7B SFT and RLHF)
> 6. Llama factory (we used it for Llama 2-7B SFT and RLHF)
> 7. …
>
> For your models in these frameworks (including the suggested modern DiT model and SoTA diffusion models like DC-AE-Diffusion), you just need to call the optimizers with the following one-line code change (change "optimizer = AdamW()" to "optimizer = Adam-mini()"), and you will see memory reduction immediately.
>
> We will open-source our code once the paper is accepted.
>
> We wonder if the above explanation fully addresses your concern? Your comments are valuable and we have put much effort into running new experiments to address your concern. We would love to receive feedback from you. We are more than happy to engage in further discussions.

---

> ### Author Response · Authors · 2024-11-30
> **Looking forward to your feedback**
>
> Dear reviewer:
>
> Thank you once again for the time and effort you have invested in reviewing our paper. We sincerely appreciate your insightful comments and suggestions. We have addressed your concerns and questions in the responses above. Please feel free to review them at your convenience.
>
> **Regarding experiments.** We kindly remind you that some of your questions are **already discussed in the submission**. As for your other questions, We also provide **new experimental results in the revised version**. We summarize the old and new results as follows.
>
> 1. **"how to adapt hyper-parameters when switching from AdamW to Adam-mini"**: This is already discussed in the submission. As we highlighted in line 517, page 10 in the submission: **"In all our experiments, Adam-mini performs well using the same hyperparameters as AdamW."**  (including learning rate, $\beta_1,\beta_2, \epsilon $, etc.). In addition, we have also investigated the hyperparam sensivity of Adam-mini and the results were presented in Figure 12. We find that Adam-mini is not overly sensivity to hyperparms.
>
>    [[Click here  to see Figure 12 in the submission]](https://anonymous.4open.science/r/Adam-mini-ICLR-2025-rebuttal-new-014B/figure12_in_submission.png)
>
>
>
> 2. **"comparisons after model convergence."**: This is already shown in the submission. Please see Table 4 in the submission (or Table 5 in the revised version). Compared with AdamW, we find that Adam-mini reaches slightly lower validation perplexity after the convergence of the complete pre-training run.
>
>    [[Click here to see Table 4 in the submission]](https://anonymous.4open.science/r/Adam-mini-ICLR-2025-rebuttal-new-014B/table4_in_submission.png)
>
>
> 3. **"results on diffusion transformers for diffusion model experiments."**: We conduct new experiments and the new results are provided in Table 6, Table 7, and Figure 17 in Appendix C.4 of the revised paper. We find that Adam-mini performs on par with AdamW on modern diffusion models such as DiT-XL-2 and DC-AE-Diffusion.
>
>     [[Click here to see the new results in the revised paper (Table 6 in Appendix C.4)]](https://anonymous.4open.science/r/Adam-mini-ICLR-2025-rebuttal-new-014B/revised_paper_non_llm.png)
>
>
>     [[Click here to see the new results in the revised paper (Table 7 and Figure 17 in Appendix C.4)]](https://anonymous.4open.science/r/Adam-mini-ICLR-2025-rebuttal-new-014B/revised_paper_figure17.png)
>
>
> We have also replied to your questions regarding "integration" and hope that we have adequately resolved your concerns. Your comments have been highly valuable to us, and we have put much effort into running new experiments to address your concern. We would love to receive feedback from you. If your concern is addressed, we humbly invite the reviewer to consider increasing the score.  Your support is deeply appreciated!

---

> > ### Comment · Reviewer_waYV · 2024-12-01
> >
> > I appreciate the authors' detailed responses. My concerns have been addressed. I decide to raise my score accordingly. Good work!

---

### Official Review · Reviewer_AcGY · 2024-11-04

**Soundness:** 3
**Presentation:** 3
**Contribution:** 3
**Rating:** 6
**Confidence:** 3

**Summary:**

Adam-mini presents a memory-saving approach to optimization, targetting large transformer models by cutting down the number of learning rates needed for each parameter. It does this by using Hessian-based partitioning to group parameters, so it assigns just one learning rate per block instead of for every single parameter, which cuts down memory use a lot. This setup holds up well in performance next to more traditional optimizers like AdamW, especially in terms of throughput and training speed, as it lowers GPU communication needs. Adam-mini shows it can work well on various deep learning tasks, optimizing models efficiently while saving both computing resources and training time.

**Strengths:**

1. Adam-mini cuts memory use by almost half (up to 50%) vs AdamW, by smartly dividing params based on Hessian structures and using block-level learning rates; this gives major memory savings without losing much performance on big language models.

2. By reducing comunication overheads between GPUs, Adam-mini boosts throughput by about 49.6% during Llama 2-7B training and lowers training time by 33%, which is really helpful in setups with limited resources.

3. Adam-mini provides a more efficient computational approach by skipping per-param learning rates while still matching AdamW's performance, closing the gap between memory efficiency and optimizer effectiveness across different neural net types, like LLMs and other deep learning architectures.

**Weaknesses:**

- This study mostly benchmarks on transformer models, leaving its effectivness on other architectures like CNNs and RNNs a bit underlooked; more benchmarking could either show it’s limitations or confirm its ability to adapt across other models types.

- The influence of the Hessian-based learning rate grouping on different gradient structures wasn’t deeply investigated, and comparing it with fully adaptive methods would make it’s efficiency clearer.

- Optimizer stability over long training durations hasn’t really been tested, so longer training runs would give more confidence in its reliability, especially for deep networks or tasks that need extended training.

- Adam-mini’s sensitivity to hyperparameters hasn’t been very explored, which might make tuning it more difficult, and more understanding here could show if it raises computational cost.

- Even though they briefly mention it, there’s no strong comparison with newer optimizers like Sophia and Lion; a head-to-head test would help show Adam-mini's specific advantages or where it could improve.

- Performance of the optimizer on tasks like fine-tuning and transfer learning is not talked about, and testing it on practical applications could confirm its strengths beyond just primary training.

- The grouping of parameters lacks deep mathematical support in the study, and adding a stronger theoretical basis would back up the partitioning method chosen and possibly lead to more improvements.

**Questions:**

1. How does Adam-mini make sure that its reduced learning rate partitions, relying on Hessian structures, can deliver a comparable optimization efficiency as full adaptive optimizers like Adam or AdamW, specially in dealing with sparse gradients and complex non-convex landscapes?

2. In which ways does Adam-mini maintain stability and avoid training instabilities, given its loss of per-parameter adaptation? Are there any specific mechanisms to prevent oscillations or divergence, particulary in deeper and more complex neural networks?

3. To what degree is the claimed memory savings consistent across varied model sizes and architectures, and does Adam-mini reliably reach faster convergence and throughput compared to the traditional adaptive optimizers?

4. How does Adam-mini approach hyperparameter sensitivity without the presence of per-parameter learning rates, and does this require additional hyperparameter tuning or raise computational costs?

5. What generalization ability does Adam-mini show when used on non-transformer models, such as CNNs or GNNs, and could its block-wise learning rate method limit performance in architectures where Hessian matrices are less structured?

6. How effectively does Adam-mini perform when compared to other recent memory-saving optimizers like Sophia or Lion, and is there potential for combining Adam-mini’s technique with finer adaptive methods to reach higher efficiency?

**Details Of Ethics Concerns:**

Last but not least, i could find the submitted paper on arxiv and github. i am considering this a violation of the iclr conference's policy on blind peer review process in which i can see the authors of the paper.

- https://arxiv.org/abs/2406.16793
Yushun Zhang∗12, Congliang Chen∗12, Ziniu Li12, Tian Ding2
,
Chenwei Wu3
, Yinyu Ye4
, Zhi-Quan Luo12, Ruoyu Sun†12
1The Chinese University of Hong Kong, Shenzhen, China
2Shenzhen Research Institute of Big Data
3 Duke University
4 Stanford University
{yushunzhang,congliangchen,ziniuli}@link.cuhk.edu.cn, dingtian@sribd.cn
cwwu@cs.duke.edu, yyye@stanford.edu, luozq@cuhk.edu.cn, sunruoyu@cuhk.edu.cn

---

> ### Author Response · Authors · 2024-11-21
> **Response (Part I)**
>
> We would like to thank the reviewer for the careful evaluation and insightful comments!  We respectfully provide our response as follows.
>
>
> **Q1: "The effectiveness (of Adam-mini) on other architectures like CNNs and RNNs a bit underlooked; more benchmarking could either show it’s limitations or confirm its ability to adapt across other models types." "What generalization ability does Adam-mini show when used on non-transformer models, such as CNNs or GNNs"**
>
>
> Thanks for the suggestions. We would like to clarify that we already reported the performance of Adam-mini on CNNs and GNNs **in the submission (Table  5 in Appendix C.2)**. For the convenience of the reviewer, we paste the Table  here.
>
> To ensure a fair comparison, we use the recommended hyperparam setting in the codebase (learning rates, batchsize, etc.) and we do not specifically tune hyper-parameters for our methods.
>
> **[Table 5 in the submission script: Adam-mini performs on par with AdamW on CNNs, GNNs, and Diffusion models.]**
>
> [[Click here to see Table 5 in the submission script]](https://anonymous.4open.science/r/Adam-mini-ICLR-2025-rebuttal-new-014B/table5_in_submission.png)
>
> | Domain | Model | Optimizer | Metric | 25% steps | 50% steps | 75% steps | 100% steps |
> |--------|--------|-----------|---------|------------|------------|------------|-------------|
> | Vision | Diffusion model | AdamW | Train loss (↓) | 0.0529 | 0.0497 | 0.0420 | 0.0394 |
> | Vision | Diffusion model | Adam-mini | Train loss (↓) | **0.0525** | **0.0495** | **0.0416** | **0.0388** |
> | Vision | ResNet18 | AdamW | Test acc (↑) | 0.6149 | 0.6478 | 0.6613 | 0.6669 |
> | Vision | ResNet18 | Adam-mini | Test acc (↑) | 0.6140 | **0.6501** | **0.6629** | 0.6667 |
> | Graph | GAT | AdamW | Val acc (↑) | 0.7277 | 0.7367 | 0.7399 | 0.7421 |
> | Graph | GAT | Adam-mini | Val acc (↑) | **0.7378** | **0.7394** | **0.7403** | **0.7429** |
> | Graph | GCN | AdamW | Val acc (↑) | 0.7347 | 0.7428 | 0.7379 | 0.7374 |
> | Graph | GCN | Adam-mini | Val acc (↑) | **0.7406** | 0.7427 | **0.7380** | **0.7423** |
>
>
> As for RNNs, we conduct new experiments of RNN on Name-Classification benchmark following the standard setup in [1].  We find that Adam-mini performs on par with AdamW.
>
> **[Table 1 (new)]: Adam-mini performs on par with AdamW on RNN.**
> | Domain | Model | Optimizer | Metric | 25% steps | 50% steps | 75% steps | 100% steps |
> |--------|--------|-----------|---------|------------|------------|------------|-------------|
> | NLP | RNN | AdamW | Val loss (↓) | **1.6173** | **1.3874**  | 1.3300 | 1.2797 |
> | NLP | RNN| Adam-mini | Val loss (↓) | 1.6177 | 1.4041 | **1.3036** | **1.2405** |
>
> [[Click here to see the loss curve of RNN.]](https://anonymous.4open.science/r/Adam-mini-ICLR-2025-rebuttal-new-014B/rnn_loss.pdf)
>
>
> For completeness, we also report the performance of Adam-mini some popular large-scale non-LLM tasks, including  DiT-XL-2-675M, and SoTA Diffusion models (DC-AE-Diffusion-500M), Vision Transformer (Swin-Transformer-88M). We train all models on ImageNet.
>
> **[Table 2 (new)]: Adam-mini performs on par with AdamW on Swin-Transformer, DiT, Diffusion models.**
> | Domain | Model | Optimizer | Metric | 25% steps | 50% steps | 75% steps | 100% steps |
> |--------|--------|-----------|---------|------------|------------|------------|-------------|
> | Vision | Swin-Transformer | AdamW | Val acc (↑) | **0.6290** |0.6940  | **0.7180**| **0.7310** |
> | Vision | Swin-Transformer | Adam-mini | Val acc (↑) | 0.6230 | **0.6960** | 0.7160 | 0.7300 |
> | Vision | DiT-XL-2 | AdamW | Train loss (↓) | 0.1605 | 0.1696 | 0.1607 | 0.1431 |
> | Vision | DiT-XL-2 | Adam-mini | Train loss (↓) | **0.1601** | **0.1693** | **0.1605** | **0.1430** |
> | Vision | DC-AE-Diffusion | AdamW | Train loss (↓) | 0.2860| **0.2820** | 0.2800 | 0.2780 |
> | Vision | DC-AE-Diffusion | Adam-mini | Train loss (↓) | **0.2860** | 0.2830 | **0.2800** | **0.2780**|
>
> The loss curves are shown here.
>
> [[The loss curve of Swin-Transformer]](https://anonymous.4open.science/r/Adam-mini-ICLR-2025-rebuttal-new-014B/swin-test-acc.pdf)
>
> [[The loss curve of DiT-XL-2]](https://anonymous.4open.science/r/Adam-mini-ICLR-2025-rebuttal-new-014B/1117_dit.pdf)
>
> [[The loss curve of DC-AE-Diffusion]](https://anonymous.4open.science/r/Adam-mini-ICLR-2025-rebuttal-new-014B/DC-AE-Diffusion.pdf)
>
> **[Table 3 (new)]: Evaluation scores: Adam-mini performs on par with AdamW.**
> | Domain | Model | Optimizer | FID (↓) | Inception Score (↑)  |
> |--------|--------|-----------|---------|------------|
> | Vision | DiT-XL-2 | AdamW | 91.83 | 12.38 |
> | Vision | DiT-XL-2 | Adam-mini | **88.20** | **13.90** |
> | Vision | DC-AE-Diffusion | AdamW | 34.72  | 41.79 |
> | Vision |  DC-AE-Diffusion | Adam-mini | **33.15** | **44.38** |
>
> We will include these results into the revised version. We wish the above results can show the effectiveness of Adam-mini on non-LLM tasks including CNNs, RNNs, GNNs, etc..
>
>
> [1] https://pytorch.org/tutorials/intermediate/char_rnn_classification_tutorial.html

---

> ### Author Response · Authors · 2024-11-21
> **Response (Part II)**
>
> **Q2. "The influence of the Hessian-based learning rate grouping on different gradient structures wasn’t deeply investigated, and comparing it with fully adaptive methods would make it’s efficiency clearer."**
>
> Thanks for the comment. We believe we have already investigated the influence of learning rate grouping with substantial depth. We will first restate the experiments in the submission and then present some new experiments.
>
> **Experiments in the submission.**  In the submission script, we investigate the influence of Hessian-based lr grouping via the following four steps.
>
> - **Step 1.** We conducted thorough  grid-search experiments on quadratic problems to show that "Adam is not effective on dense Hessian sub-blocks, so it can be simplified". This is shown in [[Figure 4]](https://anonymous.4open.science/r/Adam-mini-ICLR-2025-rebuttal-new-014B/figure4_in_submission.png) and [[Figure 5 in the submission]](https://anonymous.4open.science/r/Adam-mini-ICLR-2025-rebuttal-new-014B/figure5_in_submission.png).
>
> - **Step 2.** We also conducted thorough grid-search experiments on small Transformers to show that "a single learning rate can work well for each dense block in Transformers". This is shown in [[Figure  6 in the submission]](https://anonymous.4open.science/r/Adam-mini-ICLR-2025-rebuttal-new-014B/figure6_in_submission.png).
>
> - **Step 3.** We calculated the exact Hessian of real-world Transformers and design the learning rates grouping based on the Hessian of real-world Transfomers ([[Figure 7 in the submission]](https://anonymous.4open.science/r/Adam-mini-ICLR-2025-rebuttal-new-014B/figure7_in_submission.png)).
>
> - **Step 4.** We also provided negative results that Adam-mini does not perform well if not following the Hessian-based grouping principle ([[Figure 8 in the submission]](https://anonymous.4open.science/r/Adam-mini-ICLR-2025-rebuttal-new-014B/figure8_in_submission.png)). This shows the significant influence of Hessian-based grouping.
>
> **New experiments.**  For fully address your concern, we  further conduct the following new experiments on a small Transformer. We will show that "using single lr per block" is indeed sufficient for Transformers.
>
> The following Exp 1 and 2 extend the quadratic experiments in Figure 4 and 5 to  Transformers.
>
> - **Exp 1: Adam's lr is redundant on the dense Hessian subblock.** We take some small dense blocks in the Hessian of 1-layer Transformer and denoted as $H$. We compare  $\kappa(H)$ and $\kappa(D_{Adam} H)$ as in the paper. We find Adam is not effective in reducing the kappa of these blocks, and many lrs in Adam can be redundant.
>
>   **[Table 1 (new)]: Comparison of $\kappa(H)$ and $\kappa(D_{Adam} H)$ for the dense blocks in the Hessian of 1-layer Transformer.**
>   | Hessian Block |$\kappa(H)$ |  $\kappa(D_{Adam} H)$|
>   |--------|--------|-----------|
>   | 1st head in Query| 103.80 | 176.88 |
>   | 1st head in Key | 103.46 | 213.82 |
>   | 1st head in Value | 165.66 | 332.76 |
>   | 1st neuron in attn.proj | 39.92 | 94.56 |
>   | 1st neruon in MLP_fc1 | 22.04 | 70.92 |
>   | 1st neruon in MLP_c_proj | 63.85| 236.71 |
>
> - **Exp 2: single lr per block is sufficient.** We conduct the "block-wise GD" and we grid-search the lr for each block. The result is shown in the following figure. We find that block-wise GD outperforms AdamW.
> This extends the setting from the random quadratic problem in Figure 4.
>
>   [[Results on 1-layer Transformer: we can outperform Adam using a single lr per block.]](https://anonymous.4open.science/r/Adam-mini-ICLR-2025-rebuttal-new-014B/1118_rebuttal_blockwisegd_0.001.pdf)
>
> Combining Exp 1 and 2, we can see that Adam is redundant on the dense Hessian subblocks (Exp 1), and a single lr for each block can work well (Exp 2).  These experiments show that our conclusions on random quadratic problems can be extended to Transformers.
>
> We will include these experiments in the revised version. We are happy to further investigate the influence of Hessian-based grouping if the reviewer have any further suggestions.
>
> **As for the reviewer's comment "comparing it with fully adaptive methods would make it’s efficiency clearer.":** We kindly point out that we have already thoroughly  compared Adam-mini with AdamW in the submission, including:
>
>   1. GPT-2 pre-training, with size 125M, 330M, 1.5B.
>   2. Llama series pretraining, with sized 20M, 39M, 67M, 102M, 162M, 271M, 1B, 7B, 8B, and 13B
>   3. SFT (LoRA), SFT, and RLHF on Llama 2-7B.
>   4. ResNet and  Diffusion model on ImageNet
>   5. GCN and GAT on OGB dataset.
>
> In all these cases, Adam-mini perfroms on par or better than AdamW. We believe the comparison with "fully adaptive methods" is sufficient.

---

> ### Author Response · Authors · 2024-11-21
> **Response (Part III)**
>
> **Q3. "Optimizer stability over long training durations hasn’t really been tested."**
>
> We kindly disgree with the reviewer's comment on "long training durations hasn’t really been tested". In the submission, we have already reported the following experiments with long training durations:
>
> 1. **Complete Llama pre-train in the submission.** In Figure 12 and 14 in the submission, we conducted complete pre-training c on Llama 2 series (from 39M to 1B).  Each curve is for a  **complete pre-training run** under the definition of Chinchilla's law. Especially, **Llama 2-1B pre-train involves ~20B tokens and ~200k steps**, and we believe this falls into the category of "long duration training".
>
>    [[Figure 14 in the submission]](https://anonymous.4open.science/r/Adam-mini-ICLR-2025-rebuttal-new-014B/figure14_in_submission.png)
>
> 2. **Complete GPT-2 pre-train in the submission.**  In Figure 9 in the submission, we  also run GPT-2-125M and 330M on >20 billion tokens, which is **8x and  3x longer** than the Chinchilla's law.  Adam-mini is stable here. We believe all these experiments fall into the category of "long duration training".
>
>    [[Figure 9 in the submission]](
>     https://anonymous.4open.science/r/Adam-mini-ICLR-2025-rebuttal-new-014B/figure9_in_submission.png)
>
> We also provide the following new experiments.
>
> 3. **New experiments on GPT2-125M and Llama 2-134M with 50B tokens.** To provide extra evidence on the stability of Adam-mini, we conduct a new experiments on pre-training  GPT-125M and Llama 2-134M with 50B tokens. Both use lr = 6e-4, which is the standard choice). This is **20x longer** than the Chinchilla's law, and there are 100k and 400k steps in total, respectively. We find Adam-mini is always stable.
>
>     [[GPT-2-125M pretrain with 100k steps]
>     ](https://anonymous.4open.science/r/Adam-mini-ICLR-2025-rebuttal-new-014B/gpt2-125m-50btoken.png)
>
>     [[Llama 2-134M pretrain with 400k steps]](https://anonymous.4open.science/r/Adam-mini-ICLR-2025-rebuttal-new-014B/1120_134m_50btoken.pdf)
>
> Hope these experiments can address your concern about the stability of Adam-mini.
>
> **Final remark: potential source of concern.**  Correct us if wrong: your concern of "long duration" might come from our Llama 8B and 13B experiments, where the toal step is 10k and it does not follow Chinchila's law. We acknowledge that 10k step is indeed not a complete run, but the rest of training is too expensive. With our 4x A800-80GB, a complete pre-train run would take the following days:
> - Llama 8B:  13329 tokens/s. training on 8 * 20B tokens = **142 days.**
> - Llama 13B:  8378 tokens/s. training on 13 * 20B tokens =  **376 days.**
>
> So we need much more resources to complete the run. Nevertheless, we emphasize that it is not common to test optimizer at large scale like 8B and 13B. To our knowledge, we are not aware of any optimizer paper test their method at this scale. We believe these experiments still serve as extra evidence that Adam-mini is effective when scaling up models.
>
> **Q4. "Adam-mini’s sensitivity to hyperparameters hasn’t been very explored, which might make tuning it more difficult, and more understanding here could show if it raises computational cost." "and does this require additional hyperparameter tuning or raise computational costs?"**
>
> Thanks for the concern. We emphasize that the hyperparam-sensitivity is already explored in the script and Adam-mini does not require extra tuning code. We elaborate as follows.
>
>  1. We would like to point out that we have already reported the hyperparam sensitivity of Adam-mini in [[Figure 12 in the submission]](https://anonymous.4open.science/r/Adam-mini-ICLR-2025-rebuttal-new-014B/figure12_in_submission.png). We found that Adam-mini is not overly sensitive to hyperparams.
>  2.  Further, as we highlighted in the line 517, page 10 in the submission: **"In all our experiments, Adam-mini performs well using the same hyperparameters as AdamW."**  (including learning rate, $\beta_1,\beta_2, \epsilon $, etc.). So for all our experiments, there is no extra cost for tuning hyperparametres. For engineers, switching from AdamW to Adam-mini is almost no brainer.

---

> ### Author Response · Authors · 2024-11-21
> **Response (Part IV)**
>
> **Q5. "Even though they briefly mention it, there’s no strong comparison with newer optimizers like Sophia and Lion"**
>
> We will first discuss Lion and then discuss Sophia.
>
> **As for Lion:** we would like to  kindly remind the reviewer that we have thoroughly compared Lion and Adam-mini  in the submission (in Figure 16 in Appendix C4).   We carefully tuned the hyperparams of Lion and find that it is not easy to tune and **performs worse than Adam-mini on LLM tasks**. We restate the results here.
>
> **Results in the submission (Figure 16)**
> 1.  We consider Llama 2-20M pre-training. We use the default hyperparam $\beta_1 = 0.9, \beta_2 = 0.99$ and sweep over learning rates over [1e-5, 5e-5, 1e-4, 5e-4, 1e-3, 5e-3, 1e-2]. We find Lion consistenly performs worse. In particular, 1e-2 tends to  divergence and 1e-5 shows a clear sign of slow convergence.
>  2.  We consider Llama 2-20M pre-training. We change the hyperparam to $\beta_1 = 0.95, \beta_2 = 0.98$. This set of hyperparameters is claimed to be "helpful in mitigating instability during training" and is recommended by the authors of Lion (https://github.com/lucidrains/lion-pytorch).  We sweep over learning rates over [1e-5, 5e-5, 1e-4, 5e-4, 1e-3, 5e-3, 1e-2]. We find Lion consistently performs worse. In particular, 1e-2 tends to diverge and 1e-5 shows a clear sign of slow convergence.
> 2. We consider Llama-1B pre-training. We use the default hyperparameter $\beta_1 = 0.9, \beta_2 = 0.99$ and sweep over learning rates [ 1e-6, 1e-5, 1e-4, 1e-3, 4e-3]. We find Lion consistently performs worse. In particular, 4e-3 encounters loss spike and 1e-6 shows a clear sign of  slow convergence.
>
> [[Figure 16 in the submission:  Lion underperforms Adam-mini on Llama 2-20M]](https://anonymous.4open.science/r/Adam-mini-ICLR-2025-rebuttal-new-014B/figure16_in_submission.png)
>
> **New results of Lion on GPT-2-125M.** We further try Lion on GPT-2-125M pre-train, which is not tested before. For GPT-2-125M, AdamW uses 6e-4, so we try Lion with lr = [6e-5, 7e-5, 8e-5, 9e-5, 1e-4, 2e-4, 3e-4, 4e-4, 5e-4, 6e-4]. We find that  **Lion encounters loss spikes** on GPT-2-125M for all the lr candidates above.
>
> [[The loss curves of GPT-2-125M: Lion encounters loss spikes]](https://anonymous.4open.science/r/Adam-mini-ICLR-2025-rebuttal-new-014B/lion_gpt2.pdf)
>
> With all the effort above, we haven't managed to make Lion work, and it seems easy to encounter loss spikes on LLMs.
>
> In contrast, Adam-mini is much easier to use. In all our experiments, Adam-mini performs well using the same hyperparameters as AdamW. We believe that "easy adaptation of the hyperparameters" can serve as one advantage of Adam-mini over Lion, apart from the performance superiority.
>
> **As for Sophia:** As mentioned in Appendix A in the submission, Sophia is designed for fast convergence, not for memory saving, and it uses the same memory as AdamW. As a result, Sophia is an orthogonal method to Adam-mini, and the two methods can be combined to get "Sophia-mini", which can further improve the efficiency.
>
> Since you mentioned Sophia, we tried combining Sophia and Adam-mini in the rebuttal phase. We find that the resulting "Sophia-mini" works quite well on GPT-2-125M, and Sophia-mini generates almost the same curve as Sophia.
>
> [[Results of Sophia-mini on GPT-2-125M pre-train]](https://anonymous.4open.science/r/Adam-mini-ICLR-2025-rebuttal-new-014B/1120_sophia_valloss.pdf)
>
> **Q6. "Performance of the optimizer on tasks like fine-tuning and transfer learning is not talked about, and testing it on practical applications could confirm its strengths beyond just primary training."**
>
> We kindly disagree with the reviewer's comment on "fine-tuning ... is not talked about". We would like to point out that we have already conducted the following fine-tuning experiments in the submission.
> 1. SFT (LoRA) on Llama 2-7B
> 2. full-parameter SFT on Llama 2-7B
> 3. RLHF on Llama 2-7B.
>
>  We find Adam-mini performs on par or better than AdamW.  [[See Figure 11 and Table 3 in the submission]](https://anonymous.4open.science/r/Adam-mini-ICLR-2025-rebuttal-new-014B/figure11_in_submission.png).
>
> As for transfer learning, we haven't found suitable open-source tasks yet. We are happy to try if the reviewer can suggest some.
>
> **Q7. "The grouping of parameters lacks deep mathematical support in the study."**
>
> Thanks for the suggestion. Though most of our analyses are based on numerical observations, we are driven by fundamental optimization theory. For instance, a major motivation of Adam-mini comes from analyzing the "effectiveness of diagonal preconditioner", which is a 50-year-old theoretical open question.  Though we did not conclude this hard topic using rigorous theoretical bound, we provide new numerical results on this topic. We believe it is worth an independent paper to explore the relevant theory.

---

> ### Author Response · Authors · 2024-11-21
> **Response (Part V)**
>
> **Q8. "How does Adam-mini make sure that its reduced learning rate partitions, relying on Hessian structures, can deliver a comparable optimization efficiency as full adaptive optimizers like Adam or AdamW, specially in dealing with sparse gradients and complex non-convex landscapes?"**
>
> Adam-mini incurs no performance drop because we  **only remove  the redundant lrs in Adam**, and we keep the effective ones. In the submission script, we identify the redundant lrs by the following steps.
>
> **Step 1:** By revisiting the theory and experiments in Collobert 2004 [2], we realized that Hessian of neural nets have near-block-diagonal structure.
>
> **Step 2:** Through extensive experiments on quadratic problems and small Transformers, we found that Adam has many redundent lrs within the dense subblock. See  [[Figure 4]](https://anonymous.4open.science/r/Adam-mini-ICLR-2025-rebuttal-new-014B/figure4_in_submission.png), [[Figure 5]](https://anonymous.4open.science/r/Adam-mini-ICLR-2025-rebuttal-new-014B/figure5_in_submission.png), and [[Figure  6 in the submission]](https://anonymous.4open.science/r/Adam-mini-ICLR-2025-rebuttal-new-014B/figure6_in_submission.png).
>
> **Step 3:** We investigated the Hessian structure of Transformers and  partition the parameters by the smallest dense Hessian block.   See [[Figure 7 in the submission]](https://anonymous.4open.science/r/Adam-mini-ICLR-2025-rebuttal-new-014B/figure7_in_submission.png). Then we use one lr per block.
>
> The above steps ensure we remove the redundant lrs in Adam, and keep the effective ones. We observe no performance drop as long as we reduce lr by the above steps, even for the mainstream complex models (LLMs with size 7B, 8B, 13B, etc.).
>
> We agree that neural networks can have sparse gradient and complex landscape, but it also falls into the category of "near block-diagonal Hessian structure".  Thus, the redundancy of Adam's design still exists, and Adam-mini still can losslessly reduce the (redundant) lrs.
>
> [2] Ronan Collobert. Large scale machine learning. Thesis, Université de Paris VI, 2004
>
> **Q9. "In which ways does Adam-mini maintain stability and avoid training instabilities, given its loss of per-parameter adaptation? Are there any specific mechanisms to prevent oscillations or divergence, particularly in deeper and more complex neural networks?"**
>
> Adam-mini maintains stability by carefully partitioning the parameters by the smallest dense subblock in Hessian. Following this design principle,  we only remove the *redundant* lrs in Adam, and keep the effective ones. We showed that our partition principle can effectively prevent loss spike (as evident in [[Figure 8]](https://anonymous.4open.science/r/Adam-mini-ICLR-2025-rebuttal-new-014B/figure8_in_submission.png) in the submission), and we indeed did not observe any loss spike in current mainstream complex models (LLMs with size 7B, 8B, 13B, etc.). We are not aware of any potential obstacles affecting its effectiveness in future models.
>
>
> **Q10. "To what degree is the claimed memory savings consistent across varied model sizes and architectures, and does Adam-mini reliably reach faster convergence and throughput compared to the traditional adaptive optimizers?"**
>
> **As for memory saving:** We save  50% memory over AdamW for all mainstream LLMs (7B, 13B, etc.).  The 50% memory saving also holds for all larger models.
>
> **As for throughput:** We believe that "higher throughput" will be widely observed since Adam-mini always allows larger batch size than AdamW. The specific speed-up gain will depend on hardware and infrastructure. For instance, on 2x A800-80GB, Adam-mini allows 4x larger batch size and we observe 30% wall-clock speedup of Adam-mini over AdamW (Figure 1 in the submission).
>
> **Q11. "What generalization ability does Adam-mini show when used on non-transformer models, such as CNNs or GNNs, and could its block-wise learning rate method limit performance in architectures where Hessian matrices are less structured?"**
>
> We have responded to the question on "CNNs and GNNs" in the response to **Question 1** above.
>
> As for "architectures where Hessian matrices are less structured",  we haven't explored such tasks so far. However, according to the results in Collobert 2004 [2], most neural nets' will **indeed have highly structured Hessian**, so we have shown that Adam-mini is indeed effective on a wide range of architectures.  In particular, Collobert 2004 [2] shows that the special Hessian structure comes from the  Cross-Entropy loss and the layer-by-layer design, and both of these features widely exist in modern neural nets from CNN to GPTs (also see our analysis in Eq. (1) in the submission).
>
> [2] Ronan Collobert. Large scale machine learning. Thesis, Université de Paris VI, 2004
>
> We hope the above explanation fully addresses your concern. Your feedback is highly valuable. We would love to receive feedback from you. We are more than happy to engage in further discussions.

---

> ### Author Response · Authors · 2024-11-30
> **Looking forward to your feedback**
>
> Dear reviewer:
>
> Thanks again for the time and effort you have dedicated to reviewing our paper! We greatly appreciate your insightful comments and suggestions. We have replied to your concerns and questions in our responses above. Please check at your convenience.
>
> **Regarding experiments**, we kindly remind you that some of them are **already presented in the submission**. To fully address your concern, we also provide **many new experimental results**. We summarize the old and new results as follows.
>
> 1. **"effectiveness (of Adam-mini) on other architectures"**: This is already shown in the submission. Please see Table 5 in the submission or see the revised Table 6 and Figure 17 below.
>
>     [[Table 6 in the revised paper (with new results on DiT, Diffusion, and ViT.)]](https://anonymous.4open.science/r/Adam-mini-ICLR-2025-rebuttal-new-014B/revised_paper_non_llm.png)
>
>     [[Figure 17 in the revised paper (with new results on DiT, Diffusion, and ViT.)]](https://anonymous.4open.science/r/Adam-mini-ICLR-2025-rebuttal-new-014B/revised_paper_figure17.png)
>
>     [[New results on RNN]](https://anonymous.4open.science/r/Adam-mini-ICLR-2025-rebuttal-new-014B/rnn_loss.pdf)
>
>
> 2. **"The influence of the Hessian-based learning rate grouping on different gradient structures wasn’t deeply investigated"**: We add new results in Appendix C.1. below.
>
>     [[New results in Appendix C.1 in the revised paper]](https://anonymous.4open.science/r/Adam-mini-ICLR-2025-rebuttal-new-014B/revised_paper_Q23.png)
>
> 3. **"Optimizer stability over long training durations"**: This is already shown in the submission. Please see the Figure 9 and 14 in the submission or see the new results below.
>
>
>       [[Figure 9 in the submission (revised for visualization)]](
>     https://anonymous.4open.science/r/Adam-mini-ICLR-2025-rebuttal-new-014B/figure9_in_paper.png)
>
>       [[Figure 14 in the submission]](https://anonymous.4open.science/r/Adam-mini-ICLR-2025-rebuttal-new-014B/figure14_in_submission.png)
>
>
>     [[New results on GPT-2-125M pretrain with 100k steps]](https://anonymous.4open.science/r/Adam-mini-ICLR-2025-rebuttal-new-014B/gpt2-125m-50btoken.png)
>
>     [[New results on Llama 2-134M pretrain with 400k steps]](https://anonymous.4open.science/r/Adam-mini-ICLR-2025-rebuttal-new-014B/1120_134m_50btoken.pdf)
>
>
> 4. **"Adam-mini’s sensitivity to hyperparameters"**: This is already shown in the submission. Please see Figure 12  and the discussion in the submission.  Further, as we highlighted in the line 517, page 10 in the submission: **"In all our experiments, Adam-mini performs well using the same hyperparameters as AdamW."**  (including learning rate, $\beta_1,\beta_2, \epsilon $, etc.).
>
>    [[Figure 12 in the submission]](https://anonymous.4open.science/r/Adam-mini-ICLR-2025-rebuttal-new-014B/figure12_in_submission.png)
>
>
> 5. **"comparison with newer optimizers like Sophia and Lion"**: This is already discussed in the submission. Please see the discussion in Appendix A (Sophia) and Figure 16 (Lion) in the submission, or please see the new results below.
>
>     [[New results for Lion in the revised Appendix C.6 (Part I)]](https://anonymous.4open.science/r/Adam-mini-ICLR-2025-rebuttal-new-014B/revised_paper_Q6_1.png)
>
>     [[New results for Lion in the revised Appendix C.6 (Part II)]](https://anonymous.4open.science/r/Adam-mini-ICLR-2025-rebuttal-new-014B/revised_paper_Q6_2.png)
>
>
>
>
>     [[New results for Sophia-mini: Sophia can be orthogonally combined with Adam-mini to get Sophia-mini. Sophia-mini performs the same as Sophia but with less memory]](https://anonymous.4open.science/r/Adam-mini-ICLR-2025-rebuttal-new-014B/1120_sophia_valloss.pdf)
>
>
>
> 6. **"tasks like fine-tuning"**. This is already shown in the submission. Please see Figure 11 in the submission.
>
>     [[Figure 11 in the submission]](https://anonymous.4open.science/r/Adam-mini-ICLR-2025-rebuttal-new-014B/figure11_in_submission.png)
>
> We have also replied to your other questions as well. Hope we have addressed your concerns.  Your comments are valuable and we have put much effort into running new experiments to address your concern, and we would love to receive feedback from you. If your concern is addressed, we humbly invite the reviewer to consider increasing the score.  Your support means a lot to us!

---

### Author Response · Authors · 2024-11-27
**Summary of changes and some further clarifications**

Dear reviewers and ACs:

We sincerely thank all reviewers and ACs for their efforts in reviewing our paper and providing valuable feedback. We have carefully addressed each concern raised by the reviewers and made corresponding revisions to our paper. We highlight the revision in blue. Please feel free to check. We summarize the key changes below.

1. **New experiments in  Appendix C.4 (Page 20) and Appendix C.7 (Page22)**. We show the scalability of Adam-mini on more diverse tasks.  In particular, following the comments by Reviewer AcGY, waYV, 8UzS, we present 6 new non-LLM experiments on 3 modern models (Swin-Transformer, DiT-XL-2, and DC-AE-Diffusion) in Appendix C.4. Additionally, following the comments by Reviewer AcGY and Reviewer rJgs, we add new LLM experiments on GPT-2 and Llama models Appendix C.7. On all these diverse tasks, we demonstrate that Adam-mini performs on par with AdamW and outperforms other baselines.
2. **New experiments in Appendix C.1 (Page 18-19)**. Following Q2 by Reviewer AcGY; Q2 and Q3 by Reviewer rJgs, we add more experiments on Transformers to show that “one lr per block” is sufficient for Transformer training. These experiments extend our previous random quadratic experiments in Figure 4 and 5.
3. **New experiments in Appendix C.2 (Page 19)**. Following Q4 by Reviewer rJgs, we add more ablation studies and find that "mean(v)" performs better than other candidate quantities such as "norm(v)" or "max(v)".
4. **New discussions in Section 2.2 (Page 5).** Following  Q1  and Q5 by Reviewer rJgs, we re-organize Section 2.2  and introduce “Adam-mini (General form)” in Section 2.2.   Hope this will make the method easier to understand and shows the generality of Adam-mini.
5. **New discussions in Section 2.3 (Page 7).** Following  Q4 by Reviewer rJgs, we add new discussions  on “why mean (v)” in Section 2.3.


We believe that the revisions outlined above have improved the quality of our paper, thanks to the insightful comments from the reviewers. We hope these new results can address the reviewers' concerns and further strengthen the contributions of our work. Thank you once again for your valuable feedback.


------

**Further clarification on our contribution.**  During the rebuttal phase, we found that most reviewers focused on validating the performance of Adam-mini, and overlooked some other aspects of our contribution.   We sincerely appreciate the insightful and valuable comments by the reviewers, and we have spent great efforts to show the effectiveness of Adam-mini. However, more broadly speaking, we believe our contribution is not just proposing a new optimizer.  Apart from optimizer design, we believe that our following findings are also useful to the community.

1. We, for the first time, reveal the special structure of Transformer Hessian, including the block-diagonal structure and the non-equal-sized blocks in Querys, Keys, Values, and MLPs. We believe these findings can boost our understanding for Transformers.
2. We extensively investigate the behaviors of Adam on different Hessian structures (e.g., how the behaviors of Adam change when the eigenvectors of Hessian rotate away from the standard basis).  We believe these findings can boost our understanding of Adam and can motivate future theoretical research on this topic.
3. Adam-mini can be viewed as an example of "how to use the problem structure (findings in 1 & 2) to design practical optimizers."  We believe the implications of 1 & 2 are quite broad and they will be useful to motivate more and more future optimizers, which could either be more memory-efficient or achieve faster convergence.

We believe the above findings are also useful to the community and shall not be overlooked. Finally,  we would like to thank all the reviewers and ACs again for all the great effort and time that you devote to our paper.

---

### Meta-Review · Area_Chair_pzwK · 2024-12-19

**Metareview:**

Adam-mini introduces a memory-efficient optimization approach for large-scale transformer models by employing Hessian-based partitioning to group parameters and assigning a single learning rate to each block, significantly reducing memory usage. The method is conceptually reasonable, easy to implement, and supported by extensive experimental validation. However, the paper underwent substantial revisions and numerous experiments were added during the rebuttal phase, creating a noticeable gap between the initial submission and the revised version. Additionally, a key issue remains unclear: under FSDP training, model weights are often partitioned across GPUs in a manner that does not align with layer-wise boundaries. For example, weights from a single layer can be split across multiple GPUs, causing the Hessian blocks used for parameter grouping to become fragmented and influenced by the framework’s weight partitioning scheme. This introduces a potential inconsistency between the theoretical motivation of Hessian-based grouping and its practical implementation. While the rebuttal partially addressed this concern, further clarification is needed in the final version. Nonetheless, the paper tackles an important challenge in reducing the memory overhead of adaptive optimization methods for large language models, enabling improved token throughput, which justifies a positive recommendation for acceptance.

**Additional Comments On Reviewer Discussion:**

During the rebuttal period, the reviewers with negative stances raised concerns regarding the fairness of the comparisons in the paper, specifically questioning the motivation, methodology, and selection of baselines. In response, the authors conducted extensive additional experiments and provided detailed clarifications. These additions addressed the reviewers’ concerns partially, leading the reviewers to raise their scores. I agreed with reviewer rJgs at first, but the additional experiments were considered favorably in my final decision.

---

### Decision · Program_Chairs · 2025-01-22

Accept (Poster)